# Large-scale crossbar arrays based on three-terminal MoS$_2$ memtransistors

Thomas F. Schranghamer [1], Andrew Pannone[1], Jishnu M. Kumar[1], Dev Krishna Thiyyadi Baiju[1], Chen Chen[2], Thomas McKnight[3,4], Sean Tadekawa[5], Evan Haines[5], Richard Ordonez[5], Cody Hayashi[5], Joan M. Redwing [2,3,4] & Saptarshi Das [1,2,4,6] ✉

Memristive crossbar architectures are promising as efficient, low-power inference engines for edge AI applications. However, inputs with minor differences often yield similar outputs, requiring additional processing methods such as confidence scoring, feedback mechanisms, crossbar redundancy, or hybrid analog-digital approaches to resolve. These methods can be impractical for resource-limited edge devices. In contrast, three-terminal memtransistors can dynamically tune conductance via gate control, effectively resolving similar outputs and enhancing separability without retraining. Here, we present dense, large-scale crossbar array architectures incorporating up to 2048 MoS$_2$ memtransistors per array, achieving >92% yield across multiple arrays while individual memtransistors exhibit write energies as low as ~0.2 fJ, maintain read margins up to 10$^5$, and offer a projected retention exceeding three years. These architectures demonstrate the ability to resolve inference ambiguities through gate modulation without the need for costly retraining or reprogramming. We also validate their performance by successfully classifying handwritten digits from the MNIST database. Finally, we benchmark the performance of MoS$_2$ memtransistors against other 2D material-based architectures and project their potential compared to state-of-the-art AI accelerators. We believe that this work furthers the ongoing development of in-memory processors for decentralized edge applications and that future studies aimed at reducing device-to-device variation and improving long-term non-volatile memory would only enhance inference capabilities.

With the growing demand for the implementation of artificial intelligence (AI) in decentralized edge applications, proactive co-design of software and hardware has become critical[1,2]. This has led to a distinct paradigm shift towards in-memory computing architectures inspired by biological neural networks (BNNs) for the development of next-generation artificial neural network (ANN) accelerators and inference engines[1,3–10]. In this regard, memristive crossbar array architectures modeled on biological synapses have already attracted significant attention for their ability to parallelly implement rapid, energy-efficient vector-matrix multiplication (VMM) operations through simple employment of Ohm's and Kirchoff's laws across their constituent non-volatile memory (NVM) cells. Crossbar array architectures

[1]Engineering Science and Mechanics, Penn State University, University Park, PA, USA. [2]2D Crystal Consortium Materials Innovation Platform, Materials Research Institute, Penn State University, University Park, PA, USA. [3]Materials Science and Engineering, Penn State University, University Park, PA, USA. [4]Materials Research Institute, Penn State University, University Park, PA, USA. [5]Naval Information Warfare Center Pacific, Pearl City, HI, USA. [6]Electrical Engineering and Computer Science, Penn State University, University Park, PA, USA. ✉e-mail: sud70@psu.edu

therefore offer significant advantages for accelerating ANNs in hardware, most notably for tasks that require extensive matrix computations, such as image or speech classification, with extant experimental demonstrations including image classification[9,11,12], image/signal processing[13,14], video classification[15], etc. While such real-time, real-world data processing applications are of great interest for implementation at the network edge, e.g., in self-driving cars or automated factories, numerous challenges at the software and hardware levels still need to be addressed before data-center-independent, latency-free inference can be realized. Of these, perhaps the most persistent is the tendency for inputs (e.g., images) with minor differences to result in outputs with high degrees of classification confusion between them. This problem is only exacerbated as datasets become more complex, necessitating additional processing through methods such as confidence scoring[16–18] and recursive feedback mechanisms[19–21] at the software level or crossbar redundancy[22,23] and hybrid analog-digital approaches[24–26] at the hardware level. While each of these approaches possess their own benefits, they often add significant energy-/time-/area-overhead to inference engine operation, making them impractical for resource-constrained edge applications. Instead, an alternative path to efficient edge computation may lie in emulation of high-level neural/synaptic mechanisms at the base computation unit (i.e., NVM cell)[27,28].

While resistive memories[9,29,30], charge-trapping (FLASH) memories[31–33], and ferroelectric memories[34–36], among others, have already seen serious investigation as NVM cell technologies, one group of emerging alternatives that currently shows promise for in-memory/sensor computing through back-end-of-line (BEOL) integration with standard complementary metal-oxide-semiconductor (CMOS) technology is two-dimensional (2D) materials. Monolayer $MoS_2$, a semiconductor from the transitionmetal dichalcogenide (TMDC) family, has attracted particular attention for use in three-terminal memtransistor devices for NVM applications. These devices exploit monolayer $MoS_2$'s sensitivity to charge variation stemming from its high surface-area-to-volume ratio to realize distinct, non-volatile conductance levels (memory states) using charge-trapping gate stacks[37–39] or metallic floating gate architectures[14,40,41] akin to traditional Si-based FLASH memories. Notably, the presence of a third (gate) terminal in the memtransistor architecture sets it apart from the two-terminal NVMs commonly used in crossbar architectures by allowing for the application of a separate gate bias during crossbar operation, screening the electric field across the device channel and effectively tuning the conductance state (weight) of the device. Extant studies of three-terminal $MoS_2$-based memtransistors primarily employ the gate terminal as a programming input during weight assignment[14,40–42], shifting the threshold and enabling fixed excitatory or inhibitory behavior for later inference operations. However, from a biological perspective, this capability is highly reminiscent of the activities performed by modulatory neurons (interneurons) in heterosynaptic plasticity[28,43,44]. In this process, the stimulation of one neuron causes a change in the strength (weight) of the connections between other, inactivated, neurons. A study of repeated heterosynaptic modulation events in biological neural networks has found that this can lead to persistent changes in synaptic connections[45], thus contributing to the formation of long-term memories[46]. Developing methods of implementing heterosynaptic plasticity in hardware has therefore been deemed an important goal for realizing next-generation neuromorphic systems with greater energy efficiency[28]. For crossbar architectures, this offers a method of tuning prewritten conductance states by applying different gate biases and could potentially be used to exaggerate differences between otherwise similar classes, enhancing the separability of highly confused outputs without the need for resource-intensive data processing.

In this article, we present crossbar array architectures utilizing up to 2048 memtransistors based on monolayer $MoS_2$ integrated with a charge-trapping memory stack as NVM cells. We achieve a high yield of >92% across multiple crossbar array demonstrations, with constituent devices displaying write energies as low as ~ 0.2 fJ while retaining read margins (ratios between programmed and erased states) as high as $10^5$ and projected retentions exceeding three years. We also investigate the influence of cell area on device- and array-level operations, ultimately demonstrating equivalent performance at information densities of up to 1.94 Mb/cm² (assuming 1-bit operation). We then demonstrate how our architecture can be partitioned to accurately achieve simple inference operations (e.g., shape detection) in hardware. Uniquely, we also demonstrate how the three-terminal nature of $MoS_2$-based memtransistors can be exploited to dynamically tune conductance states during inference to modulate weights, and therefore outputs, without changing the underlying conductance state in a manner similar to modulatory synapses; notably, this process allows for weight tuning and task-specific separation of high confusion objects without requiring retraining of the network or reprogramming of individual weights, energy-/time-intensive operations that are difficult to realize in edge applications with strict energy/latency constraints. Finally, we experimentally demonstrate image classification of handwritten digits from the Modified National Institute of Standards and Technology (MNIST) database[47] using a 64 × 32 (2 kb) $MoS_2$ memtransistor crossbar array.

## Results & discussion
### Crossbar overview
A schematic representation of the base computational unit (NVM cell) utilized for our primary crossbar array architecture is shown in Fig. 1a. Control of individual NVM cells within the arrays is achieved through dedicated 5/20 nm Ti/Pt gate lines (word lines), which are deposited first onto a commercial $p^{++}$-Si/SiO₂ substrate using electron-beam (e-beam) evaporation. Atomic layer deposition (ALD) is then used to grow a charge-trapping gate stack consisting of a 15 nm $Al_2O_3$ blocking layer, a 3 nm $HfO_2$ trapping layer, and a 7 nm $Al_2O_3$ tunneling layer; the efficacy and operational principles of this gate stack for memory applications have already been reported on previously[37–39], with a summarized description being provided below. The dielectric stack is then etched near the gate contact pads to provide electrical access for later operation, and large-area, monolayer $MoS_2$ grown via metal-organic chemical vapor deposition (MOCVD) is transferred onto the substrate using a wet transfer process before being etched into the discrete channels for each constituent device. Details regarding the quality/characterization of the as-grown $MoS_2$ film are provided in Supplementary Note 1. Source and drain (bit) lines are then formed by evaporating 40/30 nm Ni/Au, with subsequent evaporation of 90 nm $Al_2O_3$ and 40/70 nm Ti/Au to form insulating crosspoints and conductive bridges, respectively, for the source/drain overlaps. Note that all access lines were purposefully kept narrow (1.5 μm wide) compared to prior array-level demonstrations to avoid large parasitic capacitances and increase the crosspoint breakdown voltage. Further details regarding the fabrication process can be found in the Methods section.

An optical image of a representative 16 × 10 array is shown in Fig. 1b along with a zoomed-in image showcasing several constituent NVM cells, each possessing a cell area of only 676 μm². Here, cell area is defined as the area (*Length × Width*) of the repeating unit for a given crossbar design. This allows us to achieve an information density, defined as the amount of storable information (i.e., weights) in bits per cm², of ~ 0.15 Mb/cm². Note that, for the purposes of this work, 1-bit (two-state) operation is assumed in all cases for the purposes of calculating information density. To further investigate the scalability of our $MoS_2$ memtransistor technology for achieving higher information density targets, more compact crossbar array architectures were also designed and tested; the most compact design featured cell areas of 51.5 μm², providing an ultimate information density of ~ 1.94 Mb/cm² while retaining comparable performance to our primary design.

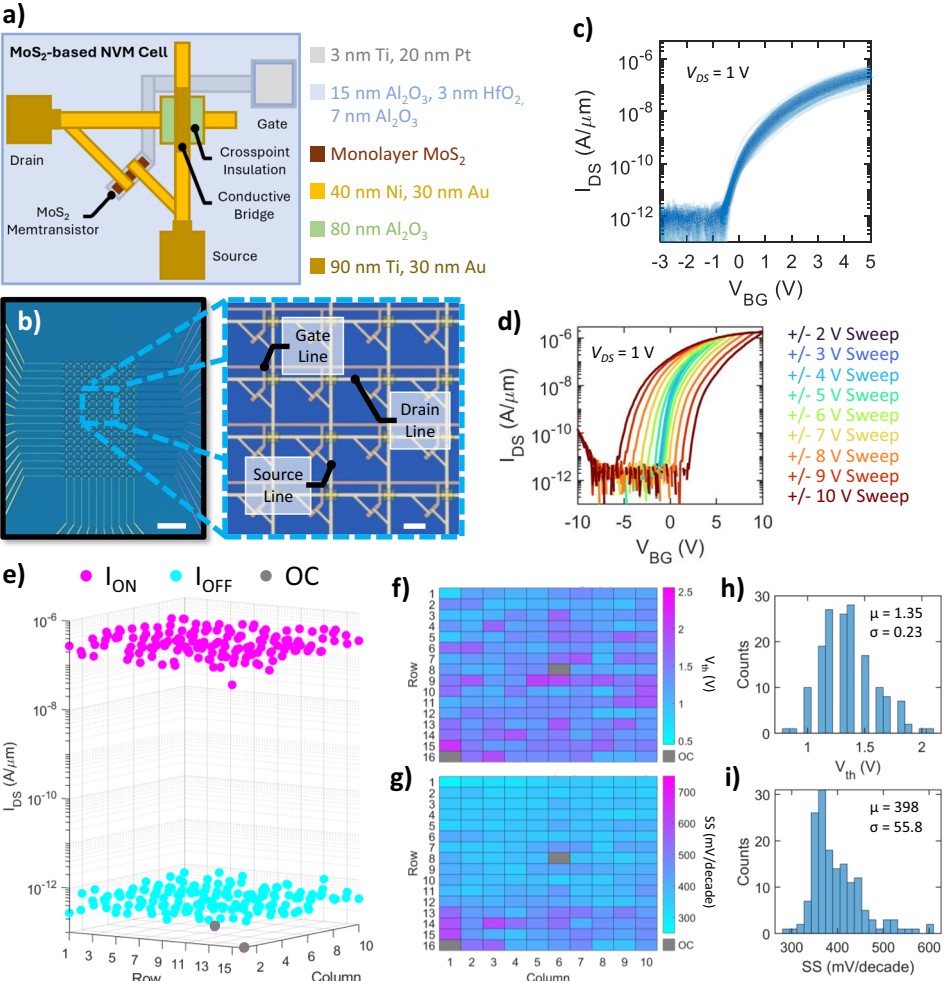

**Fig. 1 | Overview of MoS2-Memtransistor-based Crossbar Array Architecture.**
**a** Schematic of the basic non-volatile memory (NVM) cell design containing a single memtransistor. Terminals are split into row (drain and gate) and column (source) accesses, with each cell occupying an area of 676 $\mu m^2$. **b** Optical image of a representative 16 × 10 crossbar array based on the design shown in (**a**). Zoomed-in image shows constituent memtransistors with the drain, source, and gate lines labeled. Scale bar denotes 100 $\mu m$ (10 $\mu m$ for zoomed-in). **c** Overlapped transfer characteristics, i.e., drain-to-source current ($I_{DS}$) versus back-gate voltage ($V_{BG}$), taken at a drain-to-source voltage ($V_{DS}$) of 1 V from constituent memtransistors. **d** Hysteresis loops for a representative device from (**b**, **c**) taken at $V_{DS} = 1$ V. Multiple

loops were taken by sweeping $V_{BG}$ over the noted ranges to determine the presence/size of the memory window at different gate voltages; a sizable memory window of ~ 7 V can be noted for the +/− 10 V sweep. **e** Three-dimensional scatter plot showing the distribution of ON-state and OFF-state currents taken at $V_{DS} = 1$ V, denoted as $I_{ON}$ (pink) and $I_{OFF}$ (cyan), respectively; devices/cells marked in gray registered as an open circuit (OC) when measured. Notably, 158/160 devices were found to work (98.8% yield). **f**, **g** Maps of (**f**) threshold voltage ($V_{th}$) and (**g**) sub-threshold slope (SS) across the array. OC devices are marked. **h**, **i** Histograms of (**h**) $V_{th}$ and (**i**) SS. The means (μ) and standard deviations (σ) are noted for each.

Further details regarding this investigation and an overview of the results for each design are provided in Supplementary Note 2, 3. While the achieved integration/information densities reported here still lie far short of the > 1 Tb/cm² necessary for full-scale emulation of BNNs in hardware[26], they nonetheless compare favorably to other extant 2D-material-based array-level demonstrations; while some reports have demonstrated cell areas down to ~ 5 $\mu m^2$ per cell, the ultimate array size of those works was significantly less than that shown here. A comprehensive comparison of this work with existing 2D literature[14,30,48-54] is provided in Supplementary Note 4.

The memory capabilities/operations of the $Al_2O_3$/$HfO_2$/$Al_2O_3$ gate dielectric stack incorporated into each $MoS_2$ memtransistor (NVM cell) can be easily understood through the band diagrams shown in Supplementary Note 5. As has been studied elsewhere[37-39], the NVM capabilities of each device/cell are governed by the trapping/detrapping of charge carriers in the middle $HfO_2$ charge-trapping layer when bias pulses of sufficient magnitude are applied to the back-gate in program/erase (write) operations. These trapped charges screen the electric

field across the $MoS_2$ channel, shifting the threshold voltage ($V_{th}$) of the device and allowing for the realization of distinct conductance (memory) states at a given read voltage ($V_{read}$). The polarity of the applied pulse determines which charge carriers are trapped/detrapped, with holes (electrons) being trapped when negative (positive) pulses are applied and vice versa, while the pulse time and magnitude determine the change in carrier concentration in the charge-trapping layer (i.e., the size of the $V_{th}$ shift, or memory window). For the purposes of this work, the memory window ($\Delta V_{th}$) is defined as the difference in $V_{th}$, taken at a constant current level, between programmed/erased states. Note that many extant 2D-material-based crossbar demonstrations[48,55-58] program (write) devices through application of a drain bias pulse to prompt defect migration across the semiconducting channel, thereby changing the threshold (conductance) of the device; depending on the biases required, with lower biases always preferable for greater energy efficiency, this can lead to destructive read operations similar to those that affect two-terminal memristor technologies[55,56]. In addition, defect engineering of the channel, e.g.,

through low-energy Ar plasma treatment[59], may be required to provide a sufficiently high concentration of defects for appreciable memory effects to be seen at low voltages, thus complicating the fabrication process. Conversely, for our demonstration, programming is performed through the application of bias pulses to the gate terminal to promote charge-trapping in the dielectric stack, thereby shifting the threshold by screening the electric field across the channel in a manner analogous to traditional FLASH memories[37–39]. As the drain terminal is decoupled from the programming mechanism, this minimizes the risk of destructive read events. Furthermore, unlike defect-migration-based memories, charge-trapping memories do not require any pre-processing (i.e., defect engineering) of the channel material, relying solely on the as-fabricated dielectric stack for memory operation.

The transfer characteristics, i.e., drain-to-source current ($I_{DS}$) versus $V_{BG}$, taken at a drain-to-source voltage ($V_{DS}$) of 1 V are shown in Fig. 1c for the representative array shown in Fig. 1b; the hysteresis characteristics of a representative device when swept between varying $V_{BG}$ ranges are shown in Fig. 1d, demonstrating how the memory window, and thus the achievable conductance states, of a single device evolve as increasing $V_{BG}$ is applied. A histogram of the memory window extracted across 32 memtransistors is shown in Supplementary Note 6. The mean and standard deviation were found to be 7.30 V and 0.40 V, respectively. The read margins from these devices under the same conditions were also extracted, with the results also being shown in Supplementary Note 6. As can be seen, most devices achieved read margins $>10^4$, with all achieving read margins $>10^3$, which is more than sufficient for the implementation of inference operations in hardware[9,14]. Furthermore, as shown in Fig. 1e–h for the large-scale (16 × 10) array presented in Fig. 1b, the crossbar array architectures discussed here display impressively high yields of >95% with reasonably good device-to-device uniformity in ON/OFF-state performance, $V_{th}$, and subthreshold slope. As discussed in Supplementary Note 2, 3, and 6, these metrics were found to be similar across each design tested, indicating minimal influence from design on $\Delta V_{th}$, read margin, energy consumption, or yield. For comparison, please note that previous large-scale demonstrations only reached reported yields of ~ 83% or lower for single arrays[14,30,48–54] (see Supplementary Note 4 for comparative analysis of yield with previous 2D-based array-level demonstrations). As defective NVM cells can have a significant negative impact on crossbar operations, and therefore on ANN tasks such as image classification, signal processing, etc., the high yield and reliability of the $MoS_2$-based memtransistors demonstrated in this work therefore represent a notable step forward for the eventual adoption of 2D-material-based NVMs in large-scale logic accelerators. The array architectures demonstrated in this work (both here and in the Supplemental) also compare favorably in terms of write energy, ranging from tens of pJ (base) to below 1 fJ (peak); all write energies were estimated using the equation $Energy = Time \times Current \times Voltage$, which is commonly used to estimate switching energy for NVMs[60]. The terms "base" and "peak" included in the assessment of our work refer to the pulse time (write time), with base referring to our typical pulse time of 100 ms (used for all demonstrations unless otherwise states) and peak referring to our minimum confirmed pulse time of 1 μs (see Supplementary Note 7). While several other array-level works have also utilized ultrafast write times, our NVM capabilities being controlled by the gate means that the write current of our devices is limited to the gate leakage current, which remained in the region of several tens of pA even at the largest gate biases applied (~ 10 V), thus allowing for severely reduced energy expenditure. Note that, due to experimental setup limitations, the minimum reliable read time was constrained to 4 μs.

One significant advantage of utilizing memtransistors as NVMs as opposed to more mature resistive memory technologies is their multi-terminal nature. This allows for devices to be modulated through both the non-volatile program/erase operations mentioned previously and through dynamic applications of $V_{BG}$ during read operations. By varying $V_{BG}$ during reads, the conductance state (weight) programmed into a device can be selectively tuned either higher (through application of a positive gate bias) or lower (through application of a negative gate bias) as needed without expending the energy/time required for a switching event; as read margin will vary depending on where it is assessed in the memory window, this approach may also be used to increase or decrease the ratio between states as needed for different applications. Notably, no charge-trapping is required for modulating weights in this manner, meaning that this can be performed at the default read speed (clock speed) of the system being used to conduct logic operations in the array.

The ability of our memtransistor architecture to modulate preprogrammed conductance states through the application of a modulatory back-gate bias ($V_{mod}$) is shown in Supplementary Note 8. It can be seen that >6 distinct output levels can be produced from a single binary state, thus offering a mechanism to emulate intermediate synaptic strengths without the time/energy costs and variability and drift issues commonly associated with analog memory writes. As we will discuss in further detail below, this ability to control conductance states (weights) independently from the application of program/erase operations provides an extra degree of freedom when performing inference operations, allowing us to dynamically potentiate or depress weights to more clearly differentiate between outputs in high confusion. This capability stands to improve system-level performance in multiple ways. First, it increases output separability, which is especially valuable for tasks where resolution must be preserved. Second, gate tuning serves as a runtime calibration mechanism that compensates for device non-idealities or variation without requiring reprogramming or retraining. Together, these elements indicate that, unlike multilevel resistive memories, which require iterative tuning and exhibit degraded endurance at intermediate states, our method retains the simplicity and stability of binary memory while possessing some of the flexibility of analog systems. While not intended to replace multilevel storage where precision is essential, dynamic gate modulation provides an efficient alternative for inference tasks where robustness, low power, and tolerance to device variation are more critical than static precision.

To implement this approach in hardware, inference ambiguities would first need to be identified via a readout layer, which monitors the difference between dedicated output nodes ($\Delta I_{out}$). If this value falls below a predefined threshold (e.g., 5% of the maximum difference) for a given input, the output (inferred digit) would be flagged as uncertain. This function could be achieved using a comparator circuit. Following inference, flagged outputs would be subjected to dynamic modulation. A control unit would then trigger a lookup into an external memory table storing modulatory gate biases ($V_{mod}$) for the relevant output nodes. These $V_{mod}$ values would be preassigned during network training based on the object classes being analyzed and reflect coarse bias adjustments tailored to enhance the separability of commonly confused class pairs (e.g., within +/− 2 V for the demonstrations discussed in this work). This allows the system to resolve ambiguous cases through a single re-read, rather than reprogramming device states or performing retraining. The peripheral overhead for this process would be modest, consisting of the comparator circuit for $\Delta I_{out}$ evaluation and flagging, the external lookup table (with the size depending on the number of high confusion classes and desired $V_{mod}$ resolution), and a shared DAC or multiplexer to apply $V_{mod}$ to the array. Of course, it is important to acknowledge that DAC/ADC operation significantly contributes to the energy consumption of neuromorphic accelerator chips[25]; however, as modulation would only be triggered for a subset of inference cases, often defined by task-specific importance (e.g., hazard detection in surveillance), the power draw for this use case should not meaningfully contribute to the overall power budget.

## Crossbar operation

While the crossbar architectures developed as part of this work show high yield, uniformity, and information density, thus demonstrating improvements in several areas critical for the realization of 2D-based logic accelerators, assessing their potential for in-memory computing applications requires a stringent investigation of the programming behavior of individual NVMs and the accuracy of logic operations performed across the arrays. For each array design, individual devices may be independently accessed by selecting the terminals of the corresponding row (drain and gate terminals) and column (source terminal). Once a device is accessed, operations are split into two categories: programming/erasing and reading. For program/erase operations, two distinct schemes were investigated as part of this work, the schematics of which are shown in Supplementary Note 9. In the first, a bias pulse is simply applied to the corresponding gate access line with a time 100 ms and a magnitude 10 V, adjusting the carrier concentration in the charge-trapping layer as outlined in the NVM discussion above. While straightforward and effective, this scheme was found to affect all devices in a given row, making it ill-suited to programming distinct conductance states (weights) in individual memtransistors for logic operations; however, it may still be utilized effectively for setting each NVM in an array to a low conductance state (LCS) ahead of weight assignment through the application of large positive gate biases ($V_{erase}$) in an initialization step. This primarily serves to standardize conductance states between different logic operations performed during array testing. This effectively suppresses persistent low-resistance paths across the array, limiting the number of conductive routes through which sneak currents can propagate[61]. Please note that this approach does not eliminate sneak paths entirely. Even devices in the OFF-state possess finite resistance, and under large array sizes, half-selected paths can contribute measurable leakage. However, with OFF-state conductances only reaching a few pS and read voltages maintained at 1 V, the resulting leakage currents are typically below 10 pA, well within the tolerances of common sensing circuits. For applications wherein the weights of a given array are held constant, this step may not be required. Furthermore, as shown in Supplementary Note 10, experimental testing has verified that each gate access line is sufficiently electrostatically isolated to prevent programming of devices in adjacent rows, thus allowing for subsets of devices to be initialized without affecting the entire array.

An overview of the second program/erase scheme, which may be used to assign/adjust individual weights, is also provided in Supplementary Note 9. Here, a half-biasing scheme is utilized to maximize the gate-to-source voltage ($V_{GS}$) across the targeted cell while minimizing $V_{GS}$ across all other cells on the same gate access line, thus allowing for single device programming. The gate access line corresponding to the targeted cell is subjected to a given programming bias of half its typical value ($V_{program}/2$) while its corresponding source line is held to the same bias at opposite polarity (-$V_{program}/2$), thus providing a $V_{GS}$ across the cell equal to $V_{program}$. All other source lines are held at $V_{program}/2$, ensuring that $V_{GS} = 0$ V for all other cells on the same gate access line, thus preventing programming of unwanted cells. As negative gate biases are applied for programming events, the targeted cell and all others in the same row are held in the OFF-state throughout, leading to low programming currents and ensuring a low programming energy expenditure. While the application of large source biases could lead to sneak path current through cells in other rows, this is partially suppressed through the initialization step placing all devices into the OFF-state; sneak path current can be further suppressed by applying a small (< 5 V) negative bias to every other gate access line to ensure that devices in those rows are turned completely OFF without programming them. After programming, read operations may be performed to probe the conductance state of a device by applying $V_{read}$ to the respective gate access line while also applying a given $V_{DS}$ ($V_{DS,read}$) to

its respective drain terminal. For the purposes of this work, $V_{read} = 0$ V and $V_{DS,read} = 1$ V unless otherwise stated.

To verify the ability of our MoS$_2$-memtransistor-based crossbar array architecture to function as an inference engine, a proof-of-concept inference demonstration aiming to classify simple shapes in hardware was first performed. It should be noted that for this demonstration, and all others discussed in this work, the analog weights achievable using our MoS$_2$ memtransistor architecture[37–39] are deliberately binarized between logic "0" and logic "1". Binary neural networks (BNNs) have been recognized as well-suited choices for edge platforms where compute energy and memory footprint must be tightly controlled[62,63]. For such use cases, binarization serves three purposes: (i) it minimizes the need for high-resolution analog memory states, which require costly write-verify schemes to program[64] and are prone to error due to the low read margin between different states[20,63]; (ii) it simplifies on-chip multiply–accumulate operations to XNOR and bitwise logic[62,65], reducing active energy costs, while also terminating sneak paths through unwanted cells[61], reducing passive energy costs; and (iii) it provides a robust pair of initial states that can be dynamically modulated through an applied gate bias to increase inference accuracy, as will be discussed later. An overview of this demonstration is shown in Fig. 2. A set of five 5 × 5 binary (black-and-white) shapes was chosen, and software-based training was used to prepare a weight map corresponding to the conductance states of a 25 × 5 sub-array of a 100 × 10 (1 kb) memtransistor crossbar array, with a schematic representation of this process and optical images of the array being shown in Fig. 2a. See Supplementary Note 11 for details on the characterization of the sub-array used in this demonstration. As with the smaller arrays discussed above, we noted near 100% yield, read margins > 10$^3$, and no overt spatial trends for common device performance metrics across both the 25 × 5 sub-array and the full 100 × 10 array. The memory performance of devices in this array was also assessed separately from the smaller arrays detailed above to accurately determine what effects, if any, scaling has on our program/erase capabilities, with the results being presented in Supplementary Note 12, 13. From these results, we note minimal spatial influence on hysteresis (i.e., read margin and memory window) or retention. Furthermore, a fit of ~1 h retention measurements indicates that the projected long-term retention exceeds three years, in part due to the intrinsically-OFF nature of the devices meaning that change in read margin over time will predominantly depend on the change in ON-state conductance. A corresponding investigation of minimum write/read time stability for the same array is presented in Supplementary Note 14. The endurance of the MoS$_2$ memtransistor architecture utilized in this work was also assessed. Supplementary Note 15 shows the results for a representative NVM cell over 500 program/erase cycles ($V_{program} = -10$ V, $V_{erase} = +10$ V, $t_{pulse} = 100$ ms). Some degradation in the memory ratio between the ON-state and the OFF-state was seen during cycling; however, the read margin remains > 10$^2$, indicating suitable endurance for NVM cell application. The retention of the same device over a period of ~1 h, read at a constant $V_{DS}$ of 1 V, is shown in Supplementary Note 16 before and after endurance testing. While the read margin shows similar degradation to the cycling results, little-to-no change in long-term retention can be seen, indicating minimal write disturbance from repeated program/erase cycles.

An image classification test was then conducted, with each of the 5 × 5 shapes transformed into 25 × 1 input vectors before being sequentially fed to the hardware array as voltage pulse trains through the drain terminal; for each input shape/vector, white pixels were expressed as a 0 V input while blue pixels were expressed as a 1 V input voltage. As inputs were applied, multiply-accumulate (MAC) operations were conducted along each column (node) of the array, each of which were indexed to a particular shape during training. The output currents from each node were taken and compared to determine which shape corresponded to the maximum current value, as this

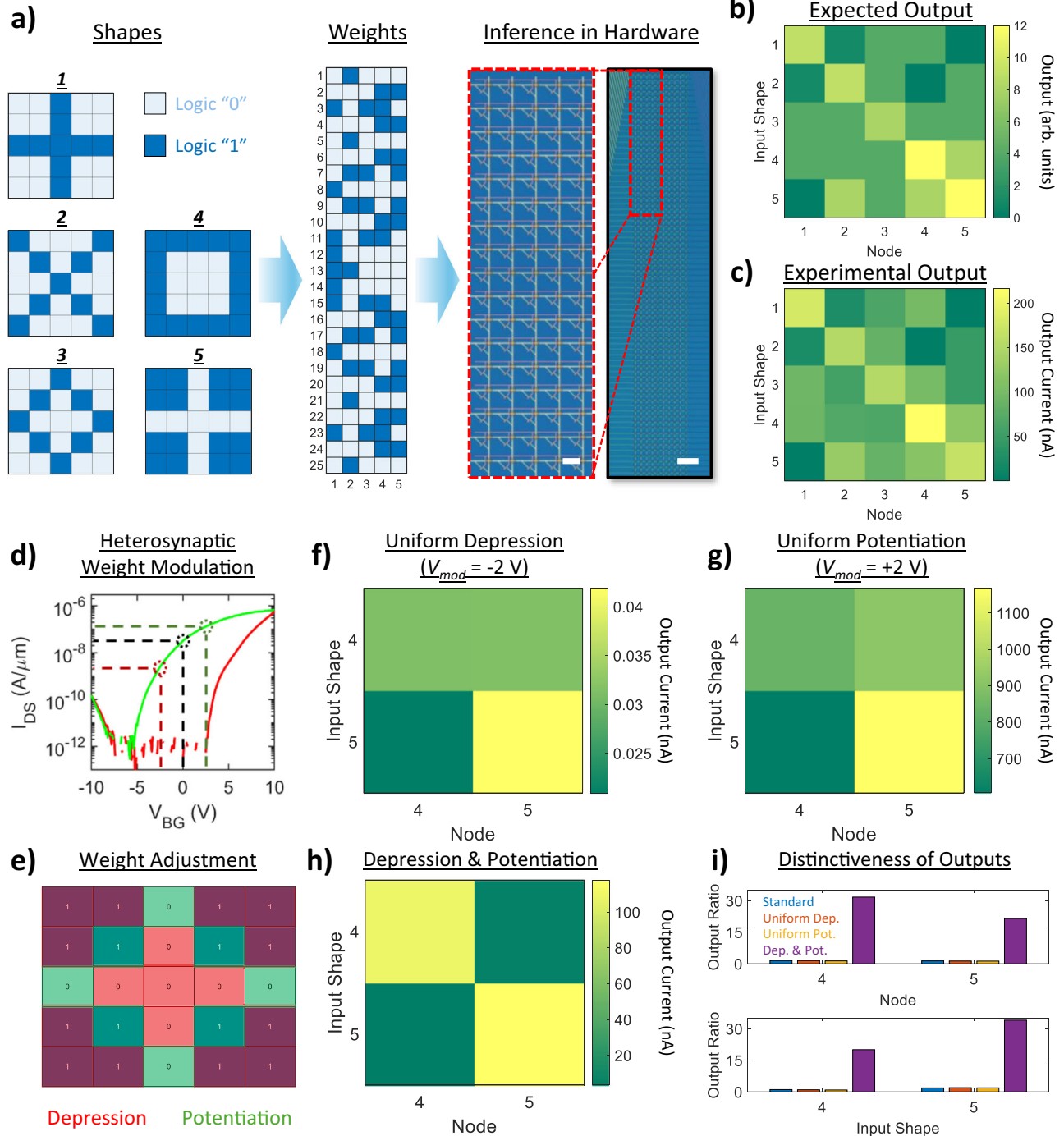

**Fig. 2 | Simple Shape Identification using an MoS₂-Memtransistor-based Crossbar Array. a** Schematic overview of shape identification. First, five 5 × 5 binary (logic "0" or "1") shapes were chosen. A 25 × 5 array of binary weights was generated and mapped to a 25 × 5 sub-array of a 100 × 10 hardware array (pictured), with logic "0" represented by a low conductance state (OFF-state) and logic "1" represented by a high conductance state (ON-state). Scale bar denotes 100 μm (20 μm for zoomed-in). **b** Expected and (**c**) experimental results when the binary shapes in (**a**) were fed to the sub-array. **d** Demonstration of heterosynaptic weight modulation, wherein varying the gate bias inside of a conductance state (ON-state, green curve) allows for the potentiation (increase, green dotted line) or depression (decrease, red dotted line) of the effective conductance state (weight) seen by the array. **e** Schematic

for heterosynaptic weight modulation for shapes "4" and "5" from (**a**), wherein weights corresponding to areas of high similarity/difference between the shapes are dynamically depressed/potentiated through the application of negative/positive gate biases ($V_{mod}$). **f**–**h** Effects of dynamic depression and potentiation on identification between "4" and "5". Uniform application of (**f**) a depressive $V_{mod}$ ($V_{mod} = -2$ V) and (**g**) a potentiating $V_{mod}$ ($V_{mod} = +2$ V) were found to have a minimal effect on distinctiveness. **h** When depressive and potentiating $V_{mod}$ were selectively applied, a high distinctiveness between outputs was achieved. **i** Bar plots showing the output ratios between nodes for each shape applied (top) and input shapes for each node tested (bottom) for each case, i.e., standard (blue), uniform depression (orange), uniform potentiation (yellow), and combined depression and potentiation (purple).

represents the result for the inference operation following the max-current sensing rule[9]. The results in Fig. 2b, c show the expected (simulated) and experimental results, respectively, for the shape identification task outlined above. From the simulation, the maximum output for each node should correlate with its respective shape, which was indeed the case for the experimental results; each shape was correctly identified at its respective node and was clearly differentiable from all other inputs. These results thus verified the array's ability to perform hardware-based image classification tasks using our $MoS_2$-memtransistor-based crossbar array architecture.

While the results are accurate in this case, it can be clearly seen that certain input/node combinations result in a higher degree of uncertainty, which could potentially lead to misclassification in more complex datasets or noisy real-world scenarios. While weights could be optimized for these scenarios on-site through retraining/reprogramming, these are resource-intensive operations that are more-than-likely infeasible in most emerging edge applications for hardware-based logic accelerators. Here, the ability to dynamically modulate the weights of memtransistors through the application of $V_{mod}$ to the gate terminal, as discussed above and further illustrated through the demonstration shown in Fig. 2d, provides an additional method of verifying inference results not achievable in traditional two-terminal resistive memories. An overview of this proposed verification scheme is presented in Fig. 2e–i. For a given set of possible classes, in this case the five binary shapes shown in Fig. 2a, the areas of greatest difference/similarity for each possible combination are identified and corresponding mask matrices generated, with the example schematic in Fig. 2e showing how heterosynaptic weight modulation could be used in the case of shapes "4" and "5". Image classification is then performed as detailed above, with the caveat that the ratio between two outputs must be over a given threshold value. If the results lie below that threshold, the mask matrix corresponding to that shape combination is recalled as an input vector, and a second MAC operation is performed as it is fed to the gate terminals as voltage pulses aiming to either potentiate (positive) or depress (negative) the weights representing areas of high difference or similarity, respectively. The output currents from the relevant nodes are then assessed to determine the true input following the max-current sensing rule.

Outputs for this process when conducted on the $25 \times 5$ sub-array discussed/detailed above for differentiation of shapes "4" and "5" are shown in Fig. 2f–h. Uniform application of a depressive $V_{mod}$ ($V_{mod} = -2$ V) and a potentiating $V_{mod}$ ($V_{mod} = +2$ V) were both found to have a minimal effect on distinctiveness, though the output current registered at each node for each input shape did vary as expected (from tens of pAs for uniform depression to hundreds of nAs for uniform potentiation). However, when depressive and potentiating $V_{mod}$ were selectively applied as outlined in Fig. 2e, high distinctiveness between outputs was achieved, thus reducing confusion. The output ratios between nodes for each shape applied and input shapes for each node tested are shown in Fig. 2i for each case presented, i.e., the standard inference case (Fig. 2c), uniform depression (Fig. 2f), uniform potentiation (Fig. 2g), and combined depression and potentiation (Fig. 2h). It can be clearly seen that application of $V_{mod}$ to selectively depress or potentiate areas of high or low similarity, respectively, results in a significant increase in the ratio between outputs for each combination tested, confirming the ability of heterosynaptic weight modulation to enhance accuracy and reduce confusion. This indicates that our memtransistor-based crossbar architectures have potential for implementing non-traditional logic operations in hardware by leveraging their multi-terminal nature.

It should also be noted that this dynamic modulation approach has significant implications for energy-efficient computing. Per standard FET operation, this dynamic modulation requires charging the oxide (back-gate) capacitance ($C_{ox}$), which is ~3.8 fF per device. The energy cost per modulation can then be approximated using the equation $E_{mod} = \frac{1}{2} \cdot C_{ox} \cdot V_{mod}^2$. For the demonstrations used in this work, modulatory voltages between $+/-2$ V are used, resulting in an additional energy expenditure of <8 fJ per device. In contrast, standard weight retraining approaches (digital or analog) require either reprogramming memory cells or propagating weight updates through the network. For resistive memories, iterative write-verify schemes consume the majority of a crossbar array's energy budget[64]; this can consume upwards of 100 pJ per device[66] depending on the type of memory, number of bits, etc. In addition, retraining often increases latency and may not be feasible in constrained edge environments. This disparity supports the use of three-terminal modulation as an efficient alternative for fine-tuning network behavior with minimal overhead.

Expanding our efforts to more closely investigate real-world ANN applications, we then performed single-layer neural network inference operations aiming to classify handwritten digits from the MNIST datasets. An overview of this demonstration is shown in Fig. 3. For this demonstration, a $64 \times 32$ (2048 devices, 2 kb) crossbar array was prepared, with an optical image of the final chip containing the array being shown in Fig. 3a–c along with zoomed-in optical images of the 2 kb array and constituent NVM cells. Raman spectroscopy was used to assess the quality and uniformity of the $MoS_2$ film used for the fabrication of the large-scale array. These results are presented in Supplementary Note 17. The Raman spectra was taken across nine points corresponding to the corners, sides, and center of the array with a 532 nm laser. The mean $E_{2g}^1$ (in-plane) and $A_{1g}$ (out-of-plane) peak locations were found to be 385.26 cm$^{-1}$ and 403.15 cm$^{-1}$, respectively, with a mean peak separation of 17.89 cm$^{-1}$. The standard deviations for each of these parameters was comparatively low (0.30 cm$^{-1}$, 0.30 cm$^{-1}$, and 0.42 cm$^{-1}$, respectively), indicating good spatial uniformity of the $MoS_2$ film.

A simple binary classification scheme was then tested with the goal of classifying all ten digits (0-9) comprising the MNIST dataset. As schematically represented in Fig. 3d, MNIST images ($28 \times 28$ pixels) were first downscaled to $13 \times 13$ pixels using bicubic interpolation and transformed into $169 \times 1$ vectors to serve as inputs. To simplify the experimental demonstration, the greyscale pixels of the MNIST dataset were also binarized to either 0 or 1 during this process, making each image black and white; further discussion on this preprocessing, and on its effects on inference accuracy, can be found in the Methods section. Software-based training was then used to prepare a binary weight map from a training dataset consisting of 10,000 MNIST images, as shown in Fig. 3e. Testing of the simulated single-layer network was then performed using a dataset consisting of 1000 randomly selected binarized $13 \times 13$ images, curated such that each digit (0–9) was represented exactly 100 times to ensure fair assessment of the network's ability to infer different digits. The results from this simulation-based MNIST inference test are presented by the confusion matrix in Fig. 3f. An overall accuracy of 88.1% was achieved; while digits such as "0", "1", and "6" demonstrated high classification accuracy, the overall accuracy was constrained by the networks limited ability to classify digits such as "5", which demonstrated a high degree of confusion with "8" and "3". While non-ideal, this relatively low overall accuracy is to be expected due to the nature of the problem presented, i.e., single-layer inference across a large dataset consisting of ten highly compressed and binarized handwritten digits. Following the simulation, the binary weights of the single-layer network were mapped to corresponding conductance states in a $64 \times 30$ sub-array of the 2 kb array shown in Fig. 3a–c, with weights of "1" being mapped to a conductance state of ~50 nS and weights of "0" being mapped to the OFF-state conductance (a few pS). A map of the final conductance states across the sub-array is shown in Fig. 3g; here, NVM cells marked NaN either display an open circuit or high gate leakage, both of which are believed to stem from damage to the corresponding horizontal drain line. Note that, despite the loss of five complete rows to these

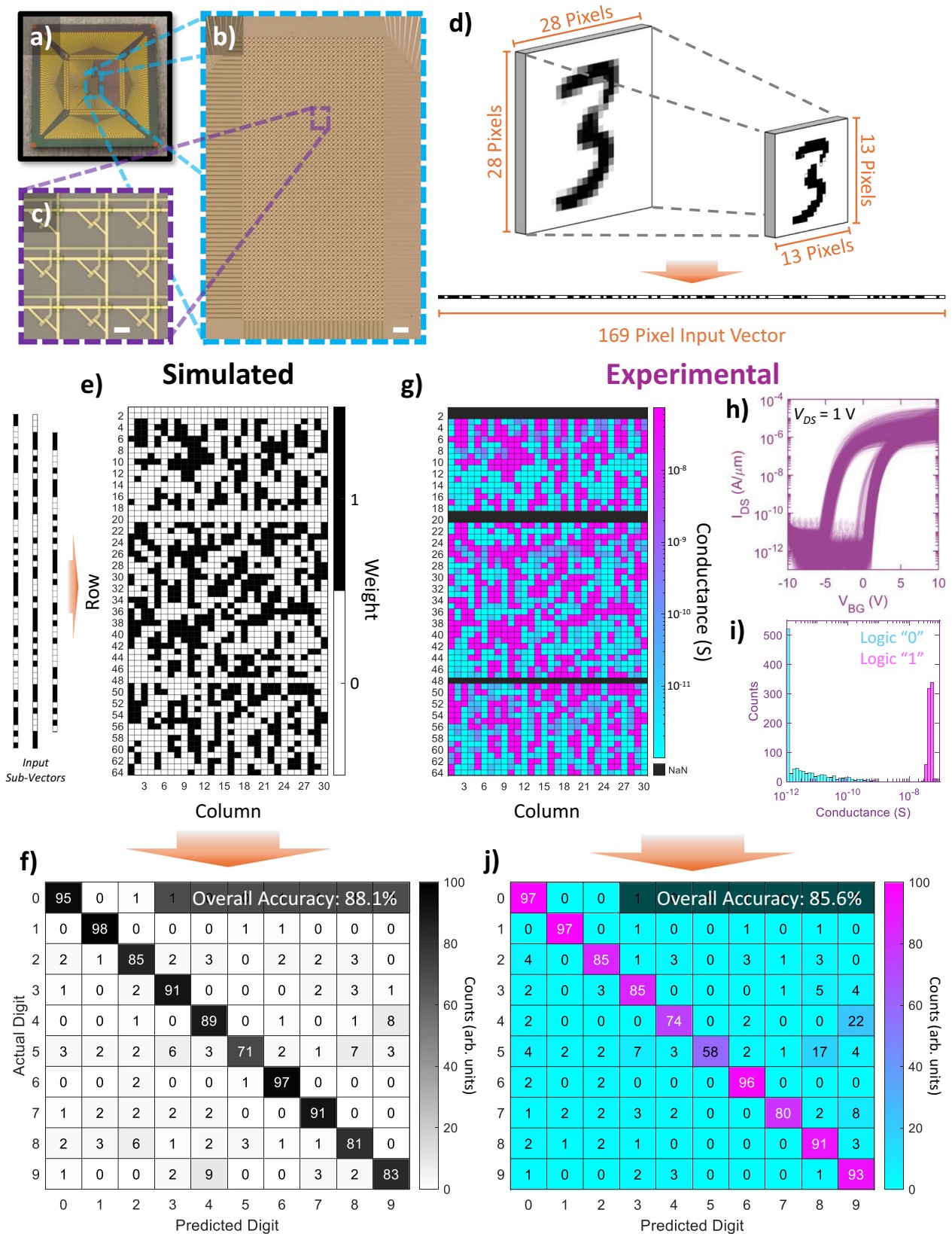

non-idealities, the overall yield of devices across the sub-array remained high at ~92.2%. Hysteresis characteristics and final conductance states (weights) for the working cells are also shown in Fig. 3h, i, respectively, demonstrating good uniformity across the breadth of the array as well as a significant read margin of >10⁴ between differently weighted cells. For the sake of clarity, devices

demonstrating a measured conductance below the noise floor conductance of the characterization system (~$10^{-12}$ S) are marked as being at the noise floor. For this hardware-based inference demonstration, the outputs from three columns of the sub-array correspond to a single digit utilized in the MNIST dataset, e.g., the first three columns in the sub-array correspond to the digit "0", the next three columns

**Fig. 3 | Demonstration of MNIST Handwritten Digit Classification on a 2 kb Memtransistor Crossbar Array.** Optical images of (**a**) a 1.5 × 1.5 cm chip containing a 64 × 32 crossbar array comprising 2048 $MoS_2$ memtransistors, including (**b**) a zoomed-in image showing the full 2 kb array (scale bar of 100 μm) and (**c**) a further zoomed-in image showing nine constituent memtransistors (scale bar of 10 μm). **d** Schematic showing preprocessing performed on training and test (inference) images taken from the Modified National Institute of Standards and Technology (MNIST) database. The original images (28 × 28 pixels) were downscaled to 13 × 13 pixels and binarized to fit into a 64 × 30 sub-array for this demonstration. Downscaled images were then converted to 169 × 1 input vectors and split into three sub-vectors for input to the array. A dataset comprising 10,000 resized/reshaped images were then used for training and weight assignment; for this demonstration, simulated weights were split between logic "0" and "1" and later converted to targeted conductance states for hardware implementation. **e** Heatmap showing the distribution of simulated weights following training. A test dataset of 1000 resized/reshaped MNIST images was fed to the simulated array for verification of network inference/classification. **f** Confusion matrix showing the classification results for the simulated inference check described in (**e**). An overall accuracy of 88.1% was

achieved. **g** Heatmap showing the distribution of conductance states assigned to the hardware array in respect to the simulated weight distribution shown in (**e**), with weights of "1" mapped to a conductance state of ~ 50 nS and weights of "0" mapped to the OFF-state conductance (a few pS). Cells marked NaN either display an open circuit or high gate leakage; the overall yield of devices remained high at ~ 92.2%. **h** Hysteresis characteristics of the 1770 working devices in the 64 × 30 sub-array extracted at $V_{DS} = 1$ V, which show low intrinsic device-to-device variation, memory windows > 5 V, and read margins > $10^4$. **i** Histogram showing the distribution of the final conductance states (weights) represented in (**g**). **j** Confusion matrix showing the classification results for hardware-based inference performed on the memtransistor array shown in (**a**–**c**) as per the conductance state (weight) distribution shown in (**g**). A test dataset comprising 1000 resized/reshaped MNIST images was applied to the drain terminals of the array as voltage inputs (either 0 V or 1 V, depending on the corresponding pixel value); output currents along corresponding columns/nodes were then individually registered and compared to all other outputs to determine the inferred digit for each case. An overall accuracy of 85.6% was registered.

correspond to "1", etc. When an image (binary 169 × 1 vector split into three sub-vectors) is applied as voltage inputs (either 0 V or 1 V depending on the corresponding pixel value) to the drain terminal, the output currents from each column set (node) are individually registered. The summed output currents from each node are then compared to determine the inferred digit. Please note that, due to limitations with the experimental setup, summation of output currents was performed externally for the sake of this demonstration. The results for this classification scheme are shown in Fig. 3j. An overall accuracy of 85.6% was registered for the hardware demonstration; at only 2.5% lower than simulated, a similar simulation-to-hardware gap to previous MNIST classification explorations[11], this indicates that the $MoS_2$-based crossbar arrays developed and demonstrated in this work can successfully implement inference operations with a high degree of similarity to simulations, thus indicating potential as neuromorphic accelerators. Deviations from the ideal case demonstrated in Fig. 3f can be attributed to device-to-device variation, such as that shown in Fig. 1, and the non-ideal yield noted in Fig. 3g; these factors can be optimized in future work through array-level compensation schemes[67–69] and optimization of the fabrication and $MoS_2$ growth processes. A comparison of the current and predicted status of our work with other emerging logic accelerators for neural networks[5–8,10,25,26,32,70–75] is presented in Supplementary Note 18.

While the above MNIST inference demonstration shows the ability of our $MoS_2$-based crossbar arrays to implement complex classification operations in hardware with a high degree of similarity to simulations, thus acting as neuromorphic accelerators, the results still leave much to be desired in terms of overall accuracy. This is fairly in line with expectations; from simulation, we can see that even the ideal case accuracy is limited to 88.1% due to limitations of performing inference across highly compressed and binarized handwritten digits, which is in turn further exacerbated by device-to-device variations and non-ideal device yield once implemented in hardware. Together, these issues contribute to the high degree of confusion shown between certain digits in Fig. 3j, such as "5" and "8", dragging down the overall accuracy. While certainly non-ideal, these complications effectively mirror the complexities of various real-world scenarios, where noisy, low-resolution datasets and uncertainty between classifications are the norm rather than the exception. While many common real-world scenarios can be addressed at the training stage depending on the application, such as rain or snow in surveillance footage, addressing unexpected events that may impact collected data, such as out-of-season weather at odd times of day, may require supervised data labeling and subsequent retraining of the network to obtain classification at reasonable accuracies, tasks that are completely untenable for isolated edge computing applications. Here, our ability to

dynamically modulate pretrained weights, as discussed above and in reference to Fig. 2, provides a distinct advantage for differentiating classes with otherwise high degrees of confusion, opening avenues for automatic verification schemes in decentralized edge applications. An example of such a verification scheme performed on a 64 × 10 subsection of the array presented in Fig. 3 is presented for the cases of two sets of highly-compressed (8 × 8-pixel) MNIST digits, "5" and "8", and "8" and "9", in Fig. 4. Simulation and experimental results for the 8 × 8-pixel MNIST digit classification across all ten digits are presented in Supplementary Note 19. As shown in Fig. 4a, by cumulatively considering pixel intensities across a wide body of images for each digit, features of generally higher or lower importance for each digit can be identified. Taking the difference in cumulative pixel intensity (Δ pixel intensity) between different digits then allows for the features of highest and lowest contrast between them to be identified, as shown in Fig. 4b for digits "5" and "8" and in Fig. 4c for digits "8" and "9". By dynamically potentiating the pre-programmed weights pertaining to features of high contrast and depressing weights for features of low contrast, as enabled through the application of positive and negative $V_{mod}$ to their appropriate gate lines, respectively, the accuracy of inference operations between two specific digits of the MNIST dataset can be enhanced; as this does not require time/energy intensive retraining or reprogramming of weights, this procedure can serve as a rapid automatic verification scheme for differentiating otherwise highly confused classes in resource-limited hardware applications. The results for implementing such a scheme on the hardware-based inference results displayed in Fig. 3 are shown in Fig. 4d–g. Figure 4d, e show confusion matrices between "5" and "8" and "8" and "9", respectively, for the standard hardware-based inference demonstration originally shown in Fig. 3; alternatively, Fig. 4f, g show confusion matrices for both cases when heterosynaptic modulation is dynamically applied to areas of high and low contrast between digits, as identified in Fig. 4b, c, respectively. As can be seen, considering "5" and "8" without weight modulation returns a classification accuracy of only 65%, which increases to 79.5%, a 14.5% increase, with potentiation/depression of respective weights. Similarly, consideration of digits "8" and "9" shows 77% classification accuracy without modulation and 81% classification accuracy with modulation. Together, these results indicate that dynamic modulation of weights, when properly applied through the exploitation of the gate terminal, allows for enhanced classification accuracy among highly confused classes in $MoS_2$-memtransistor-based crossbar arrays. As this does not require time/energy-intensive retraining or reprogramming of weights, this procedure may serve as a rapid automatic verification scheme for differentiating classes in resource-limited hardware applications.

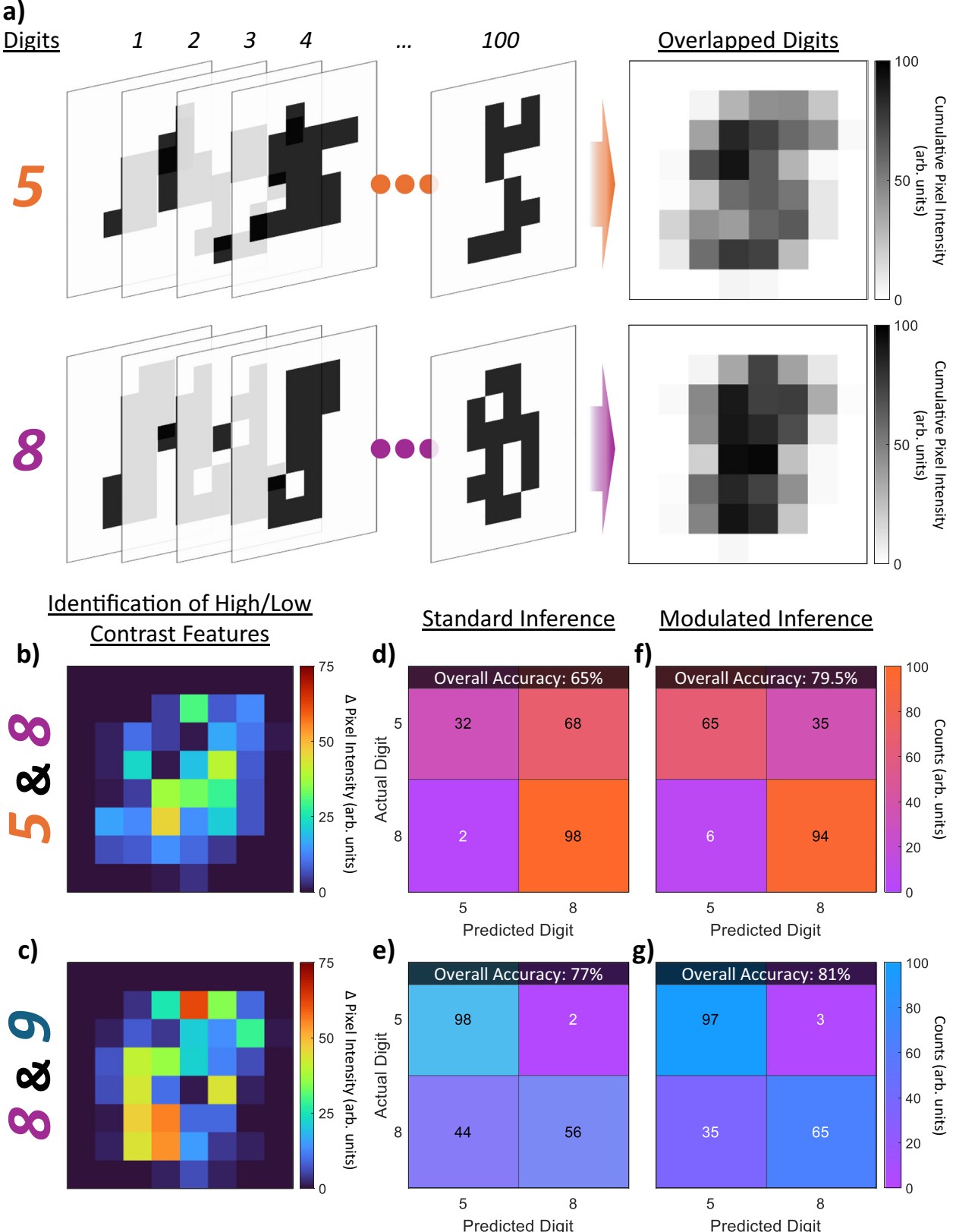

In summary, we have demonstrated crossbar array architectures utilizing monolayer-$MoS_2$-based memtransistors as their base computational unit with information densities and sizes up to 1.94 Mb/cm² (1-bit operation) and 2 kb, respectively. A high yield of >92% with low device-to-device variation was confirmed for each array investigated as part of this work, with constituent devices displaying switching energies as low as ~ 0.2 fJ, read margins as high as $10^5$, and projected retentions exceeding three years. Simple shape identification was performed accurately in hardware, while investigation of MNIST handwritten digit inference found only a 4.5% difference between simulation and experiment, establishing that our crossbar array architectures may function as neuromorphic accelerators for the

**Fig. 4 | Implementation of Heterosynaptic Modulation for Enhanced MNIST Digit Recognition. a** Schematic representation showing how consideration of MNIST digit datasets, in this case downscaled (8 × 8-pixel) and binarized "5's" and "8's", can lead to identification of areas of greater and lower importance to digit recognition, showcased here as areas of greater and lower cumulative pixel intensity (Δ pixel intensity), respectively, when images are overlapped. **b, c** By taking the difference in Δ pixel intensity between two digits of interest, (**b**) "5" and "8" and (**c**) "8" and "9", the features of highest and lowest contrast may be identified.

**d, e** Confusion matrices between (**d**) "5" and "8" and (**e**) "8" and "9" for the hardware-based inference performed on a 64 × 10 sub-section of the array in Fig. 3. **f, g** Confusion matrices between (**f**) "5" and "8" and (**g**) "8" and "9" for cases wherein heterosynaptic modulation is dynamically applied to areas of high and low contrast between digits, as identified in (**b**) and (**c**), respectively. Increases in overall classification accuracy of 14.5% and 4% can be seen for "5" and "8" and for "8" and "9", respectively, compared to the standard case.

implementation of neural network operations. Finally, we also show how the three-terminal nature of MoS₂-based memtransistors as NVM cells may be exploited to reduce confusion between similar outputs in inference tasks (e.g., shape identification and MNIST classification) in a method analogous to modulatory synapses in BNNs, thus opening avenues for automatic verification schemes without costly retraining/reprogramming of weights. We believe that this work furthers the ongoing development of in-memory processors for decentralized edge applications by demonstrating large-scale, hardware-based inference on MoS₂-based NVMs and investigating their potential for implementing non-traditional logic operations through the exploitation of their multi-terminal nature. Future material and fabrication optimization would serve to tighten variation, improve yield, and enhance the long-term non-volatile memory characteristics of individual devices, while inference accuracy could be enhanced at the array-level through dedicated compensation schemes.

## Methods

### Large-area monolayer MoS₂ film growth

The growth of MoS₂ monolayer film on 2″ diameter c-plane sapphire substrates was carried out in a metal-organic chemical vapor deposition (MOCVD) system (https://doi.org/10.60551/znh3-mjl13) equipped with a cold-wall horizontal reactor with an inductively heated graphite susceptor with gas-foil wafer rotation[76]. The molybdenum hexacarbonyl (Mo(CO)₆) (99.99%, Sigma-Aldrich) was used as the metal precursor, while hydrogen sulfide (H₂S) was the chalcogen source with H₂ as the carrier gas. The Mo(CO)₆ powder was maintained inside a stainless-steel bubbler where the temperature and pressure of the bubbler were held at 20 °C and 625 Torr, respectively. The MoS₂ monolayer was grown in a single-step process[77]. Before the growth, the sapphire was ramped up under H₂ to the growth temperature at 975 °C and pre-annealed for 10 min. During the growth, H₂S and Mo(CO)₆ were introduced to the reactor for a designated time to complete MoS₂ monolayer growth in a single step. The molybdenum flow rate was set as $3.4 \times 10^{-3}$ sccm and the chalcogen (H₂S) flow rate was set as 400 sccm while the reactor pressure was maintained at 50 Torr. Then, the MoS₂ monolayer was annealed under H₂ and H₂S ambient for 10 min at 975 °C before cooling down to inhibit the decomposition of the obtained MoS₂ film. Using this condition, the growth of a fully coalesced monolayer MoS₂ was achieved in 10–30 min. across the 2″ sapphire substrate.

### Material characterization

Atomic force microscopy (AFM) images of the as-grown MoS₂ film were taken using a Bruker Dimension Icon system; Scanasyst air probe AFM tips with a nominal tip radius of about 2 nm and spring constant of 0.4 N m − 1 were used for the measurements, and the images were collected using peak-force tapping mode with a peak force of 500 pN and a scan speed of 2 Hz. The Raman and photoluminescence (PL) spectra of the as-grown film were taken using a Witec Alpha-300 Apyron system with a 100 × objective at a 4 mW laser power; the system was enclosed within an N₂-ambient glovebox with ~ 5 ppm of O₂ and H₂O.

### Fabrication of crossbar array architectures

To define the back-gate access lines (word lines), a commercially-purchased substrate (thermally-grown 285 nm SiO₂ on p⁺⁺-Si) was spin-

coated at 4000 RPM for 45 s with a bilayer electron beam (e-beam) photoresist consisting of MMA EL6 and PMMA A3; following application, these resists were baked at 150 °C for 90 s and 180 °C for 90 s, respectively. The bilayer photoresist was then patterned using a Raith EBPG5200 e-beam lithography tool and developed using a 1:1 mixture of 4-methyl-2-pentanone (MIBK) and IPA (60 s) and then rinsed using IPA (45 s). The back-gate electrodes of 3/20 nm Ti/Pt were then deposited using e-beam evaporation in a Temescal FC-2000 Bell Jar Deposition System. Liftoff of the remaining photoresist and excess metal was achieved using acetone; the substrate was then cleaned using 2-propanol (IPA). Two subsequent e-beam lithography and evaporation processes were then conducted to deposit 60 nm Al₂O₃ and 30/60 nm Ti/Au to form insulating crosspoints and conductive bridges, respectively, for the overlapping back-gate access lines in the 1S1T design. An atomic layer deposition (ALD) process was then implemented to grow the back-gate dielectric stack consisting of 15 nm Al₂O₃ ($\varepsilon_{ox} \approx 10$), 3 nm HfO₂ ($\varepsilon_{ox} \approx 25$), and 7 nm Al₂O₃ across the entire substrate, which includes all four designs; all ALD processes were conducted at 200 °C and without breaking vacuum. Access to the individual Pt back-gate contact pads was achieved via a reactive ion etch (RIE) process conducted in a Plasma-Therm Versalock 700. First, etch patterns were defined using e-beam lithography with ZEP 1:1 photoresist. The dielectric stack was then dry etched using a BCl₃ RIE chemistry at 5 °C for 40 s; this process was split into two 18 s etch steps separated by 30 s stabilization steps to minimize heating in the substrate. The photoresist was then removed using Photo Resist Stripper (PRS 3000) heated at 50 °C for one hour and the substrate was cleaned using IPA.

MoS₂ film transfer from the growth substrate to the application substrate was performed using a PMMA-assisted wet transfer process. First, as-grown MoS₂ on a sapphire substrate was spin-coated with PMMA A6 twice and subsequently baked at 150 °C for 120 s to ensure good PMMA/MoS₂ adhesion. The corners of the spin-coated film were then scratched using a razor blade and immersed inside a de-ionized (DI) water bath for 1 hr. Capillary action caused the water to be preferentially drawn into the substrate/MoS₂ interface, owing to the hydrophilic nature of sapphire and the hydrophobic nature of MoS₂ and PMMA, separating the PMMA/MoS₂ stack from the sapphire substrate. The separated film was then fished from the water bath solution using the application substrate. Subsequently, the substrate was baked at 50 °C and 70 °C for 1 hr and 15 min, respectively, to remove moisture and promote film adhesion, thus ensuring a pristine interface, before the PMMA was removed by immersing the sample in acetone overnight, and the substrate was cleaned with a subsequent 30 min IPA bath.

To define the channel regions of the MoS₂ memtransistors found in each NVM cell, the application substrate, with MoS₂ transferred on top, was spin-coated with PMMA A6 (4000 RPM for 45 s) and baked at 180 °C for 90 s. The resist was then exposed using a Raith EBPG5200 e-beam lithography tool and developed using a 1:1 mixture of 4-methyl-2-pentanone (MIBK) and IPA (60 s) and then rinsed using IPA (45 s). The exposed monolayer MoS₂ film was subsequently etched using a sulfur hexafluoride (SF₆) RIE process at 5 °C for 15 s. Next, the sample was rinsed in acetone and IPA to remove the e-beam resist. A subsequent lithography step was conducted to form source/drain electrodes. The substrate was spin-coated at 4000 RPM for 45 s with

methyl methacrylate (MMA) EL6 and PMMA A3; following application, these resists were baked at 150 °C for 90 s and 180 °C for 90 s, respectively. E-beam lithography was again used to pattern the source and drain, and development was again performed using a 1:1 mixture of MIBK/IPA and an IPA rinse for the same times as previously. 40 nm of Ni and 30 nm of Gold (Au) were deposited using e-beam evaporation to form the electrodes. Finally, a lift-off process was performed to remove the excess Ni/Au by immersing the sample in acetone for 1 hr, followed by IPA for another 30 mins to clean the substrate. Two subsequent e-beam lithography, evaporation, and lift-off processes were then conducted to deposit 90 nm $Al_2O_3$ and 40/70 nm Ti/Au to form insulating crosspoints and conductive bridges, respectively, for the overlapping source/drain access lines found in every design. Please note that all source/drain access pads were deposited concurrently with the final conductive bridge step. For samples containing 64×32 (2 kb) arrays, 50/150 nm Ti/Au was instead deposited in the final step to allow for the formation of thick dedicated wire bonding pads on the periphery of the substrates. For all array designs/sizes discussed, the individual memtransistors have channel lengths/widths of 1/1 μm.

### Electrical characterization

Electrical characterization of individual memtransistors was performed in a Lake Shore CRX-VF probestation under atmospheric conditions using a Keysight B1500A parameter analyzer. Ultrafast program/erase and read operations were confirmed using a B1525 SPGU fast pulsing module and a Keysight PZ2100A mainframe equipped with a PZ2120A module, respectively.

### MNIST Digit classification demonstration

**Data preprocessing and weight training.** Grayscale MNIST digit images of dimension 28 × 28 were resized to 8 × 8-pixel images using bicubic interpolation. Next, the 8 × 8 resized images were binarized using a single threshold value. The first 1000 images corresponding to each class in the 60,000-image training set were extracted to create a 10,000-image training set. Furthermore, 100 images corresponding to each class were randomly selected from the 10,000-image testing set to create a 1000-image testing set. No images were repeated in the training or testing set. For the purposes of the proof-of-concept demo, a single-layer neural network was trained on the preprocessed training set using a simulated annealing algorithm. The network structure included 64 inputs (flattened binary 8 × 8 MNIST digit images) fully connected to 10 output nodes that represent each digit from 0 to 9. For this work, the output layer employed a linear activation, and the network prediction was determined by the output node that generated a maximum value during any inference operations. A preliminary weight map was generated prior to the training process by binarizing 50,000 images from the grayscale 8 × 8 training set. Next, the simulated annealing algorithm ran for 20,000 iterations at 7 descending temperatures. Within each iteration, a single weight in the binarized weight map was chosen randomly to flip into the opposite state, either from weight 0 to 1 or vice versa. This new solution was then evaluated based on an objective function defined as the training set inference accuracy minus a penalty for the number of weights used. This penalty aims to remove unnecessary weights of '1' and reward sparsity in weight representation that may reduce crossbar energy consumption and minimize hardware-related non-idealities such as sneak path. If the proposed new solution reduced the value of the objective function compared to the prior iteration, then the new solution was always accepted. If the proposed new solution increased the value of the objective function compared to the prior iteration, then a probabilistic process was used to determine if the new solution would be accepted. The difference between the output of the objective function evaluated for the new and prior solutions was used as input to an exponential function that depends on the predefined temperature. The

exponential function dictates the probability of accepting the new solution. After training, the simulated annealing algorithm reached an accuracy of 88.1% on the 1000-digit testing set.

### Hardware-based Digit Classification

Weights obtained through the software-based approach discussed above were mapped to a 64 × 32 crossbar array through program/erase operations. First, the array was initialized by setting each memtransistor to a low conductance state (LCS) through the application of a large negative gate bias to each gate access line. The trained weights were then converted to real-world conductance values, with 0 being converted to LCSs and 1 being converted to high conductance states (HCSs), by programming each device as dictated by the trained weight matrix. As discussed in the main text above, similar read margins between individual devices allowed for the implementation of an open-loop programming scheme, thus eliminating the time/energy overhead required for a write/verify scheme. For digit classification, 64 × 1 input vectors were fed to the array, with MAC operations being conducted along each column (node); the output for each node was then assessed as the probability of its respective digit being the input. For cases wherein multiple nodes displayed similar outputs (probabilities), a predetermined verification scheme was implemented based on the nodes (digits) in question. Weights were either potentiated (strengthened) or depressed (weakened) depending on whether they corresponded to areas of greater difference or similarity, respectively, between the images through the application of a modulatory gate bias (positive for potentiation, negative for depression); reconducting MAC operations across the modulated weights thus allowed for greater certainty in classification to be obtained for a given input.

## Data availability

Relevant data supporting the key findings of this study are available within the article and the Supplementary Information file. Data on $MoS_2$ samples produced in the 2DCC-MIP facility, including growth recipes and characterization data, are available at https://doi.org/10.26207/2hsj-0n18. All raw data generated during the current study are available from the corresponding authors upon request.

## Code availability

The code used for the neural network models and data processing discussed in this work are available from the corresponding author upon request.

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

## Author contributions

T.F.S. and S.D. conceived the idea and designed the experiments. T.F.S. designed the crossbar architectures, developed the fabrication process flow, fabricated the devices, performed the device characterization and inference experiments, and wrote the manuscript. A.P. developed the network models and simulations. J.M.K. and D.K.T.B. assisted with large-scale inference experiments. T.F.S., A.P., S.T., E.H., R.O., C.H., and S.D. analyzed the data, discussed the results, and agreed on their implications. C.C. and T.M. grew MOCVD-grown $MoS_2$ films under the supervision of J.M.R. All authors contributed to the preparation of the final manuscript.

## Competing interests

The authors declare no competing interests.
