## [Transparent Peer Review file · Nature Communications]

Large-scale crossbar arrays based on three-terminal MoS₂ memtransistors

Corresponding Author: Professor Saptarshi Das

Version 0:

Reviewer comments:

Reviewer #1

(Remarks to the Author)

In this work, Schranghamer et al. present the largest crossbar array of 2,048 charge-trapping-based MoS₂ memory devices, achieving a 92% yield. The crossbar array was utilized to perform recognition of MNIST handwritten digits, benefiting from significant time and energy savings due to the exploitation of the third terminal in the memory device, which helps mitigate neuron confusion during inference tasks. The impact of this work depends on demonstrating the effectiveness of dynamically modulating weights during inference and the associated benefits, which require more supporting evidence. Therefore, this work would require major revisions before being considered for publication.

Comments:

1. Pg 3-4 – Introduction: The concept of using the third terminal in the memtransistor as a modulatory terminal to prevent confusion between similar inputs is not novel and has been demonstrated in previous works. For instance, *Advanced Functional Materials* (2023), 33(45), 2302949 similarly employed a charge-trapping-based memory to shift the threshold voltage of the memtransistor, enabling excitatory and inhibitory modes that facilitate competitive neural networks with adaptive inhibitory synapses to prevent multiple neurons from learning the same feature. The use of a third terminal in that work suggests that dynamic weight modulation during inference, as proposed in this study, may not be necessary to address input similarity. The authors should discuss their motivation in light of this prior work and clarify how their methodology advances the field.
2. Pg 4 – Introduction: The authors state that "while the achieved integration/information densities reported here still lie far short of the >1 Tb/cm² necessary for full-scale emulation of BNNs in hardware" and "some reports have demonstrated integration densities down to ~5 μm² per cell." It would be beneficial to clarify the limiting factors affecting these area densities and explicitly define how the area density is calculated. For example, in Supplementary Information 4, regarding the Au/MoS₂/Au memristor with a 5 μm contact, how is the cell area determined to be 100 μm²?
3. Pg 4 – Energy savings from retraining and programming: Although the authors mentioned this in their introduction, it is not quantitatively determined in the text how much energy is saved using the dynamic modulation methodology. The authors should showcase this and benchmark to the instance without the dynamic modulation to strengthen their manuscript.
4. Pg 5 – Process Information: The authors should discuss the factors contributing to the high device yield achieved in this study. Providing a detailed analysis of fabrication processes, material uniformity, and device variability would strengthen the manuscript.
5. Pg 9 – Write Energies: The claim of achieving the lowest write energy in the order of fJ with a 1 μs write time requires further validation. In practical applications, such low write energies are insignificant if they cannot be reliably read and retained. The authors should perform retention measurements using pulse read times within the same order of magnitude as the write time to verify the non-volatility of the memory at these low energies. Additionally, since lower switching energy results in a smaller switching ratio, the authors should discuss the implications for their dynamic modulation methodology and its limitations in real-world implementations.
6. Pg 10 – Potential Confounding Memory Switching Mechanisms: Since the MoS₂ in this work is CVD-grown, sulfur vacancies are expected. The authors should address why vacancy movement is not considered in their analysis. *ACS Nano* (2021), 15(1), 1764–1774 discusses grain boundary-assisted vacancy migration, even at sub-1V drain voltage. Given that the device is read at 1V, is resistive switching due to vacancy movement a possibility? Additionally, previous studies (*ACS Nano* (2022), 16(9), 14308–14322) highlight the role of vacancy diffusion, drift, and lattice temperature in memtransistor operation. The authors should specifically discuss the interplay between carrier trapping/detrapping and vacancy migration.
7. Pg 9–10 – Potentiation and Depression: While the authors claim that dynamically tuning weights with V_{mod} leads to

significant savings, this approach likely requires substantial additional peripheral circuitry. Determining which pixel values to tune and the degree of tuning would necessitate external memory or a lookup table, increasing energy consumption and footprint. The authors should address these concerns in their discussion.

8. Pg 9–10 – Number of Possible Bits Encoded: The statement “similar outputs by dynamically tuning conductance through gate modulation” raises the question of how many distinct conductance states can be achieved. Traditional resistive memory benefits from multiple conductance states to encode more data. However, this approach appears to encode binary states (0 and 1) and later adjust the reading gate voltage to obtain additional states. The authors should explain how this technique improves upon traditional resistive memories in reducing inference ambiguities and enhancing system efficiency.

9. Pg 16 – Binarized Inputs: The use of binarized inputs appears to be a device limitation rather than a design choice. Moreover, the network's low recognition accuracy (~62%) on the MNIST dataset suggests that this hardware may struggle with more complex datasets, contradicting the authors' claims about edge applications. The authors should explore increasing the number of weight states per memory device to address this concern and improve recognition accuracy. If this is not feasible, a discussion explaining these limitations should be included.

10. Pg 19 – Comparison with Other Hardware Technologies: The authors present a comparative table without discussion. Strengthening this section with a quantitative analysis of their methodology's strengths and weaknesses relative to existing technologies would improve the manuscript. Additionally, emerging non-volatile technologies such as oxide-based memristors and FeFETs should be included in the comparison.

11. Pg 20 – Identification of Similar Features Between “5” and “8”: The authors state that similar features between “5” and “8” make them difficult to distinguish, and that dynamic inference adjustments using the third terminal help enhance certain features. However, as noted in Comment 7, where is the information about which features to enhance stored, and how is it extracted to feed into the crossbar array? The authors should provide an example of this operation and determine the energy and footprint costs to enable a fair comparison with other architectures.

12. Supplementary S8 – Device-to-Device Variation: Figure S8 shows significant device-to-device variation, particularly in threshold voltages between R1C1 and R2C1. The authors should discuss how this variability impacts MNIST classification accuracy and whether it contributes to confusion between similar outputs during inference.

13. Supplementary S9 – Inconsistency in Programming and Erase Voltages: The programming and erase voltages are both listed as -10V, which seems unusual. The authors should clarify whether this is a typo.

Reviewer #2

(Remarks to the Author)

Comments:

This manuscript demonstrates large-scale crossbar arrays based on MoS₂ memtransistors for in-memory computing, achieving high yield (>92%) and low write energy (~0.2 fJ) across arrays of up to 2048 devices. The three-terminal design enables dynamic conductance tuning through gate modulation, aiming to resolve inference ambiguities without costly retraining or reprogramming. This demonstration is interesting and highlights the potential of MoS₂ memtransistors for edge AI applications. I recommend its publication in Nature Communications after addressing the questions listed below.

1. The manuscript reports an information density of 0.15 Mb/cm² and 1.94 Mb/cm² on page 7. It would be beneficial to explicitly provide the calculation method to enhance clarity and reproducibility.

2. Supplementary Information 2 and 3 present two more condensed designs with cell areas of 105 μm²/cell and 55 μm²/cell, respectively. A detailed structural analysis of how to achieve such cell sizes is recommended.

3. Table 2 of the supplementary information does not mention energy efficiency (TOPS/W), a key metric for assessing the practical viability of neural network accelerators. While the table provides detailed comparisons of throughput and computational density, it lacks power consumption data, making it difficult to evaluate the overall efficiency of the proposed MoS₂-memtransistor-based accelerator in relation to other architectures. To strengthen the comparative analysis and provide a more comprehensive assessment of the accelerator's performance, I recommend incorporating energy efficiency data.

4. The manuscript discusses the ability to program MoS₂ memtransistors via charge trapping in the Al₂O₃/HfO₂/Al₂O₃ dielectric stack. However, it does not provide sufficient analysis of the trade-off between retention stability and write disturbance—an important issue in charge-trapping-based memories. Since repeated write operations can introduce charge-trapping asymmetry and retention drift, I recommend that the authors investigate whether extended write cycling affects the stability of stored weights over long-term inference operations.

5. While the manuscript provides an analysis of the device's long-term retention (>3 years), its assessment of cycling endurance is limited and lacks degradation analysis under repeated program/erase cycles. To strengthen the discussion, I recommend incorporating cycling endurance experiments and exploring the underlying degradation mechanisms.

6. This study demonstrates MNIST classification using a MoS₂ memtransistor crossbar array and introduces a heterosynaptic modulation method to enhance classification accuracy for highly confused digit pairs. However, even in simulation, the classification accuracy is only 62.8%, further dropping to 58.3% in hardware implementation. While heterosynaptic modulation improves specific cases (e.g., increasing the classification accuracy of “5&8” by 14.5%), its impact on overall classification performance remains limited. As thus, the benefits of using memtransistors for neuromorphic computing are not convincingly demonstrated. To better highlight the advantages of memtransistor-based computing, it is recommended that the authors first optimize the simulation to achieve at least 90% accuracy in simulation. Only after establishing a high-performance baseline should the hardware implementation be evaluated to clearly demonstrate the

advantages of the memtransistor crossbar array.

7. The manuscript mentions initializing all devices to the OFF state and using a small negative bias to suppress sneak path currents. A more detailed quantitative analysis is needed to evaluate the sneak leakage current across large arrays.

8. The study currently employs binary ON/OFF weight storage, yet MoS₂ memtransistors inherently support multi-level conductance states. Since increasing weight resolution can significantly improve inference accuracy, the authors should discuss whether multi-level programming is feasible within this architecture and how it might impact training convergence and classification performance.

9. Literature such as Sangvan et al., Nature, 2018, 554, 500–504, and Tan et al., ACS Nano, 2024, 18(21), 13652–13661, presents another type of MoS₂ memtransistor that leverages the dynamic modulation of Schottky barrier height to induce resistive switching behavior. What is the advantage of the memtransistor in this work compared to these studies? Discussing these works in the introduction would help establish a more comprehensive theoretical framework and highlight the relationship between existing progress and the present work.

10. The time axis in Fig. S12 is labelled with the unit "t", which may cause confusion as it does not explicitly indicate the standard unit of measurement (e.g., seconds, milliseconds). To ensure clarity and consistency with the rest of the manuscript, I recommend specifying the correct time unit explicitly.

Reviewer #3

(Remarks to the Author)

The authors reported here a large-scale crossbar arrays based on 2048 MoS₂ memtransistors for artificial neuron networks. The MoS₂ memtransistors utilize the charge-trapping in Al₂O₃/HfO₂/Al₂O₃ back gate stacks and work as NVM. The arrays demonstrate the ability to resolve inference ambiguities through gate modulation without the need for costly retraining or reprogramming, and image classification of handwritten digits from MNIST database. However, the benefit of these MoS₂ memtransistor arrays based artificial neuron networks is not clear, and how it compares with two terminal memristors, which already simultaneously deliver high energy efficiency, versatility to support diverse models and software-comparable accuracy at chip level. Some key performance of the MoS₂ memtransistors NVM needs to be shown.

-The authors claimed the device could achieve retention exceeding 3 years, but there isn't any retention data to support this.

-Please exhibit endurance of the devices.

-Please explain how the switching energies of ~0.2 fJ is calculated.

-The recognition accuracy for MNIST classification is very low compared to literatures. This should be improved.

-It seems that the array based on 64×32 MoS₂ memtransistors is fabricated, but isn't tested at array level. Please indicate if the array could be tested at array level. If yes, please explain the test in details and show the related experimental data.

Version 1:

Reviewer comments:

Reviewer #1

(Remarks to the Author)

The authors have addressed most of the concerns, except for the following point. Please include this additional comment in the review:

"Following up on Question 12, the authors mentioned they may not have fully understood the question.

To clarify: the question pertains to device-to-device variation. In Figure 1, the threshold voltage exhibits a standard deviation of 0.23 V, and the subthreshold slope shows σ of 55.8 mV/decade. Could you elaborate on how such variations might affect inference accuracy, particularly in terms of potential confusion between similar output states or classifications? A detailed discussion on this aspect would be appreciated."

Reviewer #2

(Remarks to the Author)

My early concerns have been addressed by the authors. I thus recommend it for publication.

Reviewer #3

(Remarks to the Author)

The authors have addressed most of the questions in the revised manuscript and supporting information. However, the retention results of MoS₂ memtransistor show the drain current decays all the time until it reaches the off state. The authors should not claim that the device could achieve retention exceeding 3 years. Moreover, the MoS₂ memtransistor shows ~10 times degradation in 500 cycles. How about the endurance after more cycles, such as 1E9 cycles, which are typical operation for NVMs endurance. From the retention and endurance tests, MoS₂ memtransistor in this work could not be considered as a NVM cell. Both retention and endurance will influence the training and inference of ANN. The authors should consider this impact in the work.

Reviewer #1 (Remarks to the Author):

In this work, Schranghamer et al. present the largest crossbar array of 2,048 charge-trapping-based MoS₂ memory devices, achieving a 92% yield. The crossbar array was utilized to perform recognition of MNIST handwritten digits, benefiting from significant time and energy savings due to the exploitation of the third terminal in the memory device, which helps mitigate neuron confusion during inference tasks. The impact of this work depends on demonstrating the effectiveness of dynamically modulating weights during inference and the associated benefits, which require more supporting evidence. Therefore, this work would require major revisions before being considered for publication.

We thank the reviewer for their feedback and have taken steps to address their concerns, as discussed below.

Comments:

1. Pg 3-4 – Introduction: The concept of using the third terminal in the memtransistor as a modulatory terminal to prevent confusion between similar inputs is not novel and has been demonstrated in previous works. For instance, Advanced Functional Materials (2023), 33(45), 2302949 similarly employed a charge-trapping-based memory to shift the threshold voltage of the memtransistor, enabling excitatory and inhibitory modes that facilitate competitive neural networks with adaptive inhibitory synapses to prevent multiple neurons from learning the same feature. The use of a third terminal in that work suggests that dynamic weight modulation during inference, as proposed in this study, may not be necessary to address input similarity. The authors should discuss their motivation in light of this prior work and clarify how their methodology advances the field.

We agree with the reviewer that the use of the third (gate) terminal in memtransistor structures to prevent/mitigate confusion has been discussed and explored experimentally

in previous works; it was not our intention to insinuate otherwise and we apologize for any confusion. Furthermore, we also thank the reviewer for bringing the manuscript “N-P Reconfigurable Dual-Mode Memtransistors for Compact Bio-Inspired Feature Extractor with Inhibitory-Excitatory Spiking Capability” (*Advanced Functional Materials* (2023), 33(45), 2302949) to our attention¹⁰. We found the demonstration of dual excitatory-inhibitory synapses for hardware-based feature extraction to be insightful both on its own merit and as an analogous bio-inspired approach for implementing high-level neural mechanisms on-chip with greater computational efficiency. However, we believe that several key differences distinguish our work from this one. Most notably, the work in question employs the gate terminal primarily as a programming input during weight assignment, shifting the threshold and enabling fixed excitatory or inhibitory behavior for later inference operations. In contrast, our work demonstrates how three-terminal memtransistors have the potential to use the gate terminal dynamically during inference to modulate weights, and therefore outputs, without changing the underlying conductance state. As we demonstrate in the manuscript, dynamic gate modulation allows for weight tuning and task-specific separation of high confusion classes without requiring retraining of the network or reprogramming of individual weights. Such energy-/time-intensive operations are difficult to realize in edge applications with strict energy/latency constraints.

Another key difference is the scale of the experimental demonstration; as mentioned in the main manuscript and supported through our benchmarking table of similar 2D-material-based technologies, our 64×32 memtransistor array is, to the best of our knowledge, the largest array-level experimental demonstration reported thus far in literature. While small-scale experimental arrays¹⁻⁹ and large-scale array simulations based on individual devices¹⁰ are both useful for exploratory efforts, comprehensive experimental investigation at scales relevant to practical applications is required to validate the use of 2D-material-based crossbar arrays in neuromorphic accelerators. Another point of distinction is how programming is performed. As the reviewer mentions in one of their comments below, many extant 2D-material-based crossbar demonstrations^{2,11-14} program (write) devices through application of a drain bias pulse to prompt defect migration across the semiconducting

channel, thereby changing the threshold (conductance) of the device; depending on the biases required, with lower biases always preferable for greater energy efficiency, this can lead to destructive read operations similar to those that affect two-terminal memristor technologies^{11,12}. Additionally, defect engineering of the channel, e.g., through low-energy Ar plasma treatment¹⁵, may be required to provide a sufficiently high concentration of defects for appreciable memory effects to be seen at low voltages, thus complicating the fabrication process. Conversely, for our demonstration, programming is performed through the application of bias pulses to the gate terminal to promote charge-trapping in the dielectric stack, thereby shifting the threshold by screening the electric field across the channel in a manner analogous to traditional FLASH memories¹⁶⁻¹⁸. As the drain terminal is decoupled from the programming mechanism, this minimizes the risk of destructive read events. Furthermore, unlike defect-migration-based memories, charge-trapping memories do not require any modification (i.e., defect engineering) of the channel material, relying solely on the as-fabricated dielectric stack for memory operation.

We have adjusted the introduction of the revised manuscript as follows to better highlight the unique contributions of this work compared to previous investigations of 2D-material-based memories:

While resistive memories^{8,19,20}, charge-trapping (FLASH) memories²¹⁻²³, and ferroelectric memories²⁴⁻²⁶, among others, have already seen serious investigation as NVM cell technologies, one group of emerging alternatives that currently show promise for in-memory/sensor computing through back-end-of-line (BEOL) integration with standard complementary metal-oxide-semiconductor (CMOS) technology is two-dimensional (2D) materials. Monolayer MoS₂, a semiconductor from the transition metal dichalcogenide (TMDC) family, has attracted particular attention for use in three-terminal memtransistor devices for NVM applications. These devices exploit monolayer MoS₂'s sensitivity to charge variation stemming from its high surface-area-to-volume ratio to realize distinct, non-volatile conductance levels (memory states) using charge-trapping gate stacks¹⁶⁻¹⁸ or

metallic floating gate architectures^{1,27,28} akin to traditional Si-based FLASH memories. Notably, the presence of a third (gate) terminal in the memtransistor architecture sets it apart from the two-terminal NVMs commonly used in crossbar architectures by allowing for the application of a separate gate bias during crossbar operation, screening the electric field across the device channel and effectively tuning the conductance state (weight) of the device. Extant studies of three-terminal MoS₂-based memtransistors primarily employ the gate terminal as a programming input during weight assignment^{1,10,27,28}, shifting the threshold and enabling fixed excitatory or inhibitory behavior for later inference operations. However, from a biological perspective, this capability is highly reminiscent of the activities performed by modulatory neurons (interneurons) in heterosynaptic plasticity²⁹⁻³¹. In this process, the stimulation of one neuron causes a change in the strength (weight) of the connections between other, inactivated, neurons. Study of repeated heterosynaptic modulation events in biological neural networks has found that this can lead to persistent changes in synaptic connections³², thus contributing to the formation of long-term memories³³. Developing methods of implementing heterosynaptic plasticity in hardware has therefore been deemed an important goal for realizing next-generation neuromorphic systems with greater energy efficiency³⁰. For crossbar architectures, this offers a method of tuning prewritten conductance states by applying different gate biases and could potentially be used to exaggerate differences between otherwise similar classes, enhancing the separability of highly confused outputs without the need for resource-intensive data processing.

In this article, we present crossbar array architectures utilizing up to 2,048 memtransistors based on monolayer MoS₂ integrated with a charge-trapping memory stack as NVM cells. We achieved a high yield of >92% across multiple crossbar array demonstrations, with constituent devices displaying write energies as low as ~0.2 fJ while retaining read-margins (ratio between programmed and erased states) as high as 10⁵ and projected long-term non-volatile retentions exceeding 3 years. We also investigated the influence of cell area on device- and array-level operations, ultimately demonstrating equivalent performance at information densities of up to 1.94 Mb/cm² (assuming 1-bit operation). We then demonstrate how our architecture can be partitioned to accurately achieve simple inference operations

(i.e., shape detection) in hardware. Uniquely, we also demonstrate how the three-terminal nature of MoS₂-based memtransistors can be exploited to dynamically tune conductance states during inference to modulate weights, and therefore outputs, without changing the underlying conductance state in a manner similar to modulatory synapses; notably, this process allows for weight tuning and task-specific separation of high confusion objects without requiring retraining of the network or reprogramming of individual weights, energy-/time-intensive operations that are difficult to realize in edge applications with strict energy/latency constraints. Finally, we experimentally demonstrate image classification of handwritten digits from the Modified National Institute of Standards and Technology (MNIST) database³⁴ using a 64×32 (2 kb) MoS₂ memtransistor crossbar array. To the best of our knowledge, this represents the largest array-level demonstration using 2D materials reported to date.

2. Pg 4 – Introduction: The authors state that "while the achieved integration/information densities reported here still lie far short of the >1 Tb/cm² necessary for full-scale emulation of BNNs in hardware" and "some reports have demonstrated integration densities down to ~5 μm² per cell." It would be beneficial to clarify the limiting factors affecting these area densities and explicitly define how the area density is calculated. For example, in Supplementary Information 4, regarding the Au/MoS₂/Au memristor with a 5 μm contact, how is the cell area determined to be 100 μm²?

We agree with the reviewer that the term integration/information density can be better defined. For the purposes of this work, integration/information density was defined as the amount of storable information (i.e., weights) in bits per cm²; for assessing the information density of our work, 1-bit (two-state) operation was assumed in all cases. Conversely, cell area was defined as the area (Length × Width) of the repeating unit for each crossbar design benchmarked in literature and demonstrated experimentally as part of our work. Also, please note that the statement "some reports have demonstrated integration densities down

to $\sim 5 \mu\text{m}^2$ per cell," should refer to cell area, not integration density; we apologize for the confusion and have corrected similarly erroneous statements in the revised manuscript.

We have adjusted the following statement to better define integration/information density in the revised manuscript:

This allows us to achieve an information density, defined as the amount of storable information (i.e., weights) in bits per cm^2 , of $\sim 0.15 \text{ Mb}/\text{cm}^2$. Note that, in this work, 1-bit (two-state) operation is assumed in all cases for the purposes of calculating information density.

We have also added the following statement to better define cell area in the revised manuscript:

Here, cell area is defined as the area (Length \times Width) of the repeating unit for a given crossbar design.

All erroneous statements referencing integration density instead of cell area have been corrected.

3. Pg 4 – Energy savings from retraining and programming: Although the authors mentioned this in their introduction, it is not quantitatively determined in the text how much energy is saved using the dynamic modulation methodology. The authors should showcase this and benchmark to the instance without the dynamic modulation to strengthen their manuscript.

We agree with the reviewer that a more in-depth discussion of energy costs would only strengthen our assertion that dynamic modulation of conductance states saves energy compared to retraining/reprogramming of the array. As discussed in the main manuscript, adapting synaptic strength in our MoS_2 -memtransistor-based architecture is achieved through small gate voltage adjustments (V_{mod}), without altering the stored binary

conductance state. Per standard FET operation, this dynamic modulation requires charging the oxide (back-gate) capacitance (C_{ox}), which is ~ 3.8 fF per device. The energy cost per modulation can then be approximated using the equation $E_{mod} = \frac{1}{2} \cdot C_{ox} \cdot V_{mod}^2$. For the demonstrations used in this work, modulatory voltages between ± 2 V are used, resulting in an additional energy expenditure of < 8 fJ per device. In contrast, standard weight retraining approaches (digital or analog) require either reprogramming memory cells or propagating weight updates through the network. For resistive memories, iterative write-verify schemes consume the majority of a crossbar array's energy budget³⁵, and can consume upwards of 100 pJ per device³⁶ depending on the type of memory, number of bits, etc. Additionally, retraining often increases latency and may not be feasible in constrained edge environments. This disparity supports the use of three-terminal modulation as an efficient alternative for fine-tuning network behavior with minimal overhead.

We have added the following discussion regarding the energy expenditure of the dynamic modulation approach to the revised manuscript:

It should also be noted that this dynamic modulation approach has significant implications for energy-efficient computing. Per standard FET operation, this dynamic modulation requires charging the oxide (back-gate) capacitance (C_{ox}), which is ~ 3.8 fF per device. The energy cost per modulation can then be approximated using the equation $E_{mod} = \frac{1}{2} \cdot C_{ox} \cdot V_{mod}^2$. For the demonstrations used in this work, modulatory voltages between ± 2 V are used, resulting in an additional energy expenditure of < 8 fJ per device. In contrast, standard weight retraining approaches (digital or analog) require either reprogramming memory cells or propagating weight updates through the network. For resistive memories, iterative write-verify schemes consume the majority of a crossbar array's energy budget³⁵, and can consume upwards of 100 pJ per device³⁶ depending on the type of memory, number of bits, etc. Additionally, retraining often increases latency and may not be feasible in constrained edge environments. This disparity supports the use of three-terminal modulation as an efficient alternative for fine-tuning network behavior with minimal overhead.

4. Pg 5 – Process Information: The authors should discuss the factors contributing to the high device yield achieved in this study. Providing a detailed analysis of fabrication processes, material uniformity, and device variability would strengthen the manuscript.

We are glad the reviewer found the high device yield of the arrays reported in this work notable enough to warrant further details. As a detailed Methods section and numerous heatmaps/histograms correlating various device parameters of note across the arrays investigated are already included in the manuscript, we feel that further details on the fabrications processes and device variability, unless specified by the reviewer, would contribute little to further discussions already present. However, we do agree with the reviewer that further discussion on material uniformity is warranted due to the large size of the arrays investigated; even with a relatively small individual cell area ($676 \mu\text{m}^2$) compared to other experimentally demonstrated 2D-material-based arrays^{1,2,4,8}, the 64×32 array discussed in this work still covers an area $> 1.38 \times 10^6 \mu\text{m}^2$ (0.0138 cm^2). Ensuring uniformity of the monolayer MoS_2 film across a large area is therefore of high importance to the overall performance of the final arrays.

We have added the following discussion to address the uniformity of the monolayer MoS_2 film utilized in this work to the supplemental material of the revised manuscript:

Raman spectroscopy was used to assess the quality and uniformity of the MoS_2 film used for fabrication of the large-scale array. These results are presented in **Supplementary Figure 17** (included here as **Figure R1**). The Raman spectra was taken across nine points corresponding to the corners, sides, and center of the array with a 532 nm laser. The mean E_{2g}^1 (in-plane) and A_{1g} (out-of-plane) peak locations were found to be 385.26 cm^{-1} and 403.15 cm^{-1} , respectively, with a mean peak separation of 17.89 cm^{-1} . The standard

Figure R1. Raman Spectroscopy Analysis of 64×32 Crossbar Array. The Raman spectra was taken across nine points corresponding to the corners, sides, and center of the array with a 532 nm laser. The mean E_{2g}^1 (in-plane) and A_{1g} (out-of-plane) peak locations were found to be 385.26 cm^{-1} and 403.15 cm^{-1} , respectively, with a mean peak separation of 17.89 cm^{-1} .

deviations for each of these parameters was comparatively low (0.30 cm^{-1} , 0.30 cm^{-1} , and 0.42 cm^{-1} , respectively), indicating good spatial uniformity of the MoS_2 film.

5. Pg 9 – Write Energies: The claim of achieving the lowest write energy in the order of fJ with a $1\text{ }\mu\text{s}$ write time requires further validation. In practical applications, such low write energies are insignificant if they cannot be reliably read and retained. The authors should perform retention measurements using pulse read times within the same order of magnitude as the write time to verify the non-volatility of the memory at these low energies. Additionally, since lower switching energy results in a smaller switching ratio, the authors should discuss the implications for their dynamic modulation methodology and its limitations in real-world implementations.

The reviewer makes an excellent point regarding the insignificance of low write energies if the corresponding conductance states (weights) cannot be reliably read or retained. We have repeated the $1\text{ }\mu\text{s}$ write pulse experiment outlined in the original manuscript on a representative MoS_2 memtransistor and subsequently monitored the retention over 40 ms with ten-thousand $4\text{ }\mu\text{s}$ read pulses. These results are outlined below and confirm that the

Figure R2. 1 μ s Program/Erase Pulse Demonstration. Transfer characteristics, i.e., drain-to-source current (I_{DS}) versus back-gate voltage (V_{BG}), taken at $V_{DS} = 1$ V of a representative memtransistor before and after application of a 1 μ s programming pulse (-10 V). As can be clearly seen, the pulse shifts the V_{th} of the device, indicating a shift to a different distinct conductance state (weight). This establishes that 1 μ s pulse times can program/erase the constituent NVM cells of the crossbar arrays developed and investigated in this work, thus indicating that the arrays may be operated at high speeds (frequencies) than utilized in this work.

charge-trapping memory effect discussed in the main text remains non-volatile at low write energies. The reviewer also makes a good point regarding how a lower switching energy will naturally result in a smaller switching ratio between states. As we show in the results below, a switching ratio $>10^2$ was maintained for μ s-scale write-read testing; this ratio remains comparable to other developing memristor technologies and is more than sufficient for the implementation of inference operations in hardware^{1,19}.

We have adjusted our minimum pulse timing statement as follows in the revised manuscript:

Figure R3. Memory Testing across 100×10 Crossbar Array. a-c) Hysteresis loops for devices located at the top (R1C1), middle (R50C5), and bottom (R100C10) of a representative 100×10 array. For each device, multiple hysteresis loops were taken by sweeping the back-gate voltage between +/- 2 V, +/- 3 V, +/- 4 V, +/- 5 V, +/- 6 V, +/- 7 V, +/- 8 V, +/- 9 V, and +/- 10 V so as to determine the presence/size of the memory window for different program/erase voltages; a sizable memory window of ~10 V can be noted for +/- 10 V sweeps irrespective of array position, indicating that array size has minimal effect on program/erase capabilities. d-f) Retention tests for the top, middle, and bottom devices shown in (a-c), respectively. Devices were subjected to a -10 V programming pulse (putting them in the ON-state) and a +10 V erasing pulse (putting them in the OFF-state) and read at a gate voltage of 0 V and drain voltage of 1 V for ~600 seconds (10 minutes) to observe the decay in their ON/OFF ratio over time. This test was conducted three times to observe retention/programming consistency. The timing of each program/erase bias pulse was 100 ms. g) To analyze the long-term retention and uniformity of these devices, a simple power law fit was extracted for the median retention curve of each device and plotted over 10⁶ seconds (~11.6 days). The final fitted ON-current values are all >100 pA; if the OFF-current for each device remains constant at ~2 pA, the ON/OFF ratio should remain > 100 for over a week, which is more than sufficient for edge computing applications.

The terms “base” and “peak” included in the assessment of our work refer to the pulse time (write time), with base referring to our typical pulse time of 100 ms (used for all

Figure R4. Long Term Memory Testing. To verify the accuracy of the fits shown in **Figure R3**, the ON-state and OFF-state of a representative device/cell were read over periods of ~ 10 minutes (left column) and ~ 1 hour (middle column) and simple power law fits were calculated. These fits were then plotted (right column) over 10^8 seconds (~ 3.5 years), during which time they remained above the designated OFF-state current (~ 2 pA); notably, the 10 minute and 1-hour fits were in close agreement throughout the entire time span, indicating that the 10 minutes fit discussed above are relatively accurate. Additionally, no degradation in the OFF-state was noted even for the longer retention tests, indicating that the change in ON/OFF ratio over time will predominantly depend on the change in ON-state conductance.

demonstrations unless otherwise states) and peak referring to our minimum confirmed pulse time of $1 \mu\text{s}$ (see **Supplemental Information 7**, included here as **Figure R2** for reference). While several other array-level works have also utilized ultrafast write times, our NVM capabilities being controlled by the gate means that the write current of our devices is limited to the gate leakage current, which remained in the region of several tens of pA even at the largest gate biases applied (~ 10 V), thus allowing for severely reduced energy expenditure. Note that, due to experimental setup limitations, the minimum reliable read time was constrained to $4 \mu\text{s}$.

We have added the following statement on the investigation/discussion of μs -scale write-read testing to the revised manuscript:

The memory performance of devices in this array was also assessed separately from the smaller arrays detailed above to accurately determine what effects, if any, scaling has on our program/erase capabilities, with the results being presented in **Supplemental Figure 12-13** (included here as **Figure R3** and **Figure R4**, respectively, for reference). A corresponding

Figure R5. Minimum Write/Read Stability Testing. To verify memory stability even at the minimum confirmed write/read times of 1 μs and 4 μs , respectively, the ON-state (blue) and OFF-state (orange) of a representative device/cell were read over a period of 40 ms using ten-thousand 4 μs read pulses. Programming/erasing were each achieved using a single ± 10 V write pulse, respectively, with a 1 μs pulse time. A read margin $>10^2$ is maintained throughout the entire timespan, indicating that the charge-trapping memory effect remains non-volatile even at low write energies. Note that the high frequency read setup used for this experiment possesses a noise floor of $\sim 10^{-10}$ A; as a result, the actual OFF-state current may be closer to that presented in **Figure R3-4** with a correspondingly larger read margin.

investigation of minimum write/read time stability for the same array is presented in **Supplemental Information 14** (included here as **Figure R5** for reference).

6. Pg 10 – Potential Confounding Memory Switching Mechanisms: Since the MoS_2 in this work is CVD-grown, sulfur vacancies are expected. The authors should address why vacancy movement is not considered in their analysis. *ACS Nano* (2021), 15(1), 1764–1774 discusses grain boundary-assisted vacancy migration, even at sub-1V drain voltage. Given that the device is read at 1V, is resistive switching due to vacancy movement a possibility? Additionally, previous studies (*ACS Nano* (2022), 16(9), 14308–14322) highlight the role of vacancy diffusion, drift, and lattice temperature in memtransistor operation. The authors should specifically discuss the interplay between carrier trapping/detrapping and vacancy migration.

The reviewer posits a good question regarding potential contributions from defect (sulfur vacancy) migration in the observed resistive switching of our memtransistors based on CVD-grown MoS₂. We agree that numerous extant 2D-material-based crossbar demonstrations^{2,11-14} program (write) devices through application of a drain bias pulse to prompt defect migration across the semiconducting channel, thereby changing the threshold (conductance) of the device; depending on the biases required, with lower biases always preferable for greater energy efficiency, this can lead to destructive read operations similar to those that affect two-terminal memristor technologies^{11,12}. Additionally, defect engineering of the channel, e.g., through low-energy Ar plasma treatment¹⁵, may be required to provide a sufficiently high concentration of defects for appreciable memory effects to be seen at low voltages, thus complicating the fabrication process. Conversely, for our demonstration, programming is performed through the application of bias pulses to the gate terminal to promote charge-trapping in the dielectric stack, thereby shifting the threshold by screening the electric field across the channel in a manner analogous to traditional FLASH memories¹⁶⁻¹⁸. As the drain terminal is decoupled from the programming mechanism, this minimizes the risk of destructive read events. Furthermore, unlike defect-migration-based memories, charge-trapping memories do not require any preprocessing (i.e., defect engineering) of the channel material, relying solely on the as-fabricated dielectric stack for memory operation.

We have adjusted the following statement on charge trapping in the revised manuscript to better differentiate the mechanisms utilized in this work from vacancy migration:

The memory capabilities/operations of the Al₂O₃/HfO₂/Al₂O₃ gate dielectric stack incorporated into each MoS₂ memtransistor (NVM cell) can be easily understood through the band diagrams shown in **Supplementary Information 5** (included here as **Figure R6** for reference). As has been studied elsewhere¹⁶⁻¹⁸, the NVM capabilities of each device/cell are governed by the trapping/detrapping of charge carriers in the middle HfO₂ charge-trapping

Figure R6. Band diagram of $\text{Al}_2\text{O}_3/\text{HfO}_2/\text{Al}_2\text{O}_3$ gate dielectric stack. The $\text{Al}_2\text{O}_3/\text{HfO}_2/\text{Al}_2\text{O}_3$ gate dielectric stack enables non-volatile memory in 2D MoS_2 memtransistors by allowing the trapping/detrapping of charge carriers in the HfO_2 (charge-trapping) layer when bias pulses of sufficient magnitude are applied to the back-gate. The polarity of the pulse determines which charge carriers are trapped/detrapped, with holes (electrons) being trapped when negative (positive) pulses are applied, and vice versa. These trapped charges screen the electric field across the MoS_2 channel, changing the conductance of the device and allowing for the realization of distinct conductance (memory) states.

layer when bias pulses of sufficient magnitude are applied to the back-gate in program/erase (write) operations. These trapped charges screen the electric field across the MoS_2 channel, shifting the threshold voltage (V_{th}) of the device and allowing for the realization of distinct conductance (memory) states at a given read voltage (V_{read}). The polarity of the applied pulse determines which charge carriers are trapped/detrapped, with holes (electrons) being trapped when negative (positive) pulses are applied and vice versa, while the pulse time and magnitude determine the change in carrier concentration in the charge-trapping layer (i.e., the size of the V_{th} shift, or memory window). For the purposes of this work, the memory window (ΔV_{th}) is defined as the difference in V_{th} , taken at a constant current level, between programmed/erased states. Note that many extant 2D-material-based crossbar demonstrations^{2,11-14} program (write) devices through application of a drain bias pulse to

prompt defect migration across the semiconducting channel, thereby changing the threshold (conductance) of the device; depending on the biases required, with lower biases always preferable for greater energy efficiency, this can lead to destructive read operations similar to those that affect two-terminal memristor technologies^{11,12}. Additionally, defect engineering of the channel, e.g., through low-energy Ar plasma treatment¹⁵, may be required to provide a sufficiently high concentration of defects for appreciable memory effects to be seen at low voltages, thus complicating the fabrication process. Conversely, for our demonstration, programming is performed through the application of bias pulses to the gate terminal to promote charge-trapping in the dielectric stack, thereby shifting the threshold by screening the electric field across the channel in a manner analogous to traditional FLASH memories¹⁶⁻¹⁸. As the drain terminal is decoupled from the programming mechanism, this minimizes the risk of destructive read events. Furthermore, unlike defect-migration-based memories, charge-trapping memories do not require any preprocessing (i.e., defect engineering) of the channel material, relying solely on the as-fabricated dielectric stack for memory operation.

7. Pg 9–10 – Potentiation and Depression: While the authors claim that dynamically tuning weights with V_{mod} leads to significant savings, this approach likely requires substantial additional peripheral circuitry. Determining which pixel values to tune and the degree of tuning would necessitate external memory or a lookup table, increasing energy consumption and footprint. The authors should address these concerns in their discussion.

We agree with the reviewer regarding the necessity of a peripheral lookup table or memory unit for implementation of the dynamic weight modulation scheme presented in the main manuscript. However, while such peripheral circuitry would undoubtedly contribute to the overall energy consumption and footprint of the proposed neuromorphic accelerator, we must clarify that this scheme is not intended to be applied uniformly across the array or for all inputs. As we discuss in the main text, the overall role envisioned for this hardware-based

Figure R7. Demonstration of In-State Conductance/Weight Modulation. I_{DS} monitored over time for a representative memtransistor programmed into the ON-state through the application of a 100 ms, -10 V back-gate bias pulse and subsequently read at different modulatory back-gate biases (V_{mod}) for a constant $V_{DS} = 1$ V. Note that, despite the device being programmed to a set conductance state through the aforementioned programming event, varying V_{mod} allows for the effective realization of multiple in-state current/conductance levels, thus presenting an avenue for the dynamic potentiation/depression of weights in crossbar arrays through the application of a positive/negative V_{mod} to the respective gate lines. Also note that, for the case of $V_{mod} = 0$ V, the same current/conductance level is retained when V_{mod} switches between values, indicating that this modulatory process does not affect the programmed device state.

computation scheme would be supplemental to standard inference operations, e.g., to differentiate uncertain classes under unexpected (untrained-for) phenomena such as environmental noise, weight drift, or a high degree of class overlap, and thus would only be used on a case-by-case basis when higher accuracy is absolutely required. This selective usage model ensures that the energy and area overhead associated with modulation control is minimal in practice.

For the implementations discussed in the main manuscript, V_{mod} is only applied at the row level, and the modulation levels themselves are low-resolution (3-bit), as shown in **Supplemental Information 8** (included here as **Figure R7** for reference); as a result, an

integrated realization of this technology could be achieved with only a small number of shared DACs or static control registers. From a system perspective, this approach parallels "attention-based" or "event-driven" processing models in neuromorphic hardware, where additional computational resources are engaged only for uncertain or high-priority cases^{37,38}. In all other cases, inference would be conducted on the trained array in a manner like that described for standard MNIST digit classification. For example, in monitoring applications, only the subset of outputs representing critical states (e.g., hazards, anomalies) may be dynamically modulated, while the remainder operate in binary inference mode. This allows the hardware to retain the energy advantages of binary weights while preserving adaptability for edge-case performance tuning.

Ultimately, the dynamic weight modulation scheme presented in this work may be considered as a unique parameter for tuning the performance of crossbar architectures based on three-terminal memtransistors, similar to how parameters such as the number of weights and percent of array utilized are tuned for energy/accuracy tradeoffs in crossbar architectures based on other emerging NVM technologies³⁹⁻⁴⁵.

8. Pg 9–10 – Number of Possible Bits Encoded: The statement “similar outputs by dynamically tuning conductance through gate modulation” raises the question of how many distinct conductance states can be achieved. Traditional resistive memory benefits from multiple conductance states to encode more data. However, this approach appears to encode binary states (0 and 1) and later adjust the reading gate voltage to obtain additional states. The authors should explain how this technique improves upon traditional resistive memories in reducing inference ambiguities and enhancing system efficiency.

The reviewer raises a good point regarding analog versus binary operation of MoS₂ memtransistors. We have already demonstrated analog memory based on our MoS₂ memtransistor approach in previous works¹⁶⁻¹⁸. However, arrays of analog memories require time-/energy-intensive write-verify schemes for accurate weight assignment³⁵, limiting the

applicability of large-scale crossbar arrays for resource-constrained edge implementations such as that targeted in this work. Additionally, incorporating a greater number of conductance states (weights) inherently reduces the ratio between them, leading to more sneak path current through unwanted cells and thus greater passive energy consumption; this further limits suitability for edge application. For the purposes of this work, the MoS₂ memtransistor architecture strictly utilizes binary conductance states (LRS and HRS) to demonstrate a method of circumventing these issues through dynamic gate modulation during readout. Unlike traditional resistive memory technologies (RRAM, PCM, etc.), which rely on multilevel conductance states to represent synaptic weights, our approach separates data storage from signal modulation. By adjusting the gate voltage within a controlled range, we can effectively modulate the channel conductivity without altering the stored conductance state. The experimental characterization shown in **Supplemental Information 8** of the original manuscript (reproduced here as **Figure R7** for reference) shows that >6 distinct output levels can be produced from each binary state, offering a mechanism to emulate intermediate synaptic strengths without the time/energy costs and variability and drift issues commonly associated with analog memory writes.

This capability improves system-level performance in several ways. First, it increases output separability, which is especially valuable for tasks where resolution must be preserved. Second, gate tuning serves as a runtime calibration mechanism that compensates for device non-idealities or variation without requiring reprogramming or retraining. The results presented in the main manuscript confirm that classification accuracy lost to binary network limitations can be substantially recovered through modest gate adjustment. Third, as discussed in our responses above, the energy cost of this modulation is low. Together, these elements indicate that, unlike multilevel resistive memories, which require iterative tuning and exhibit degraded endurance at intermediate states, our method retains the simplicity and stability of binary memory while possessing some of the flexibility of analog systems.

In practice, this approach allows for post-write adjustment of effective synaptic weights, offering a way to implement adaptive behavior with minimal hardware overhead. While not intended to replace multilevel storage where precision is essential, dynamic gate

modulation provides an efficient alternative for inference tasks where robustness, low power, and tolerance to device variation are more critical than static precision.

We have adjusted our statement regarding the benefits of dynamic modulation as follows in the revised manuscript:

The ability of our memtransistor architecture to modulate preprogrammed conductance states through the application of a modulatory back-gate bias (V_{mod}) is shown in **Supplemental Information 8** (included here as **Figure R7** for reference). It can be seen that >6 distinct output levels can be produced from a single binary state, thus offering a mechanism to emulate intermediate synaptic strengths without the time/energy costs and variability and drift issues commonly associated with analog memory writes. As we will discuss in further detail below, this ability to control conductance states (weights) independently from the application of program/erase operations provides an extra degree of freedom when performing inference operations, allowing us to dynamically potentiate or depress weights to more clearly differentiate between outputs in high confusion. This capability stands to improve system-level performance in multiple ways. First, it increases output separability, which is especially valuable for tasks where resolution must be preserved. Second, gate tuning serves as a runtime calibration mechanism that compensates for device non-idealities or variation without requiring reprogramming or retraining. Together, these elements indicate that, unlike multilevel resistive memories, which require iterative tuning and exhibit degraded endurance at intermediate states, our method retains the simplicity and stability of binary memory while possessing some of the flexibility of analog systems. While not intended to replace multilevel storage where precision is essential, dynamic gate modulation provides an efficient alternative for inference tasks where robustness, low power, and tolerance to device variation are more critical than static precision.

9. Pg 16 – Binarized Inputs: The use of binarized inputs appears to be a device limitation

rather than a design choice. Moreover, the network's low recognition accuracy (~62%) on the MNIST dataset suggests that this hardware may struggle with more complex datasets, contradicting the authors' claims about edge applications. The authors should explore increasing the number of weight states per memory device to address this concern and improve recognition accuracy. If this is not feasible, a discussion explaining these limitations should be included.

The reviewer presents a reasonable concern regarding the use of binarized weights and the low observed MNIST classification accuracy of 62.8%. They further raise two interrelated questions: (i) whether the MoS₂ memtransistors utilized in this work are inherently limited to two memory states (ON/OFF) and (ii) whether the low inference accuracy undermines this work's relevance to real-world edge applications. We find both questions extremely relevant to the work as presented in the main manuscript and agree that both topics should be addressed in the revised manuscript.

First, as mentioned above, we have already demonstrated analog memory based on our MoS₂ memtransistor approach in previous works¹⁶⁻¹⁸; for this work, weight binarization was a deliberate choice, not solely a device constraint. Binary neural networks (BNNs) have been recognized as well-suited choices for edge platforms where compute energy and memory footprint must be tightly controlled^{39,45}. For such use cases, binarization serves three purposes: (i) it minimizes the need for high-resolution analog memory states, which require costly write-verify schemes to program³⁵ and are prone to error due to the low read margin between different states^{39,42}; (ii) it simplifies on-chip multiply-accumulate operations to XNOR and bitwise logic^{45,46}, reducing active energy costs, while also terminating sneak paths through unwanted cells⁴⁷, reducing passive energy costs; and (iii) it aligns with our memtransistors' ability to store robust binary states while retaining the ability to emulate higher effective weights through dynamic gate modulation to increase inference accuracy. Thus, in the context of this work, weight binarization serves as an optimization approach for energy efficiency and robustness, rather than a workaround for an intrinsic device limitation.

Second, while the simulated MNIST inference accuracy of 62.8% is undoubtedly low by conventional analog standards, we feel that it must be discussed/understood in the context of this work's binary approach, which uses downscaled 8×8 images and a simplified network topology (i.e., single layer neural network with no peripheral neuronal circuits for activation function implementation) to match the hardware array's dimensions. Our primary goal was not to achieve state-of-the-art accuracy but instead to validate closed-loop inference using a dense, large-scale hardware array with non-volatile and gate-modulated weight elements. However, as discussed below, we have made further efforts to refine the neural network simulation and hardware implementation in the revised manuscript to address the reviewers' comments, achieving a simulated accuracy of 88.1% and an experimental accuracy of 85.6% for 13×13 images.

We have adjusted the following discussion to clarify the rationale for weight binarization in the revised manuscript:

To verify the ability of our MoS₂-memtransistor-based crossbar array architecture to function as an inference engine, a proof-of-concept inference demonstration aiming to classify simple shapes in hardware was first performed. It should be noted that for this demonstration, and all others discussed in this work, the analog weights achievable using our MoS₂ memtransistor architecture¹⁶⁻¹⁸ are deliberately binarized between logic “0” and logic “1”. Binary neural networks (BNNs) have been recognized as well-suited choices for edge platforms where compute energy and memory footprint must be tightly controlled^{39,45}. For such use cases, binarization serves three purposes: (i) it minimizes the need for high-resolution analog memory states, which require costly write-verify schemes to program³⁵ and are prone to error due to the low read margin between different states^{39,42}; (ii) it simplifies on-chip multiply-accumulate operations to XNOR and bitwise logic^{45,46}, reducing active energy costs, while also terminating sneak paths through unwanted cells⁴⁷, reducing passive energy costs; and (iii) it aligns with our memtransistors' ability to store robust binary states

Figure R8. Demonstration of MNIST Handwritten Digit Classification on a 2 kb Memtransistor Crossbar Array. a) Optical images of i) a 1.5×1.5 cm chip containing a 64×32 crossbar array comprising 2,048 MoS₂ memtransistors, ii) a zoomed-in image showing the full 2 kb array (scale bar of 100 μm), and iii) a further zoomed-in image showing nine constituent memtransistors (scale bar of 10 μm). b) Schematic showing preprocessing performed on training and test (inference) images taken from the Modified National Institute of Standards and Technology (MNIST) database. The original images (28×28 pixels) were downsampled to 13×13 pixels and binarized to fit into a 64×30 sub-array for this demonstration. Downsampled images were then converted to 169×1 input vectors and split into three sub-vectors for input to the array. A dataset comprising 10,000 resized/reshaped images were then used for training and weight assignment; for this demonstration, simulated weights were split between logic “0” and “1” and later converted to targeted conductance states for hardware implementation. c) Heatmap showing the distribution of simulated weights following training. A test dataset of 1,000 resized/reshaped MNIST images was fed to the simulated array for verification of network inference/classification. d) Confusion matrix showing the classification results for the inference check described in (c). An overall accuracy of 88.1% was achieved. e) Heatmap showing the distribution of conductance states assigned to the hardware array in respect to the simulated weight distribution shown in (c), with weights of “1” mapped to a conductance state of ~50 nS and weights of “0” mapped to the OFF-state conductance (a few pS). Cells marked NaN either display an open circuit or high gate leakage; the overall yield of devices remained high at ~92.2%. f) Hysteresis characteristics of the 1,770 working devices in the 64×30 sub-array extracted at V_{DS} = 1 V, which show low intrinsic device-to-device variation, memory windows >5 V, and read margins >10⁴. g) Histogram showing the distribution of the final conductance states (weights) represented in (e). h) Confusion matrix showing the classification results for hardware-based inference performed on the memtransistor array shown in (a) as per the conductance state (weight) distribution shown in (e). A test dataset comprising 1,000 resized/reshaped MNIST images was applied to the drain terminals of the array as voltage inputs (either 0 V or 1 V depending on the corresponding pixel value); output currents along corresponding columns/nodes were then individually registered and compared to all other outputs to determine the inferred digit for each case. An overall accuracy of 85.6% was registered.

while retaining the ability to emulate higher effective weights through dynamic gate modulation to increase inference accuracy, as will be discussed later.

We have adjusted our discussion on implementing MNIST classification operations in hardware using our MoS₂-memtransistor-based crossbar array architecture as follows in the revised manuscript:

Expanding our efforts to more closely investigate real-world ANN applications, we then performed single-layer neural network inference operations aiming to classify handwritten digits from the MNIST datasets. An overview of this demonstration is shown in **Figure 3** (included here as **Figure R8** for reference). For this demonstration, a 64×32 (2,048 devices, 2 kb) crossbar array was prepared, with an optical image of the final chip containing the array being shown in **Figure R8a** along with zoomed-in optical images of the 2 kb array and constituent NVM cells. A simple binary classification scheme was then tested with the goal of classifying all ten digits (0-9) comprising the MNIST dataset. As schematically represented in **Figure R8b**, MNIST images (28×28 pixels) were first downsampled to 13×13 pixels using

bicubic interpolation and transformed into 169×1 vectors to serve as inputs. To simplify the experimental demonstration, the greyscale pixels of the MNIST dataset were also binarized to either 0 or 1 during this process, making each image black and white; further discussion on this preprocessing, and on its effects on inference accuracy, can be found in the **Methods** section. Software-based training was then used to prepare a binary weight map from a training dataset consisting of 10,000 MNIST images, as shown in **Figure R8c**. Testing of the simulated single-layer network was then performed using a dataset consisting of 1,000 randomly selected binarized 13×13 images, curated such that each digit (0-9) was represented exactly 100 times to ensure fair assessment of the network's ability to infer different digits. The results from this simulation-based MNIST inference test are presented by the confusion matrix in **Figure R8d**. An overall accuracy of 88.1% was achieved; while digits such as "0", "1", and "8" demonstrated high classification accuracy, the overall accuracy was constrained by the network's limited ability to classify digits such as "5" which demonstrated a high degree of confusion with "8" and "3". While non-ideal, this relatively low overall accuracy is to be expected due to the nature of the problem presented, i.e., single-layer inference across a large dataset consisting of ten highly compressed and binarized handwritten digits. Following simulation, the binary weights of the single-layer network were mapped to corresponding conductance states in a 64×30 sub-array of the 2 kb array shown in **Figure R8a**, with weights of "1" being mapped to a conductance state of ~ 50 nS and weights of "0" being mapped to the OFF-state conductance (a few pS). A map of the final conductance states across the sub-array is shown in **Figure R8e**; here, NVM cells marked NaN either display an open circuit or high gate leakage, both of which are believed to stem from damage to the corresponding horizontal drain line. Note that, despite the loss of five complete rows to these non-idealities, the overall yield of devices across the sub-array remained high at $\sim 92.2\%$. Hysteresis characteristics and final conductance states (weights) for the working cells are also shown in **Figure R8f-g**, respectively, demonstrating good uniformity across the breadth of the array as well as a significant read margin of $> 10^4$ between differently weighted cells. For the sake of clarity, devices demonstrating a measured conductance below the noise floor conductance of the characterization system

($\sim 10^{-12}$ S) are marked as being at the noise floor. For this hardware-based inference demonstration, the outputs from three columns of the sub-array correspond to a single digit utilized in the MNIST dataset, e.g., the first three columns in the sub-array correspond to the digit “0”, the next three columns correspond to “1”, etc. When an image (binary 169×1 vector split into three sub-vectors) is applied as voltage inputs (either 0 V or 1 V depending on the corresponding pixel value) to the drain terminal, the output currents from each column set (node) are individually registered. The summed output currents from each node are then compared to determine the inferred digit. Please note that, due to limitations with the experimental setup, summation of output currents was performed externally for the sake of this demonstration. The results for this classification scheme are shown in **Figure R8h**. An overall accuracy of 85.6% was registered for the hardware demonstration; at only 2.5% lower than simulated, a similar simulation-to-hardware gap to previous MNIST classification explorations⁴⁸, this indicates that the MoS₂-based crossbar arrays developed and demonstrated in this work can successfully implement inference operations with a high degree of similarity to simulations, thus indicating potential as neuromorphic accelerators. Deviations from the ideal case demonstrated in **Figure R8d** can be attributed to device-to-device variation and non-ideal yield and can be optimized for future work through array-level compensation schemes⁴⁹⁻⁵¹ and optimization of the fabrication and MoS₂ growth processes.

10. Pg 19 – Comparison with Other Hardware Technologies: The authors present a comparative table without discussion. Strengthening this section with a quantitative analysis of their methodology’s strengths and weaknesses relative to existing technologies would improve the manuscript. Additionally, emerging non-volatile technologies such as oxide-based memristors and FeFETs should be included in the comparison.

We thank the reviewer for their suggestion of a quantitative analysis of our methodology’s strengths and weaknesses relative to other existing and emerging technologies. We would like to point out to the reviewer that a similar discussion was already present in

Supplementary Information 14 of the original manuscript. However, we do acknowledge that this discussion does not include a comparison of charge-trapping memories (such as that used in this work) with other emerging non-volatile memory options, such as oxide-based memristors, ferroelectric field-effect transistors (FeFETs), or phase-change memories, and agree that a more quantitative analysis of their respective tradeoffs and benefits would only serve to strengthen the manuscript. We have now added such a discussion to the revised manuscript, as detailed below.

We have added the following discussion comparing non-volatile memory technologies to Supplemental Information 4 of the revised manuscript:

2D memtransistors such as those explored in this work offer several compelling advantages over conventional neuromorphic memory technologies such as resistive random-access memory (RRAM), phase-change memory (PCM), and ferroelectric memories. Conventional two-terminal RRAM devices frequently encounter challenges related to variability⁵² and sneak path currents, typically requiring access/selector devices in each NVM cell that increase complexity and footprint^{2,3,52-54}. Three-terminal memtransistors address these issues with electrostatic gating, providing lower leakage currents and more linear conductance modulation^{2,3}. While RRAM technology is relatively mature and can be densely integrated, 2D memtransistor arrays achieve comparable density with the added benefit of atomically-thin channels that could enable 3D stacking of layers⁵⁵⁻⁵⁷. An overview of the NVM cell areas achieved in various MoS₂-based crossbar array demonstrations is presented as part of the benchmarking in **Supplemental Table 1** (included here as **Table R1** for reference). However, a major trade-off is that MoS₂ devices are at an earlier stage; improving their endurance and uniform wafer-scale fabrication are ongoing research challenges that are beyond the scope of this study. PCM devices, while relatively stable and mature, rely on thermally induced phase changes that lead to higher energy consumption and slower speeds compared to the charge-trapping mechanisms in MoS₂ devices⁵². Notably, PCMs

Table 1: Crossbar-Level Demonstrations based on 2D Materials											
Device Structure	Active Length/Width	Cell Area	Yield (%)	Array Size	Array Configuration	Switching Energy Consumption	Retention (Exp./Calc.)	ON/OFF Ratio	Terminals	Multi-level Capability?	Reference
MoS ₂ Memtransistor	1 um / 1 um	676 um ²	92.2	64×32	1T	~ 20 pJ (base), ~ 0.2 fJ (peak)	> 4*10 ⁵ s / ~3.17 yrs	~10 ⁵	3	Y	This Work
MoS ₂ Memtransistor	1 um / 1 um	~ 105 um ²	100	16×10	1T	~ 20 pJ (base), ~ 0.2 fJ (peak)	> 4*10 ⁵ s / ~3.17 yrs	~10 ⁵	3	Y	This Work
MoS ₂ Memtransistor	1 um / 1 um	~ 51.5 um ²	99.4	16×10	1T	~ 20 pJ (base), ~ 0.2 fJ (peak)	> 4*10 ⁵ s / ~3.17 yrs	~10 ⁴	3	Y	This Work
MoS ₂ FGFET	~3.1 um / ~49.5 um	~ 32500 um ²	83.1	32×32	1T	NA (~ 8 - 20 nJ estimated)	> 3.5*10 ³ s / NA	~ 100 - 300	3	Y	[1]
MoS ₂ Memtransistor	0.4 um / 20 um	~ 4000 um ²	64	10×10	1T	~ 20 fJ	> 800 min / NA	> 100	3	Y	[2]
MoS ₂ Memtransistor	0.9 um / 0.7 um	~ 5 um ²	NA	10×9	1T	~ 20 nJ	> 10 ⁵ s / > 10 yrs	> 1000	4	Y	[3]
MoS ₂ Memristor	1 um / 50 um	~ 7250 um ²	NA	2x2	1T-1R	NA	> 3.6*10 ³ s / NA	~ 10 ⁵ - 10 ⁶	2	Y	[4]
Au/h-BN/Au Memristor	5 um / 5 um	~ 100 um ²	NA	10×10	1R	5.47 pJ	NA / NA	~10 ⁶	2	Y	[5]
Au/MoS ₂ /Ag Memristor	30 um ²	NA	NA	4x4	1R	NA	> 2*10 ⁵ s / NA	~10 ⁵	2	N	[6]
Pt/MoS ₂ /Ag Memristor	10 um / 10 um	NA	NA	4x4	1R	20 uJ	> 10 ⁴ s / NA	~ 10	2	N	[7]
Pt/MoS ₂ /Ti Memristor	5 um / 5 um	~ 900 um ²	NA	6x6	1R	~ 40 pJ	> 10 ⁵ s / > 3 yrs	~ 100	2	Y	[8]
Au/h-BN/Au Memristor	3 um / 3 um	~25 um ²	98 (DA)	10×10	1R	~ 2.9 uJ	> 100 s / NA	> 10 ⁵	2	Y	[9]

NA: Not announced

DA: Different arrays; data taken across multiple experimental crossbar arrays

Table R1. Crossbar-Level Demonstrations based on 2D Materials. Benchmarking of this work against extant demonstrations of crossbar arrays based on 2D materials¹⁻⁹. The cell areas for the arrays developed in this work range from 676 μm² (base design) to 51.5 μm² (densest design), demonstrating our ability to successfully scale our crossbar array architectures to information densities of up to 1.94 Mb/cm² (assuming 1-bit operation). However, even the base design displays a significantly higher information density than almost all experimentally demonstrated 2D-material-based arrays to-date; while some reports have demonstrated cell areas down to ~5 μm² per cell, the ultimate array size of those works was significantly less than that achieved in this effort. Notably, only a single other work has demonstrated comparable array sizes to ours, though at a significantly lower yield than what we have thus far demonstrated. Our array architectures also compare favorably in terms of switching energy, ranging from tens of pJ (base) to below 1 fJ (peak); all switching energies shown here were estimated using the equation Energy = Time × Current × Voltage, which is commonly used to estimate switching energy for NVMs. The terms “base” and “peak” included in the assessment of our work refer to the pulse time (switching time), with base referring to our typical pulse time of 100 ns and peak referring to our minimum confirmed pulse time of 1 μs. While several of the other works included in this comparison also utilized ultrafast switching times, our NVM capabilities being controlled by the gate means that the switching current of our devices is limited to the gate leakage current, which remained in the region of several tens of pA even at the largest gate biases applied (~10 V), thus allowing for severely reduced energy expenditure. Other metrics (retention, ON/OFF ratio, number of terminals, and ability to realize multiple memory states) compare similarly to other demonstrations, indicating suitability for future applications.

have been demonstrably integrated into memory chips, though adding them onto logic circuits is non-trivial due to thermal budget constraints⁵⁸. In contrast, MoS₂ devices are relatively flexible: they can be fabricated at back-end compatible conditions or transferred onto substrates, potentially allowing monolithic integration of sensing, memory, and logic on the same chip^{55,56}. Ferroelectric memories, though CMOS-compatible and fast-switching, often exhibit abrupt polarization transitions, complicating analog weight tuning⁵⁹. MoS₂ memtransistors, by contrast, inherently offer more linear, gradual, and energy-efficient analog conductance updates^{2,3,12,60-62}.

It should also be noted that numerous works have investigated hybrid memory architectures (e.g., an MoS₂ channel FET with a ferroelectric gate material^{26,63,64}) to harness the strengths of each. These technologies, while promising, use different material stacks and operational principles from those explored in this work, and as such represent a complementary rather than directly comparable approach. Overall, while each memory technology has distinct advantages, we believe the combination of non-destructive operation, runtime tunability, and integration flexibility offered by our MoS₂ memtransistor crossbar arrays provides a compelling path for future low-power, adaptable neuromorphic systems.

11. Pg 20 – Identification of Similar Features Between “5” and “8”: The authors state that similar features between “5” and “8” make them difficult to distinguish, and that dynamic inference adjustments using the third terminal help enhance certain features. However, as noted in Comment 7, where is the information about which features to enhance stored, and how is it extracted to feed into the crossbar array? The authors should provide an example of this operation and determine the energy and footprint costs to enable a fair comparison with other architectures.

The reviewer again raises a good question regarding the need to access some form of external/peripheral data storage, such as a lookup table or memory unit, to implement the dynamic weight modulation scheme discussed in this work. As discussed in our response to

the reviewer's other comment above, we agree that such peripheral circuitry would be needed for practical implementation. Here, we detail a proposed approach to store and access application-relevant information for dynamic gate modulation. First, inference ambiguities are identified by a readout layer which monitors the difference between dedicated output nodes (ΔI_{out}). If this value falls below a predefined threshold (e.g., 5% of maximum difference) for a given input, the output (inferred digit) is flagged as uncertain. This function could be achieved using a comparator circuit. Following inference, flagged outputs are subjected to dynamic modulation. A control unit triggers a lookup into an external memory table storing modulatory gate biases (V_{mod}) for the relevant output nodes. These V_{mod} values would be preassigned during network training based on the classes being analyzed and reflect coarse bias adjustments tailored to enhance the separability of commonly confused class pairs (e.g., within +/- 2 V for the demonstrations discussed in the main manuscript). This allows the system to resolve ambiguous cases through a single re-read, rather than reprogramming device states or performing retraining. The peripheral overhead for this process would be modest, consisting of the comparator circuit for ΔI_{out} evaluation and flagging, the external lookup table (with the size depending on the number of high confusion classes and desired V_{mod} resolution), and a shared DAC or multiplexer to apply V_{mod} to the array. Of course, it is important to acknowledge that DAC/ADC operation significantly contributes to the energy consumption of neuromorphic accelerator chips⁴³; however, as modulation is only triggered for a subset of inference cases, often defined by task-specific importance (e.g., hazard detection in surveillance), the power draw for this use case should not meaningfully contribute to the overall power budget.

We have adjusted the following statement on dynamic gate modulation scheme to also posit how it may be implemented in the revised manuscript:

One significant advantage of utilizing memtransistors as NVMs as opposed to more mature resistive memory technologies is their multi-terminal nature. This allows for devices to be modulated through both the non-volatile program/erase operations mentioned previously

and through dynamic applications of V_{BG} during read operations. By varying V_{BG} during reads, the conductance state (weight) programmed into a device can be selectively tuned either higher (through application of a positive gate bias) or lower (through application of a negative gate bias) as needed without expending the energy required for a switching event; as read margin will vary depending on where it is assessed in the memory window, this approach may also be used to increase or decrease the ratio between states as needed for different applications. Notably, no charge-trapping is required for modulating weights in this manner, meaning that this can be performed at the default read speed (clock speed) of the system being used to conduct logic operations in the array.

To implement this approach in hardware, inference ambiguities would first need to be identified via a readout layer, which monitors the difference between dedicated output nodes (ΔI_{out}). If this value falls below a predefined threshold (e.g., 5% of maximum difference) for a given input, the output (inferred digit) would be flagged as uncertain. This function could be achieved using a comparator circuit. Following inference, flagged outputs would be subjected to dynamic modulation. A control unit would then trigger a lookup into an external memory table storing modulatory gate biases (V_{mod}) for the relevant output nodes. These V_{mod} values would be preassigned during network training based on the object classes being analyzed and reflect coarse bias adjustments tailored to enhance the separability of commonly confused class pairs (e.g., within +/- 2 V for the demonstrations discussed in this work). This allows the system to resolve ambiguous cases through a single re-read, rather than reprogramming device states or performing retraining. The peripheral overhead for this process would be modest, consisting of the comparator circuit for ΔI_{out} evaluation and flagging, the external lookup table (with the size depending on the number of high confusion classes and desired V_{mod} resolution), and a shared DAC or multiplexer to apply V_{mod} to the array. Of course, it is important to acknowledge that DAC/ADC operation significantly contributes to the energy consumption of neuromorphic accelerator chips⁴³; however, as modulation would only be triggered for a subset of inference cases, often defined by task-specific importance (e.g., hazard detection in surveillance), the power draw for this use case

Figure R9. Bias Scheme Testing. (a-c) Testing of two devices (R1C1 and R2C1) on the same word/gate-line of a representative MoS₂-memtransistor-based crossbar array using a full biasing scheme. (a) The as-fabricated devices are read before being (b) programmed to high conductance states using a negative voltage pulse ($V_{\text{Program}} = -10$ V) and (c) erased to their initial conductance states using a positive voltage pulse ($V_{\text{Erase}} = +10$ V). All other access lines are held at 0 V during pulsing. (d-f) Testing of R1C1 and R1C2 using a half-biasing scheme. (d) The devices are read before being (e) subjected to a negative gate voltage pulse ($V_{\text{Program}} = -5$ V) while the source of Row 1 ($V_{\text{S,R1}} = +5$ V) and the source of Row 2 ($V_{\text{S,R2}} = -5$ V). This maximizes the gate-to-source voltage (VGS) across R1C1 while minimizing the VGS across R2C1; as a result, only R1C1 experiences a change in its conductance state. When the biases are then flipped (f), R1C1 is again the only device to display any change, returning closer to its initial conductance state.

should not meaningfully contribute to the overall power budget.

12. *Supplementary S8 – Device-to-Device Variation:* Figure S8 shows significant device-to-device variation, particularly in threshold voltages between R1C1 and R2C1. The authors should discuss how this variability impacts MNIST classification accuracy and whether it contributes to confusion between similar outputs during inference.

Figure R10. Word/Gate-Line Isolation Testing. (i-v) Demonstration of gate line isolation in a representative 2×2 crossbar array. Dotted lines are to help show shift in transfer characteristics after each program/erase operation. All four as-fabricated devices in the 2×2 array (R1C1, R1C2, R2C1, and R2C2) were read (i) before the word/gate-line in Row 2 (R2) was sequentially subjected to (ii) a -10 V bias pulse ($V_{G,R2}$), (iii) a $+10$ V bias pulse, (iv) a -10 V bias pulse and (v) a -10 V bias pulse, for 100 ms each. As can be seen, while the devices in R2 (R2C1 and R2C2) were programmed and erased as expected when exposed to negative and positive bias pulses, respectively, the devices in R1 (R1C1 and R1C2) did not show any appreciable shift in their transfer characteristics throughout all applied pulses, confirming the isolation of the separate word/gate lines.

We apologize, but we are unsure of what “significant device-to-device variation” the reviewer is referring to in **Figure S8** from the original manuscript (included here as **Figure R9** for

reference). The initial states for devices R1C1 and R2C1, as represented by their respective transfer characteristics in the absence of any program/erase events, are shown to be extremely similar for both the full biasing scheme (**Figure R9a**) and the half-biasing scheme (**Figure R9d**). If the reviewer is instead referring to the programmed (**Figure R9b,e**) and/or erased (**Figure R9c,f**) cases, we would like to point out that these were taken for the same devices following application of program/erase biases, as outlined in the main text and again below, and should not be misconstrued for device-to-device variation. We hope that this adequately addresses the reviewer's comment but are open to further discussion if we have misunderstood the reviewer's intent.

13. Supplementary S9 – Inconsistency in Programming and Erase Voltages: The programming and erase voltages are both listed as -10V, which seems unusual. The authors should clarify whether this is a typo.

We appreciate the reviewer's concern regarding the nomenclature used in **Figure S9** (included here as **Figure R10** for reference). However, we believe that the reviewer misunderstood the order of operations presented. All four as-fabricated devices in the 2×2 array (R1C1, R1C2, R2C1, and R2C2) were read (**Figure R10i**) before the word/gate-line in Row 2 (R2) was sequentially subjected to (**Figure R10ii**) a -10 V bias pulse ($V_{G,R2}$), (**Figure R10iii**) a +10 V bias pulse, (**Figure R10iv**) a -10 V bias pulse and (**Figure R10v**) a -10 V bias pulse, for 100 ms each. In other words, the order of operations was thus program-erase-program-program, which matches that given by the figure.

Reviewer #2 (Remarks to the Author):

Comments:

This manuscript demonstrates large-scale crossbar arrays based on MoS₂ memtransistors for in-memory computing, achieving high yield (>92%) and low write energy (~0.2 fJ) across

arrays of up to 2048 devices. The three-terminal design enables dynamic conductance tuning through gate modulation, aiming to resolve inference ambiguities without costly retraining or reprogramming. This demonstration is interesting and highlights the potential of MoS2 memtransistors for edge AI applications. I recommend its publication in Nature Communications after addressing the questions listed below.

We thank the reviewer for their comments and for their recommendation of publication in *Nature Communications* upon revision.

1. The manuscript reports an information density of 0.15 Mb/cm² and 1.94 Mb/cm² on page 7. It would be beneficial to explicitly provide the calculation method to enhance clarity and reproducibility.

We agree with the reviewer that explicit disclosure of how information density was calculated would enhance clarity and reproducibility for readers. For the sake of this work, information density was defined as the amount of storable information (i.e., weights) in bits per cm². This was estimated assuming 1-bit (two-state) operation for each individual NVM cell, with area calculated from the length and width of the repeating unit for each respective crossbar design (i.e., cell area).

We have adjusted the following statement to better define integration/information density in the revised manuscript:

This allows us to achieve an information density, defined as the amount of storable information (i.e., weights) in bits per cm², of ~0.15 Mb/cm². Note that, for the purposes of this work, 1-bit (two-state) operation is assumed in all cases for the purposes of calculating information density.

Figure R11. Overview of Condensed Crossbar Array Architecture. a) Optical image of a representative 16×10 MoS₂-memtransistor-based crossbar array based on an alternative design from that reported in **Figure 1**, denoted here as ‘Condensed’. This design features a non-volatile memory (NVM) cell area of 105 $\mu\text{m}^2/\text{cell}$, an ~84.5% reduction from the base design discussed in **Figure 1** and elsewhere in the main text; this corresponds to an information density of ~0.95 Mb/cm² for 1-bit operation. Inset shows a zoomed-in image of constituent memtransistors. Scale bar denotes 25 μm (10 μm for inset). b) Three-dimensional scatter plot showing distribution of ON-state and OFF-state currents taken at a drain-to-source voltage (V_{DS}) of 1 V, denoted as I_{ON} (pink) and I_{OFF} (cyan), respectively, across the 16×10 array. Notably, all 160 devices in the array were found to work (100% yield) despite the reduction in cell area from the base design. c-d) Maps of threshold voltage (V_{th}) and subthreshold slope (SS), respectively, across the array. e-f) Histograms of V_{th} and SS, respectively, for the 160 devices in the array.

We have also added the following statement to better define cell area in the revised manuscript:

Here, cell area is defined as the area (Length × Width) of the repeating unit for a given crossbar design.

2. Supplementary Information 2 and 3 present two more condensed designs with cell areas of 105 $\mu\text{m}^2/\text{cell}$ and 55 $\mu\text{m}^2/\text{cell}$, respectively. A detailed structural analysis of how to achieve such cell sizes is recommended.

We thank the reviewer for bringing attention to the condensed crossbar array designs presented in the Supplemental Information and agree that providing more details on the distinctions between each design would provide greater clarity for readers. The greatest point of distinction is the number of shared access line (input/output line) terminals shared between adjacent cells. The base design presented in the main manuscript utilizes individual terminals for each cell, while the ‘Condensed’ design presented in **Supplemental Information 2** (included here as **Figure R11** for reference) utilizes shared output line terminals between cells in adjacent rows and the ‘Dense’ design presented in **Supplemental Information 3** (included here as **Figure R12** for reference) utilizes shared input and output line terminals between cells, effectively halving the horizontal routing density while maintaining distinct gate and output (source) lines. This distinction allows for greater area efficiency while preserving device functionality and access line dimensions. Please note that the fabrication process described in the Methods section of the main manuscript was universally applied for all designs discussed, with the only exception being the pattern inscribed during lithography. These alternative crossbar designs were developed to demonstrate the feasibility of improving integration/information density through layout optimization alone, without modifying the device stack or fabrication process flow. Further improvements in density are conceivable by advancing to smaller technology nodes or adopting multilayer integration schemes.

Figure R12. Overview of Dense Crossbar Array Architecture. a) Optical image of a representative 16 \times 10 MoS₂-memtransistor-based crossbar array based on an alternative design from that reported in **Figure 1**, denoted here as ‘Dense’. This design features an NVM cell area of 51.5 $\mu\text{m}^2/\text{cell}$, an \sim 92.4% reduction from the base design discussed in **Figure 1** and elsewhere in the main text; this corresponds to an information density of \sim 1.94 Mb/cm² for 1-bit operation. Inset shows a zoomed-in image of constituent memtransistors. Scale bar denotes 20 μm (5 μm for inset). b) Three-dimensional scatter plot showing distribution of I_{ON} (pink) and I_{OFF} (cyan) taken at $V_{DS} = 1$ V across the 16 \times 10 array; devices/cells marked in gray registered as an open circuit (OC) when measured. Notably, 159/160 devices in the array were found to work (99.4% yield) which is comparable to the base design despite the reduction in cell area. c-d) Maps of V_{th} and SS, respectively, across the array. e-f) Histograms of V_{th} and SS, respectively, for the 159 working devices in the array.

3. Table 2 of the supplementary information does not mention energy efficiency (TOPS/W), a key metric for assessing the practical viability of neural network accelerators. While the table provides detailed comparisons of throughput and computational density, it lacks power consumption data, making it difficult to evaluate the overall efficiency of the proposed MoS₂-memtransistor-based accelerator in relation to other architectures. To strengthen the comparative analysis and provide a more comprehensive assessment of the accelerator's performance, I recommend incorporating energy efficiency data.

The reviewer makes an excellent point regarding the use of throughput power efficiency (TOPS/W) as an important benchmarking metric for neural network accelerators. Furthermore, we wholeheartedly agree with the reviewer's stance that a thorough assessment of our proposed MoS₂-memtransistor-based accelerator's throughput power efficiency would strengthen our comparison with other emerging logic accelerators. However, we must stress that we purposefully chose to omit such benchmarking/discussion from the manuscript since throughput power efficiency estimations generally account for the energy consumption by peripheral circuitry, which can be an appreciable percentage of the entire energy budget^{43,65}. As the crossbar array architecture investigated in this work was tested without integration of peripherals, we felt that benchmarking any estimated throughput power efficiency to that of other architectures, even with proper disclaimer, could misrepresent our results and lead to a faulty perception of system efficiency. Still, we recognize that omitting any discussion of power efficiency could be viewed as limiting.

For the MNIST inference demonstration presented in the main manuscript, devices in the LRS state (logic "1") are set to a conductance of ~50 nS. When a black pixel is fed to the array ($V_{in} = 1$ V), this results in a dynamic energy consumption of 5 nJ/cell for a 100 ms read time (used for characterization and testing); this energy consumption can be as low as 0.2 pJ/cell for the experimentally confirmed minimum reliable read time of 4 μ s presented in the Supplemental Information, which we consider the more relevant value for inference applications. Conversely, devices in the HRS state (logic "0") are set to a conductance of ~1 pS and dissipate comparatively negligible energy: approximately 0.1 pJ for a 100 ms read)

and 4 aJ for a 4 μ s read. Assuming an active device density of $\sim 50\%$ (960 LRS devices across a 64×30 sub-array), the total sub-array-level dynamic read energy is estimated at $\sim 4.8 \mu$ J under slow readout conditions and as low as ~ 192 pJ for fast readout, with an active power consumption of $\sim 48 \mu$ W. Note that this estimation precludes contributions from sneak path current due to the tendency of binary weights to terminate sneak paths. Although a full TOPS/W metric requires knowledge of system-level power requirements including peripheral circuitry (DACs, ADCs, multiplexers, lookup tables, etc.)⁴³, these values allow us to conservatively estimate the energy efficiency of the array itself. We may estimate the throughput, in TOPS, using the equation:

$$\text{Throughput} = \frac{\text{Inputs}(\text{Rows} \times \text{Columns})}{\text{Time}}$$

Here, *Inputs* refers to the number of inputs applied to the array, *Rows* and *Columns* refers to the number of rows and columns in the array being utilized, and *Time* refers to the operational time ($1/f$). For preliminary assessment of the throughput of our arrays, the number of columns being utilized at a time is restricted to 1 due to experimental limitations in assessing outputs in parallel; improvements in parallelization capabilities would therefore lead to a substantial improvement in throughput for all cases. In this manner, a base throughput of 1.23×10^{-7} TOPS is estimated for a read time of 100 ms and a peak throughput of 3.06×10^{-3} TOPS is estimated for a read time of 4 μ s. From the discussion above, this translates to a power efficiency of ~ 638 TOPS/W for the crossbar array under peak conditions. It is extremely important to note that this high value represents a theoretical upper limit of efficiency and does not account for peripheral power draw. For example, previous investigations report that ADCs alone can account for 40–50% of the total power in similar systems⁴³. We therefore expect that accounting for even minimal peripheral load will reduce the overall system-level efficiency by at least one or two orders of magnitude.

We have added the following statement on throughput power efficiency to Supplemental Information 18 of the revised manuscript to provide context on any perceived lack of energy efficiency benchmarking:

Also note that estimations of throughput power efficiency (TOPS/W), a widely regarded metric for assessing neural network accelerator viability in the context of energy consumption, have been purposefully omitted from this benchmarking table to prevent any misleading comparisons of the power consumption of our work. As power consumption by peripherals can be an appreciable percentage of the entire power budget^{43,65}, widespread consensus holds that a comparative analysis of power consumption between different accelerator architectures must account for peripherals to present an accurate perception of system efficiency. A discussion of power efficiency with this caveat in mind is presented below.

For the MNIST inference demonstration presented in the main manuscript, devices in the LRS state (logic “1”) are set to a conductance of ~ 50 nS. When an active pixel is fed to the array ($V_{in} = 1$ V), this results in a dynamic energy consumption of 5 nJ/cell for a 100 ms read time (used for characterization and testing); this energy consumption can be as low as 0.2 pJ/cell for the experimentally confirmed minimum reliable read time of 4 μ s presented in the Supplemental Information, which we consider the more relevant value for inference applications. Conversely, devices in the HRS state (logic “0”) are set to a conductance of ~ 1 pS and dissipate comparatively negligible energy: approximately 0.1 pJ for a 100 ms read) and 4 aJ for a 4 μ s read. Assuming an active device density of $\sim 50\%$ (960 LRS devices across a 64×30 sub-array), the total sub-array-level dynamic read energy is estimated at ~ 4.8 μ J under slow readout conditions and as low as ~ 192 pJ for fast readout, with an active power consumption of ~ 48 μ W. Note that this estimation precludes contributions from sneak path current due to the tendency of binary weights to terminate sneak paths. Although a full TOPS/W metric requires knowledge of system-level power requirements including peripheral circuitry (DACs, ADCs, multiplexers, lookup tables, etc.)⁴³, these values allow us to conservatively estimate the energy efficiency of the array itself based on the throughput estimations provided above, translating to a power efficiency of ~ 638 TOPS/W for the

crossbar array under peak conditions. It is extremely important to note that this high value represents a theoretical upper limit of efficiency and does not account for peripheral power draw. For example, previous investigations report that ADCs alone can account for 40–50% of the total power in similar systems⁴³. We therefore expect that accounting for even minimal peripheral load will reduce the overall system-level efficiency by at least one or two orders of magnitude.

4. The manuscript discusses the ability to program MoS₂ memtransistors via charge trapping in the Al₂O₃/HfO₂/Al₂O₃ dielectric stack. However, it does not provide sufficient analysis of the trade-off between retention stability and write disturbance—an important issue in charge-trapping-based memories. Since repeated write operations can introduce charge-trapping asymmetry and retention drift, I recommend that the authors investigate whether extended write cycling affects the stability of stored weights over long-term inference operations.

We agree with the reviewer that repeated write/erase cycling could have negative effects on weight stability (i.e., retention) over long periods of time and have now conducted additional experiments to investigate this phenomena.

We have added the following investigation/discussion of repeated write/erase (endurance) testing and subsequent retention testing to the supplemental information of the revised manuscript:

The endurance of the MoS₂ memtransistor architecture utilized in this work was also assessed. **Supplemental Information 15** (included here as **Figure R13** for reference) shows the results for a representative NVM cell over 500 program/erase cycles ($V_{\text{program}} = -10 \text{ V}$, $V_{\text{erase}} = +10 \text{ V}$, $t_{\text{pulse}} = 100 \text{ ms}$). Some degradation in the memory ratio between the ON-state and the OFF-state was seen during cycling; however, the read margin remains $>10^2$, indicating suitable endurance for NVM cell application. The retention of the same device over a period

Figure R13. Memtransistor Endurance Testing. ON-state and OFF-state of a representative device/cell taken over 500 program/erase cycles ($V_{\text{program}} = -10 \text{ V}$, $V_{\text{erase}} = +10 \text{ V}$, $t_{\text{pulse}} = 100 \text{ ms}$). Some degradation in the memory ratio between the ON-state and the OFF-state was seen during cycling; however, the read margin remained $>10^2$, indicating suitable endurance for NVM cell application.

of ~ 1 hour, read at a constant V_{DS} of 1 V, is shown in **Supplementary Information 16** (included here as **Figure R14** for reference) before and after endurance testing. While the read margin shows similar degradation to the cycling results, little-to-no change in long-term retention can be seen, indicating minimal write disturbance from repeated program/erase cycles.

5. While the manuscript provides an analysis of the device's long-term retention (>3 years), its assessment of cycling endurance is limited and lacks degradation analysis under repeated program/erase cycles. To strengthen the discussion, I recommend incorporating cycling endurance experiments and exploring the underlying degradation mechanisms.

We agree with the reviewer that assessment of cycling endurance is important for understanding the degradation mechanisms of memory technologies and have added these results to the revised manuscript. We have also added a statement regarding potential degradation mechanisms for the $\text{Al}_2\text{O}_3/\text{HfO}_2/\text{Al}_2\text{O}_3$ charge-trapping memory used in this work

Figure R14. Pre-/Post-Endurance Retention Testing. ON-state and OFF-state of a representative device/cell taken over ~1 hour before (left) and after (right) the endurance testing shown in **Figure R13**. For both cases, $V_{program} = -10$ V, $V_{erase} = +10$ V, and $t_{pulse} = 100$ ms. Some degradation in the read margin between the ON-state and the OFF-state was seen after cycling; however, the read margin remained $>10^2$ with little degradation in long-term retention, indicating minimal write disturbance from repeated program/erase cycles.

We have added the following investigation/discussion of endurance cycling to the supplemental material of the revised manuscript:

Degradation in the read margin following endurance testing may stem from several physical mechanisms. For instance, the relatively high defect density of Al_2O_3 , while beneficial for phenomena such as trap-assisted tunneling, can lead to a gradual accumulation of charge carriers in trap states over the course of many program/erase cycles, shifting the endurance characteristics⁶⁶. Additionally, high electric field stresses have previously been shown to induce irreversible charge trapping in HfO_2 -related defects, permanently damaging the charge-trapping gate stack⁶⁷. This may be addressed in future work by doping the HfO_2 charge-trapping layer with Al, reducing the concentration of oxygen vacancies and introducing reversible deep trap states that can reliably filled/depleted during memory operations to help achieve distinct conductance states⁶⁸. Rapid thermal annealing under in an oxygen environment has also been shown to heal oxygen vacancies in similar charge-trapping stacks, improving their resistance to degradation under high electric fields (i.e., endurance)⁶⁹.

6. This study demonstrates MNIST classification using a MoS₂ memtransistor crossbar array and introduces a heterosynaptic modulation method to enhance classification accuracy for highly confused digit pairs. However, even in simulation, the classification accuracy is only 62.8%, further dropping to 58.3% in hardware implementation. While heterosynaptic modulation improves specific cases (e.g., increasing the classification accuracy of “5&8” by 14.5%), its impact on overall classification performance remains limited. As thus, the benefits of using memtransistors for neuromorphic computing are not convincingly demonstrated. To better highlight the advantages of memtransistor-based computing, it is recommended that the authors first optimize the simulation to achieve at least 90% accuracy in simulation. Only after establishing a high-performance baseline should the hardware implementation be evaluated to clearly demonstrate the advantages of the memtransistor crossbar array.

The reviewer expresses a valid concern regarding the accuracy of the simulation discussed in the original manuscript and its ramifications for the subsequent hardware implementation. We agree that a high-performance baseline of at least 90% would more clearly demonstrate the efficacy of our memtransistor-based crossbar array architecture in implementing neural network operations on-chip. For the revised manuscript, we have prepared a new simulation based on 13×13-pixel images (up from 8×8-pixel) with significantly higher classification accuracy (88.1%) and successfully applied it to our hardware array to obtain a final classification accuracy of 85.6%.

We have adjusted our discussion on implementing MNIST classification operations in hardware using our MoS₂-memtransistor-based crossbar array architecture as follows in the revised manuscript:

Expanding our efforts to more closely investigate real-world ANN applications, we then performed single-layer neural network inference operations aiming to classify handwritten

Figure R15. Demonstration of MNIST Handwritten Digit Classification on a 2 kb Memtransistor Crossbar Array. a) Optical images of i) a 1.5×1.5 cm chip containing a 64×32 crossbar array comprising 2,048 MoS₂ memtransistors, ii) a zoomed-in image showing the full 2 kb array (scale bar of 100 μm), and iii) a further zoomed-in image showing nine constituent memtransistors (scale bar of 10 μm). b) Schematic showing preprocessing performed on training and test (inference) images taken from the Modified National Institute of Standards and Technology (MNIST) database. The original images (28×28 pixels) were downsampled to 13×13 pixels and binarized to fit into a 64×30 sub-array for this demonstration. Downsampled images were then converted to 169×1 input vectors and split into three sub-vectors for input to the array. A dataset comprising 10,000 resized/reshaped images were then used for training and weight assignment; for this demonstration, simulated weights were split between logic “0” and “1” and later converted to targeted conductance states for hardware implementation. c) Heatmap showing the distribution of simulated weights following training. A test dataset of 1,000 resized/reshaped MNIST images was fed to the simulated array for verification of network inference/classification. d) Confusion matrix showing the classification results for the inference check described in (c). An overall accuracy of 88.1% was achieved. e) Heatmap showing the distribution of conductance states assigned to the hardware array in respect to the simulated weight distribution shown in (c), with weights of “1” mapped to a conductance state of ~50 nS and weights of “0” mapped to the OFF-state conductance (a few pS). Cells marked NaN either display an open circuit or high gate leakage; the overall yield of devices remained high at ~92.2%. f) Hysteresis characteristics of the 1,770 working devices in the 64×30 sub-array extracted at V_{DS} = 1 V, which show low intrinsic device-to-device variation, memory windows >5 V, and read margins >10⁴. g) Histogram showing the distribution of the final conductance states (weights) represented in (e). h) Confusion matrix showing the classification results for hardware-based inference performed on the memtransistor array shown in (a) as per the conductance state (weight) distribution shown in (e). A test dataset comprising 1,000 resized/reshaped MNIST images was applied to the drain terminals of the array as voltage inputs (either 0 V or 1 V depending on the corresponding pixel value); output currents along corresponding columns/nodes were then individually registered and compared to all other outputs to determine the inferred digit for each case. An overall accuracy of 85.6% was registered.

digits from the MNIST datasets. An overview of this demonstration is shown in **Figure 3** (included here as **Figure R15** for reference). For this demonstration, a 64×32 (2,048 devices, 2 kb) crossbar array was prepared, with an optical image of the final chip containing the array being shown in **Figure R15a** along with zoomed-in optical images of the 2 kb array and constituent NVM cells. A simple binary classification scheme was then tested with the goal of classifying all ten digits (0-9) comprising the MNIST dataset. As schematically represented in **Figure R15b**, MNIST images (28×28 pixels) were first downsampled to 13×13 pixels using bicubic interpolation and transformed into 169×1 vectors to serve as inputs. To simplify the experimental demonstration, the greyscale pixels of the MNIST dataset were also binarized to either 0 or 1 during this process, making each image black and white; further discussion on this preprocessing, and on its effects on inference accuracy, can be found in the **Methods** section. Software-based training was then used to prepare a binary weight map from a training dataset consisting of 10,000 MNIST images, as shown in **Figure R15c**. Testing of the simulated single-layer network was then performed using a dataset consisting of 1,000 randomly selected binarized 13×13 images, curated such that each digit (0-9) was represented exactly 100 times to ensure fair assessment of the network’s ability to infer

different digits. The results from this simulation-based MNIST inference test are presented by the confusion matrix in **Figure R15d**. An overall accuracy of 88.1% was achieved; while digits such as “0”, “1”, and “8” demonstrated high classification accuracy, the overall accuracy was constrained by the networks limited ability to classify digits such as “5” which demonstrated a high degree of confusion with “8” and “3”. While non-ideal, this relatively low overall accuracy is to be expected due to the nature of the problem presented, i.e., single-layer inference across a large dataset consisting of ten highly compressed and binarized handwritten digits. Following simulation, the binary weights of the single-layer network were mapped to corresponding conductance states in a 64×30 sub-array of the 2 kb array shown in **Figure R15a**, with weights of “1” being mapped to a conductance state of ~50 nS and weights of “0” being mapped to the OFF-state conductance (a few pS). A map of the final conductance states across the sub-array is shown in **Figure R15e**; here, NVM cells marked NaN either display an open circuit or high gate leakage, both of which are believed to stem from damage to the corresponding horizontal drain line. Note that, despite the loss of five complete rows to these non-idealities, the overall yield of devices across the sub-array remained high at ~92.2%. Hysteresis characteristics and final conductance states (weights) for the working cells are also shown in **Figure R15f-g**, respectively, demonstrating good uniformity across the breadth of the array as well as a significant read margin of $>10^4$ between differently weighted cells. For the sake of clarity, devices demonstrating a measured conductance below the noise floor conductance of the characterization system ($\sim 10^{-12}$ S) are marked as being at the noise floor. For this hardware-based inference demonstration, the outputs from three columns of the sub-array correspond to a single digit utilized in the MNIST dataset, e.g., the first three columns in the sub-array correspond to the digit “0”, the next three columns correspond to “1”, etc. When an image (binary 169×1 vector split into three sub-vectors) is applied as voltage inputs (either 0 V or 1 V depending on the corresponding pixel value) to the drain terminal, the output currents from each column set (node) are individually registered. The summed output currents from each node are then compared to determine the inferred digit. Please note that, due to limitations with the experimental setup, summation of output currents was performed externally for the sake of

this demonstration. The results for this classification scheme are shown in **Figure R15h**. An overall accuracy of 85.6% was registered for the hardware demonstration; at only 2.5% lower than simulated, a similar simulation-to-hardware gap to previous MNIST classification explorations⁴⁸, this indicates that the MoS₂-based crossbar arrays developed and demonstrated in this work can successfully implement inference operations with a high degree of similarity to simulations, thus indicating potential as neuromorphic accelerators. Deviations from the ideal case demonstrated in **Figure R15d** can be attributed to device-to-device variation and non-ideal yield and can be optimized for future work through array-level compensation schemes⁴⁹⁻⁵¹ and optimization of the fabrication and MoS₂ growth processes.

7. The manuscript mentions initializing all devices to the OFF state and using a small negative bias to suppress sneak path currents. A more detailed quantitative analysis is needed to evaluate the sneak leakage current across large arrays.

The reviewer brings up a good point regarding the influence of sneak path leakage current across large arrays such as that demonstrated in this manuscript. We agree that sneak path current flowing along partially-selected cells can strongly influence the results of multiply-accumulate (MAC) operations along each column/node. Such influences have already seen substantial investigation elsewhere^{12,19,39,43,65,70}. However, we feel that it is important to point out that the approach discussed in this work addresses this issue through a combination of two key strategies: (1) initializing all devices to the OFF state and (2) using dynamic gate modulation to introduce analog behavior during readout, without storing multiple conductance levels. First, all memtransistor devices are programmed into a high-resistance OFF state prior to operation. This effectively suppresses persistent low-resistance paths across the array, limiting the number of conductive routes through which sneak currents can propagate⁴⁷. In contrast to analog-mapped arrays where a large portion of the devices may hold intermediate conductance values and collectively contribute to leakage, our binary weight mapping ensures that a significant portion of devices remain in a non-conductive state during inference. Second, the use of dynamic gate modulation during readout allows

us to emulate analog weight scaling by adjusting the channel conductance locally and transiently, without modifying the stored resistance. Because the gate terminal provides independent control, this modulation can be applied selectively and only during inference. This decoupling enables the array to operate primarily in a binary regime, where sneak path exposure is minimized, while still allowing higher-resolution output when needed. Together, this combination of sparse binary initialization and transient, gate-based analog modulation provides effective sneak path mitigation without requiring selector devices or static multilevel programming. This approach allows us to preserve array-level efficiency and scalability while achieving inference behavior comparable to analog-weight networks.

Additionally, please note that we do not claim that this approach eliminates sneak paths entirely. Even devices in the OFF state possess finite resistance, and under large array sizes, half-selected paths can contribute measurable leakage. However, with OFF-state conductances only reaching a few pS and read voltages maintained at 1 V, the resulting leakage currents are typically below 10 pA, well within the tolerances of common sensing circuits. Furthermore, while not utilized in this study, during inference under scenarios where the cumulative sneak path current is appreciable, the gates of unselected rows may be held at voltages that maintain channel depletion, further suppressing unintended conduction.

We have adjusted the following statement on array initialization in the revised manuscript to further elaborate on sneak paths and their influence in analog/binary networks:

While straightforward and effective, this scheme was found to affect all devices in a given row, making it ill-suited to programming distinct conductance states (weights) in individual memtransistors for logic operations; however, it may still be utilized effectively for setting each NVM in an array to a low conductance state (LCS) ahead of weight assignment through the application of large positive gate biases (V_{erase}) in an initialization step. This primarily serves to standardize conductance states between different logic operations performed during array testing. This effectively suppresses persistent low-resistance paths across the

array, limiting the number of conductive routes through which sneak currents can propagate⁴⁷. Please note that this approach does not eliminate sneak paths entirely. Even devices in the OFF state possess finite resistance, and under large array sizes, half-selected paths can contribute measurable leakage. However, with OFF-state conductances only reaching a few pS and read voltages maintained at 1 V, the resulting leakage currents are typically below 10 pA, well within the tolerances of common sensing circuits. For applications wherein the weights of a given array are held constant, this step may not be required.

8. The study currently employs binary ON/OFF weight storage, yet MoS₂ memtransistors inherently support multi-level conductance states. Since increasing weight resolution can significantly improve inference accuracy, the authors should discuss whether multi-level programming is feasible within this architecture and how it might impact training convergence and classification performance.

The reviewer raises a good point regarding analog (multi-level) versus binary (two-level) operation of MoS₂ memtransistors. We have previously demonstrated the operation of multiple conductance states in MoS₂ memtransistors¹⁶⁻¹⁸ and agree that increased weight resolution would lead to higher inference accuracy at the same array sizes. This notion is supported by the results presented in **Figure R16**, which compares inference accuracy for MNIST simulations based on binarized 13×13-pixel images similar to that presented in the revised manuscript with that of non-binarized (greyscale) images with varying numbers of distinct weights. The highest achievable accuracy for binarized (2-state) inference is ~85% while analog inference gives accuracies of ~87%, ~87.3%, ~87.4%, and ~87.4% for 4, 6, 8, and 10 weight states, respectively. However, while these results serve to demonstrate the advantages offered by analog weight operation, the overall increase in accuracy is minimal

Figure R16. Inference Accuracy Across Weight States. Training and test accuracy as a function of the number of weight states for MNIST simulations based on binarized 13×13 -pixel images.

(~2.4%) even at 10 weight states. Additionally, it is important to also consider realistic use cases. It is already widely recognized that fabrication imperfections can lead to mismatch between otherwise identical devices in analog arrays, with the severity of the mismatch increasing as the number of states increases due to the ratio between states becoming smaller. When these arrays are then used for computation, this mismatch leads to correspondingly severe inference errors unless mitigated through advanced compensation schemes such as per-device calibration, iterative parameter storage, chip-in-loop training, etc., each of which add to the overall energy/area requirements of the system. Thus, while analog arrays may deliver higher inference accuracy, their additional system requirements limit their potential for decentralized edge applications. Conversely, digital (binary) devices largely avoid mismatch issues due to their intrinsically high ON/OFF ratios while also benefiting from reduced computational complexity and simplified memory/routing requirements. This makes binary neural networks (BNNs) an attractive option for “inference-only” architectures, wherein all training occurs before deployment with limited subsequent fine-tuning from system processors; such use cases match well with the demands of compact, low-resource edge applications, making them a relevant line of study. As the potential of MoS₂ for back-end-of-line (BEOL) and/or edge applications has already been thoroughly studied/demonstrated^{5,14,61,71,72}, and is indeed the primary field of interest for

TMD-based devices over front-end-of-line (FEOL) or high-performance computing applications, our work therefore focuses on BNN implementation with dynamic weight tuning through the gate terminal.

9. Literature such as Sangvan et al., Nature, 2018, 554, 500–504, and Tan et al., ACS Nano, 2024, 18(21), 13652–13661, presents another type of MoS₂ memtransistor that leverages the dynamic modulation of Schottky barrier height to induce resistive switching behavior. What is the advantage of the memtransistor in this work compared to these studies? Discussing these works in the introduction would help establish a more comprehensive theoretical framework and highlight the relationship between existing progress and the present work.

The author posits a good question regarding potential contributions from defect (sulfur vacancy) migration, which has been previously shown to modulate Schottky barrier height at the semiconductor/contact interface, in the observed resistive switching of our memtransistors based on CVD-grown MoS₂. We agree that numerous extant 2D-material-based crossbar demonstrations^{2,11-14} program (write) devices through application of a drain bias pulse to prompt defect migration across the semiconducting channel, thereby changing the threshold (conductance) of the device; depending on the biases required, with lower biases always preferable for greater energy efficiency, this can lead to destructive read operations similar to those that affect two-terminal memristor technologies^{11,12}. Additionally, defect engineering of the channel, e.g., through low-energy Ar plasma treatment¹⁵, may be required to provide a sufficiently high concentration of defects for appreciable memory effects to be seen at low voltages, thus complicating the fabrication process. Conversely, for our demonstration, programming is performed through the application of bias pulses to the gate terminal to promote charge-trapping in the dielectric stack, thereby shifting the threshold by screening the electric field across the channel in a manner analogous to traditional FLASH memories. As the drain terminal is decoupled from the programming mechanism, this minimizes the risk of destructive read events.

Furthermore, unlike defect-migration-based memories, charge-trapping memories do not require any preprocessing (i.e., defect engineering) of the channel material, relying solely on the as-fabricated dielectric stack for memory operation.

We have adjusted the following statement on charge trapping in the revised manuscript to better differentiate the mechanisms utilized in this work from vacancy migration:

The memory capabilities/operations of the $\text{Al}_2\text{O}_3/\text{HfO}_2/\text{Al}_2\text{O}_3$ gate dielectric stack incorporated into each MoS_2 memtransistor (NVM cell) can be easily understood through the band diagrams shown in **Supplementary Information 5** (included here as **Figure R17** for reference). As has been studied elsewhere¹⁶⁻¹⁸, the NVM capabilities of each device/cell are governed by the trapping/detrapping of charge carriers in the middle HfO_2 charge-trapping layer when bias pulses of sufficient magnitude are applied to the back-gate in program/erase (write) operations. These trapped charges screen the electric field across the MoS_2 channel, shifting the threshold voltage (V_{th}) of the device and allowing for the realization of distinct conductance (memory) states at a given read voltage (V_{read}). The polarity of the applied pulse determines which charge carriers are trapped/detrapped, with holes (electrons) being trapped when negative (positive) pulses are applied and vice versa, while the pulse time and magnitude determine the change in carrier concentration in the charge-trapping layer (i.e., the size of the V_{th} shift, or memory window). For the purposes of this work, the memory window (ΔV_{th}) is defined as the difference in V_{th} , taken at a constant current level, between programmed/erased states. Note that many extant 2D-material-based crossbar demonstrations^{2,11-14} program (write) devices through application of a drain bias pulse to prompt defect migration across the semiconducting channel, thereby changing the threshold (conductance) of the device; depending on the biases required, with lower biases always preferable for greater energy efficiency, this can lead to destructive read operations similar to those that affect two-terminal memristor technologies^{11,12}. Additionally, defect engineering of the channel, e.g., through low-energy Ar plasma treatment¹⁵, may be required to provide a sufficiently high concentration of defects for appreciable memory effects to be

Figure R17. Band diagram of $\text{Al}_2\text{O}_3/\text{HfO}_2/\text{Al}_2\text{O}_3$ gate dielectric stack. The $\text{Al}_2\text{O}_3/\text{HfO}_2/\text{Al}_2\text{O}_3$ gate dielectric stack enables non-volatile memory in 2D MoS_2 memtransistors by allowing the trapping/detrapping of charge carriers in the HfO_2 (charge-trapping) layer when bias pulses of sufficient magnitude are applied to the back-gate. The polarity of the pulse determines which charge carriers are trapped/detrapped, with holes (electrons) being trapped when negative (positive) pulses are applied, and vice versa. These trapped charges screen the electric field across the MoS_2 channel, changing the conductance of the device and allowing for the realization of distinct conductance (memory) states.

seen at low voltages, thus complicating the fabrication process. Conversely, for our demonstration, programming is performed through the application of bias pulses to the gate terminal to promote charge-trapping in the dielectric stack, thereby shifting the threshold by screening the electric field across the channel in a manner analogous to traditional FLASH memories. As the drain terminal is decoupled from the programming mechanism, this minimizes the risk of destructive read events. Furthermore, unlike defect-migration-based memories, charge-trapping memories do not require any preprocessing (i.e., defect engineering) of the channel material, relying solely on the as-fabricated dielectric stack for memory operation.

10. The time axis in Fig. S12 is labelled with the unit "t", which may cause confusion as it

does not explicitly indicate the standard unit of measurement (e.g., seconds, milliseconds). To ensure clarity and consistency with the rest of the manuscript, I recommend specifying the correct time unit explicitly.

We thank the reviewer for bringing this discrepancy to our attention.

We have corrected all units in the revised manuscript and supplemental material.

Reviewer #3 (Remarks to the Author):

The authors reported here a large-scale crossbar arrays based on 2048 MoS₂ memtransistors for artificial neuron networks. The MoS₂ memtransistors utilize the charge-trapping in Al₂O₃/HfO₂/Al₂O₃ back gate stacks and work as NVM. The arrays demonstrate the ability to resolve inference ambiguities through gate modulation without the need for costly retraining or reprogramming, and image classification of handwritten digits from MNIST database. However, the benefit of these MoS₂ memtransistor arrays based artificial neuron networks is not clear, and how it compares with two terminal memristors, which already simultaneously deliver high energy efficiency, versatility to support diverse models and software-comparable accuracy at chip level. Some key performance of the MoS₂ memtransistors NVM needs to be shown.

We thank the reviewer for their feedback and have attempted to address their comments to the best of our ability, as discussed below.

-The authors claimed the device could achieve retention exceeding 3 years, but there isn't any retention data to support this.

Figure R18. Long Term Memory Testing. To verify the accuracy of the fits shown in **Figure R3**, the ON-state and OFF-state of a representative device/cell were read over periods of ~ 10 minutes (left column) and ~ 1 hour (middle column) and simple power law fits were calculated. These fits were then plotted (right column) over 10^8 seconds (~ 3.5 years), during which time they remained above the designated OFF-state current (~ 2 pA); notably, the 10 minute and 1-hour fits were in close agreement throughout the entire time span, indicating that the 10 minutes fit discussed above are relatively accurate. Additionally, no degradation in the OFF-state was noted even for the longer retention tests, indicating that the change in ON/OFF ratio over time will predominantly depend on the change in ON-state conductance.

We thank the reviewer for their question regarding the long-term retention of our MoS₂ memtransistors. The data pertaining to our extrapolation of a retention exceeding 3 years is presented in **Supplementary Information 13** of the original manuscript (included here as **Figure R18** for reference). Upon review of the main text, we noted that the location of this data was not explicitly mentioned in the original draft, so we apologize for any confusion.

We have amended the following statement to acknowledge the long-term retention data in the revised manuscript:

The memory performance of devices in this array was also assessed separately from the smaller arrays detailed above to accurately determine what effects, if any, scaling has on our program/erase capabilities, with the results being presented in **Supplemental Figure 12-13**. From these results, we note minimal spatial influence on hysteresis (i.e., read margin and memory window) or retention. Furthermore, a fit of ~ 1 hour retention measurements indicates that long-term non-volatile retention exceeds three years, in part due to the

Figure R19. Memtransistor Endurance Testing. ON-state and OFF-state of a representative device/cell taken over 500 program/erase cycles ($V_{\text{program}} = -10 \text{ V}$, $V_{\text{erase}} = +10 \text{ V}$, $t_{\text{pulse}} = 100 \text{ ms}$). Some degradation in the memory ratio between the ON-state and the OFF-state was seen during cycling; however, the read margin remained $>10^2$, indicating suitable endurance for NVM cell application.

intrinsically-OFF nature of the devices meaning that change in read margin over time will predominantly depend on the change in ON-state conductance.

-Please exhibit endurance of the devices.

We agree with the reviewer that assessment of cycling endurance is important for understanding the degradation mechanisms of memory technologies and have added these results to the revised manuscript.

We have added the following investigation/discussion of endurance cycling to the supplemental material of the revised manuscript:

The endurance of the MoS₂ memtransistor architecture utilized in this work was also assessed. **Supplemental Information 15** (included here as **Figure R19** for reference) shows

the results for a representative NVM cell over 500 program/erase cycles ($V_{\text{program}} = -10 \text{ V}$, $V_{\text{erase}} = +12 \text{ V}$, $t_{\text{pulse}} = 100 \text{ ms}$). Some degradation in the memory ratio between the ON-state and the OFF-state was seen during cycling; however, the read margin remains $>10^2$, indicating suitable endurance for NVM cell application.

-Please explain how the switching energies of $\sim 0.2 \text{ fJ}$ is calculated.

As stated in the main manuscript, all write energies were estimated using the equation $\text{Energy} = \text{Time} \times \text{Current} \times \text{Voltage}$, which is commonly used to estimate switching energy for NVMs⁷³. Here, 'Time' refers to the pulse time (write time) of each respective write event, 'Current' refers to the current flowing through the array (predominantly gate-to-source current (I_{GS}) through the targeted cell as $V_{\text{DS}} = 0 \text{ V}$ during programming), and 'Voltage' refers to the magnitude of the gate bias pulse applied. A low switching energy of $\sim 0.2 \text{ fJ}$ was estimated for our minimum confirmed pulse time of $1 \mu\text{s}$, with a magnitude of 10 V and a registered current of $\sim 20 \text{ pA}$.

-The recognition accuracy for MNIST classification is very low compared to literatures. This should be improved.

The reviewer expresses a valid concern regarding the accuracy of the simulation discussed in the original manuscript and its ramifications for the subsequent hardware implementation. We acknowledge that a high-performance baseline of $\sim 90\%$ would more clearly demonstrate the efficacy of our memtransistor-based crossbar array architecture in implementing neural network operations on-chip. For the revised manuscript, we have prepared a new simulation based on 13×13 -pixel images (up from 8×8 -pixel) with significantly higher classification accuracy (88.1%) and successfully applied it to our hardware array to obtain a final classification accuracy of 85.6% .

We have adjusted our discussion on implementing MNIST classification operations in hardware using our MoS₂-memtransistor-based crossbar array architecture as follows in the revised manuscript:

Expanding our efforts to more closely investigate real-world ANN applications, we then performed single-layer neural network inference operations aiming to classify handwritten digits from the MNIST datasets. An overview of this demonstration is shown in **Figure 3** (included here as **Figure R20** for reference). For this demonstration, a 64×32 (2,048 devices, 2 kb) crossbar array was prepared, with an optical image of the final chip containing the array being shown in **Figure R20a** along with zoomed-in optical images of the 2 kb array and constituent NVM cells. A simple binary classification scheme was then tested with the goal of classifying all ten digits (0-9) comprising the MNIST dataset. As schematically represented in **Figure R20b**, MNIST images (28×28 pixels) were first downsampled to 13×13 pixels using bicubic interpolation and transformed into 169×1 vectors to serve as inputs. To simplify the experimental demonstration, the greyscale pixels of the MNIST dataset were also binarized to either 0 or 1 during this process, making each image black and white; further discussion on this preprocessing, and on its effects on inference accuracy, can be found in the **Methods** section. Software-based training was then used to prepare a binary weight map from a training dataset consisting of 10,000 MNIST images, as shown in **Figure R20c**. Testing of the simulated single-layer network was then performed using a dataset consisting of 1,000 randomly selected binarized 13×13 images, curated such that each digit (0-9) was represented exactly 100 times to ensure fair assessment of the network's ability to infer different digits. The results from this simulation-based MNIST inference test are presented by the confusion matrix in **Figure R20d**. An overall accuracy of 88.1% was achieved; while digits such as “0”, “1”, and “8” demonstrated high classification accuracy, the overall accuracy was constrained by the network's limited ability to classify digits such as “5” which demonstrated a high degree of confusion with “8” and “3”. While non-ideal, this relatively low overall accuracy is to be expected due to the nature of the problem presented, i.e., single-layer inference across a large dataset consisting of ten highly

Figure R20. Demonstration of MNIST Handwritten Digit Classification on a 2 kb Memtransistor Crossbar Array. a) Optical images of i) a 1.5×1.5 cm chip containing a 64×32 crossbar array comprising 2,048 MoS₂ memtransistors, ii) a zoomed-in image showing the full 2 kb array (scale bar of 100 μm), and iii) a further zoomed-in image showing nine constituent memtransistors (scale bar of 10 μm). b) Schematic showing preprocessing performed on training and test (inference) images taken from the Modified National Institute of Standards and Technology (MNIST) database. The original images (28×28 pixels) were downsampled to 13×13 pixels and binarized to fit into a 64×30 sub-array for this demonstration. Downsampled images were then converted to 169×1 input vectors and split into three sub-vectors for input to the array. A dataset comprising 10,000 resized/reshaped images were then used for training and weight assignment; for this demonstration, simulated weights were split between logic “0” and “1” and later converted to targeted conductance states for hardware implementation. c) Heatmap showing the distribution of simulated weights following training. A test dataset of 1,000 resized/reshaped MNIST images was fed to the simulated array for verification of network inference/classification. d) Confusion matrix showing the classification results for the inference check described in (c). An overall accuracy of 88.1% was achieved. e) Heatmap showing the distribution of conductance states assigned to the hardware array in respect to the simulated weight distribution shown in (c), with weights of “1” mapped to a conductance state of ~50 nS and weights of “0” mapped to the OFF-state conductance (a few pS). Cells marked NaN either display an open circuit or high gate leakage; the overall yield of devices remained high at ~92.2%. f) Hysteresis characteristics of the 1,770 working devices in the 64×30 sub-array extracted at V_{DS} = 1 V, which show low intrinsic device-to-device variation, memory windows >5 V, and read margins >10⁴. g) Histogram showing the distribution of the final conductance states (weights) represented in (e). h) Confusion matrix showing the classification results for hardware-based inference performed on the memtransistor array shown in (a) as per the conductance state (weight) distribution shown in (e). A test dataset comprising 1,000 resized/reshaped MNIST images was applied to the drain terminals of the array as voltage inputs (either 0 V or 1 V depending on the corresponding pixel value); output currents along corresponding columns/nodes were then individually registered and compared to all other outputs to determine the inferred digit for each case. An overall accuracy of 85.6% was registered.

compressed and binarized handwritten digits. Following simulation, the binary weights of the single-layer network were mapped to corresponding conductance states in a 64×30 sub-array of the 2 kb array shown in **Figure R20a**, with weights of “1” being mapped to a conductance state of ~50 nS and weights of “0” being mapped to the OFF-state conductance (a few pS). A map of the final conductance states across the sub-array is shown in **Figure R20e**; here, NVM cells marked NaN either display an open circuit or high gate leakage, both of which are believed to stem from damage to the corresponding horizontal drain line. Note that, despite the loss of five complete rows to these non-idealities, the overall yield of devices across the sub-array remained high at ~92.2%. Hysteresis characteristics and final conductance states (weights) for the working cells are also shown in **Figure R20f-g**, respectively, demonstrating good uniformity across the breadth of the array as well as a significant read margin of >10⁴ between differently weighted cells. For the sake of clarity, devices demonstrating a measured conductance below the noise floor conductance of the characterization system (~10⁻¹² S) are marked as being at the noise floor. For this hardware-based inference demonstration, the outputs from three columns of the sub-array correspond to a single digit utilized in the MNIST dataset, e.g., the first three

columns in the sub-array correspond to the digit “0”, the next three columns correspond to “1”, etc. When an image (binary 169×1 vector split into three sub-vectors) is applied as voltage inputs (either 0 V or 1 V depending on the corresponding pixel value) to the drain terminal, the output currents from each column set (node) are individually registered. The summed output currents from each node are then compared to determine the inferred digit. Please note that, due to limitations with the experimental setup, summation of output currents was performed externally for the sake of this demonstration. The results for this classification scheme are shown in **Figure R20h**. An overall accuracy of 85.6% was registered for the hardware demonstration; at only 2.5% lower than simulated, a similar simulation-to-hardware gap to previous MNIST classification explorations⁴⁸, this indicates that the MoS₂-based crossbar arrays developed and demonstrated in this work can successfully implement inference operations with a high degree of similarity to simulations, thus indicating potential as neuromorphic accelerators. Deviations from the ideal case demonstrated in **Figure R20d** can be attributed to device-to-device variation and non-ideal yield and can be optimized for future work through array-level compensation schemes⁴⁹⁻⁵¹ and optimization of the fabrication and MoS₂ growth processes.

-It seems that the array based on 64×32 MoS₂ memtransistors is fabricated, but isn't tested at array level. Please indicate if the array could be tested at array level. If yes, please explain the test in details and show the related experimental data.

The reviewer makes a good point regarding how many devices from the 64×32 (2 kb) MoS₂-memtransistor array have been thoroughly characterized. The original manuscript only concerns the use of a 64×10 sub-array for MNIST inference testing, with only the characterization results for the corresponding devices being included; however, for the revised manuscript, a much larger portion of the array (64×30) is used for inference implementation due to the use of larger (13×13 -pixel) images. To support these results,

characterization of the constituent working devices is now presented in **Figure 3** of the revised manuscript (included here as **Figure R20** for your reference).

Rebuttal References

- 1 Marega, G. M. *et al.* A large-scale integrated vector–matrix multiplication processor based on monolayer molybdenum disulfide memories. *Nature Electronics* **6**, 991-998, doi:10.1038/s41928-023-01064-1 (2023).
- 2 Feng, X. *et al.* Self-Selective Multi-Terminal Memtransistor Crossbar Array for In-Memory Computing. *ACS Nano* **15**, 1764-1774, doi:10.1021/acsnano.0c09441 (2021).
- 3 Lee, H.-S. *et al.* Dual-Gated MoS₂ Memtransistor Crossbar Array. *Advanced Functional Materials* **30**, doi:10.1002/adfm.202003683 (2020).
- 4 Fu, S. *et al.* Two-Terminal MoS₂ Memristor and the Homogeneous Integration with a MoS₂ Transistor for Neural Networks. *Nano Letters* **23**, 5869-5876, doi:10.1021/acs.nanolett.2c05007 (2023).
- 5 Kumar, P. *et al.* Hybrid architecture based on two-dimensional memristor crossbar array and CMOS integrated circuit for edge computing. *npj 2D Materials and Applications* **6**, doi:10.1038/s41699-021-00284-3 (2022).
- 6 Bala, A., Sen, A., Shim, J., Gandla, S. & Kim, S. Back-End-of-Line Compatible Large-Area Molybdenum Disulfide Grown on Flexible Substrate: Enabling High-Performance Low-Power Memristor Applications. *ACS Nano* **17**, 13784–13791, doi:10.1021/acsnano.3c03407 (2023).
- 7 Naqi, M. *et al.* Multilevel artificial electronic synaptic device of direct grown robust MoS₂ based memristor array for in-memory deep neural network. *npj 2D Materials and Applications* **6**, doi:10.1038/s41699-022-00325-5 (2022).
- 8 Tang, B. *et al.* Wafer-scale solution-processed 2D material analog resistive memory array for memory-based computing. *Nature Communications* **13**, doi:10.1038/s41467-022-30519-w (2022).
- 9 Chen, S. *et al.* Wafer-scale integration of two-dimensional materials in high-density memristive crossbar arrays for artificial neural networks. *Nature Electronics* **3**, 638-645, doi:10.1038/s41928-020-00473-w (2020).
- 10 Leong, J. F. *et al.* N-P Reconfigurable Dual-Mode Memtransistors for Compact Bio-Inspired Feature Extractor with Inhibitory-Excitatory Spiking Capability. *Advanced Functional Materials* **33**, doi:10.1002/adfm.202302949 (2023).
- 11 Sivan, M. *et al.* Physical Insights into Vacancy-Based Memtransistors: Toward Power Efficiency, Reliable Operation, and Scalability. *ACS Nano* **16**, 14308–14322, doi:10.1021/acsnano.2c04504 (2022).
- 12 Deng, W. *et al.* Two-dimensional materials based memtransistors: Integration strategies, switching mechanisms and advanced characterizations. *Nano Energy* **128**, doi:10.1016/j.nanoen.2024.109861 (2024).
- 13 Sangwan, V. K. *et al.* Multi-terminal memtransistors from polycrystalline monolayer molybdenum disulfide. *Nature* **554**, 500-504, doi:10.1038/nature25747 (2018).
- 14 Tan, T. *et al.* Integration of MoS₂ Memtransistor Devices and Analogue Circuits for Sensor Fusion in Autonomous Vehicle Target Localization. *ACS Nano* **18**, 13652–13661, doi:10.1021/acsnano.4c00456 (2024).
- 15 Rajput, M. *et al.* Defect-engineered monolayer MoS₂ with enhanced memristive and synaptic functionality for neuromorphic computing. *Communications Materials* **5**, doi:10.1038/s43246-024-00632-y (2024).
- 16 Zhang, E. *et al.* Tunable Charge-Trap Memory Based on Few-Layer MoS₂. *ACS Nano* **9**, 612-619, doi:10.1021/nn5059419 (2015).
- 17 Hou, X. *et al.* Operation mode switchable charge-trap memory based on few-layer MoS₂. *Semiconductor Science and Technology* **33**, doi:10.1088/1361-6641/aaa79e (2018).

- 18 Schranghamer, T. F. *et al.* Radiation Resilient Two-Dimensional Electronics. *ACS Applied Materials & Interfaces* **15**, 26946–26959, doi:10.1021/acsami.3c02406 (2023).
- 19 Jeon, K. *et al.* Purely self-rectifying memristor-based passive crossbar array for artificial neural network accelerators. *Nature Communications* **15**, doi:10.1038/s41467-023-44620-1 (2024).
- 20 Sivan, M. *et al.* All WSe₂ 1T1R resistive RAM cell for future monolithic 3D embedded memory integration. *Nature Communications* **10**, doi:10.1038/s41467-019-13176-4 (2019).
- 21 Hwang, S., Yu, J., Song, M. S., Hwang, H. & Kim, H. Memcapacitor Crossbar Array with Charge Trap NAND Flash Structure for Neuromorphic Computing. *Advanced Science* **10**, doi:10.1002/advs.202303817 (2023).
- 22 Bavandpour, M., Sahay, S., Mahmoodi, M. R. & Strukov, D. B. 3D-aCortex: an ultra-compact energy-efficient neurocomputing platform based on commercial 3D-NAND flash memories. *Neuromorphic Computing and Engineering* **1**, doi:10.1088/2634-4386/ac0775 (2021).
- 23 Guo, X. *et al.* in *IEEE Custom Integrated Circuits Conference* 1-4 (Austin, TX, USA, 2017).
- 24 Hur, J. *et al.* Nonvolatile Capacitive Crossbar Array for In-Memory Computing. *Advanced Intelligent Systems* **4**, doi:10.1002/aisy.202100258 (2022).
- 25 Soliman, T. *et al.* First demonstration of in-memory computing crossbar using multi-level Cell FeFET. *Nature Communications* **14**, doi:10.1038/s41467-023-42110-y (2023).
- 26 Kim, I.-J. & Lee, J.-S. Ferroelectric Transistors for Memory and Neuromorphic Device Applications. *Advanced Materials* **35**, doi:10.1002/adma.202206864 (2022).
- 27 Marega, G. M. *et al.* Logic-in-memory based on an atomically thin semiconductor. *Nature* **587**, 72-77, doi:10.1038/s41586-020-2861-0 (2020).
- 28 Marega, G. M. *et al.* Low-Power Artificial Neural Network Perceptron Based on Monolayer MoS₂. *ACS Nano* **16**, 3684–3694, doi:10.1021/acsnano.1c07065 (2022).
- 29 Huh, W. *et al.* Synaptic Barristor Based on Phase-Engineered 2D Heterostructures. *Advanced Materials* **30**, doi:10.1002/adma.201801447 (2018).
- 30 He, C. *et al.* Artificial Synapse Based on van der Waals Heterostructures with Tunable Synaptic Functions for Neuromorphic Computing. *ACS Applied Materials & Interfaces* **12**, 11945-11954, doi:10.1021/acsami.9b21747 (2020).
- 31 Schranghamer, T. F., Oberoi, A. & Das, S. Graphene memristive synapses for high precision neuromorphic computing. *Nature Communications* **11**, doi:10.1038/s41467-020-19203-z (2020).
- 32 Lynch, G. S., Dunwiddie, T. & Gribkoff, V. Heterosynaptic depression: a postsynaptic correlate of long-term potentiation. *Nature* **266**, 737-739, doi:10.1038/266737a0 (1977).
- 33 Bailey, C. H., Giustetto, M., Huang, Y.-Y., Hawkins, R. D. & Kandel, E. R. Is Heterosynaptic modulation essential for stabilizing hebbian plasticity and memory. *Nature Reviews Neuroscience* **1**, 11-20, doi:10.1038/35036191 (2000).
- 34 Lecun, Y., Bottou, L., Bengio, Y. & Haffner, P. Gradient-based learning applied to document recognition. *Proceedings of the IEEE* **86**, 2278-2324, doi:10.1109/5.726791 (1998).
- 35 Lin, Y. *et al.* A High-Speed and High-Efficiency Diverse Error Margin Write-Verify Scheme for an RRAM-Based Neuromorphic Hardware Accelerator. *IEEE Transactions on Circuits and Systems II: Express Briefs* **70**, 1366-1370, doi:10.1109/TCSII.2022.3224470 (2023).
- 36 Perez, E., Mahadevaiah, M. K., Quesada, E. P.-B. & Wenger, C. Variability and Energy Consumption Tradeoffs in Multilevel Programming of RRAM Arrays. *IEEE Transactions on Electron Devices* **68**, 2693-2698, doi:10.1109/TED.2021.3072868 (2021).
- 37 Guo, M.-H. *et al.* Attention mechanisms in computer vision: A survey. *Computational Visual Media* **8**, 331-368, doi:10.1007/s41095-022-0271-y (2022).

- 38 Wei, Z., Zhao, B. & Su, J. Event-Driven Computation Offloading in IoT With Edge Computing. *IEEE Transactions on Wireless Communications* **21**, 6847-6860, doi:10.1109/TWC.2022.3152573 (2022).
- 39 Parmar, V., Kingra, S. K., Negi, S. & Suri, M. Analysis of VMM computation strategies to implement BNN applications on RRAM arrays. *APL Machine Learning* **1**, doi:10.1063/5.0139583 (2023).
- 40 Sheridan, P. M. *et al.* Sparse coding with memristor networks. *Nature Nanotechnology* **12**, 784-789, doi:10.1038/nnano.2017.83 (2017).
- 41 Anwar, S., Hwang, K. & Sung, W. Structured Pruning of Deep Convolutional Neural Networks. *ACM Journal on Emerging Technologies in Computing Systems* **13**, 1-18, doi:10.1145/3005348 (2017).
- 42 Mittal, S. A Survey of ReRAM-Based Architectures for Processing-In-Memory and Neural Networks. *Machine Learning and Knowledge Extraction* **1**, 75-114, doi:10.3390/make1010005 (2018).
- 43 Xiao, T. P., Bennett, C. H., Feinberg, B., Agarwal, S. & Marinella, M. J. Analog architectures for neural network acceleration based on non-volatile memory. *Applied Physics Reviews* **7**, doi:10.1063/1.5143815 (2020).
- 44 Zhao, R. *et al.* A framework for the general design and computation of hybrid neural networks. *Nature Communications* **13**, doi:10.1038/s41467-022-30964-7 (2022).
- 45 Wang, E. *et al.* Enabling Binary Neural Network Training on the Edge. *ACM Transactions on Embedded Computing Systems* **22**, doi:10.1145/3626100 (2023).
- 46 Kim, Y. *et al.* Memristor crossbar array for binarized neural networks. *AIP Advances* **9**, doi:10.1063/1.5092177 (2019).
- 47 Zhao, Y., Wang, Y., Wang, R., Rong, Y. & Jiang, X. A Highly Robust Binary Neural Network Inference Accelerator Based on Binary Memristors. *Electronics* **10**, doi:10.3390/electronics10212600 (2021).
- 48 Li, C. *et al.* Efficient and self-adaptive in-situ learning in multilayer memristor neural networks. *Nature Communications* **9**, doi:10.1038/s41467-018-04484-2 (2018).
- 49 Jain, S. & Raghunathan, A. CxDNN: Hardware-software Compensation Methods for Deep Neural Networks on Resistive Crossbar Systems. *ACM Transactions on Embedded Computing Systems* **18**, 1-23, doi:10.1145/3362035 (2019).
- 50 Li, B., Xia, L., Gu, P., Wang, Y. & Yang, H. in *Design Automation Conference*. (Association for Computing Machinery).
- 51 Zidan, M. A., Omran, H., Sultan, A., Fahmy, H. A. H. & Salama, K. N. Compensated Readout for High-Density MOS-Gated Memristor Crossbar Array. *IEEE Transactions on Nanotechnology* **14**, 3-6, doi:10.1109/TNANO.2014.2363352 (2014).
- 52 Wan, Q., Sharbati, M. T., Erickson, J. R., Du, Y. & Xiong, F. Emerging Artificial Synaptic Devices for Neuromorphic Computing. *Advanced Materials Technologies* **4**, doi:10.1002/admt.201900037 (2019).
- 53 Dai, S. *et al.* Recent Advances in Transistor-Based Artificial Synapses. *Advanced Functional Materials* **29**, doi:10.1002/adfm.201903700 (2019).
- 54 Chakraborty, I. *et al.* Resistive Crossbars as Approximate Hardware Building Blocks for Machine Learning: Opportunities and Challenges. *Proceedings of the IEEE* **108**, 2276-2310, doi:10.1109/JPROC.2020.3003007 (2020).
- 55 Jayachandran, D. *et al.* Three-dimensional integration of two-dimensional field-effect transistors. *Nature* **625**, 276-281, doi:10.1038/s41586-023-06860-5 (2024).

- 56 Pendurthi, R. *et al.* Monolithic three-dimensional integration of complementary two-dimensional field-effect transistors. *Nature Nanotechnology* **19**, 970-977, doi:10.1038/s41565-024-01705-2 (2024).
- 57 Ghosh, S. *et al.* Monolithic and heterogeneous three-dimensional integration of two-dimensional materials with high-density vias. *Nature Electronics* **7**, doi:10.1038/s41928-024-01251-8 (2024).
- 58 Prabhathan, P. *et al.* Roadmap for phase change materials in photonics and beyond. *iScience* **26**, doi:10.1016/j.isci.2023.107946 (2023).
- 59 Aabrar, K. A. *et al.* BEOL-Compatible Superlattice FEFET Analog Synapse With Improved Linearity and Symmetry of Weight Update. *IEEE Transactions on Electron Devices* **69**, 2094 - 2100, doi:10.1109/TED.2022.3142239 (2022).
- 60 Oberoi, A., Dodda, A., Liu, H., Terrones, M. & Das, S. Secure Electronics Enabled by Atomically Thin and Photosensitive Two-Dimensional Memtransistors. *ACS Nano* **15**, 19815-19827 (2021).
- 61 Dodda, A., Trainor, N., Redwing, J. M. & Das, S. All-in-One, Bio-Inspired, and Low-Power Crypto Engines for Near-Sensor Security Based on Two-Dimensional Memtransistors. *Nature Communications* **13**, 1-12 (2022).
- 62 Zheng, Y. *et al.* Hardware Implementation of Bayesian Network Based on Two-Dimensional Memtransistors. *Nature Communications* **13**, 1-11 (2022).
- 63 Zhang, S. *et al.* Low Voltage Operating 2D MoS₂ Ferroelectric Memory Transistor with Hf_{1-x}Zr_xO₂ Gate Structure. *Nanoscale Research Letters* **15**, 1-9 (2020).
- 64 Tan, C. *et al.* Gate-Switchable BST Ferroelectric MoS₂ FETs for Non-Volatile Digital Memory and Analog Memristor. *Advanced Functional Materials* **34**, doi:10.1002/adfm.202405293 (2024).
- 65 Aguirre, F. *et al.* Hardware implementation of memristor-based artificial neural networks. *Nature Communications* **15**, doi:10.1038/s41467-024-45670-9 (2024).
- 66 Larcher, L. & Padovani, A. in *IEEE International Conference on Electronics, Circuits and Systems* (IEEE, Athens, Greece, 2010).
- 67 Spassov, D. & Paskaleva, A. Challenges to Optimize Charge Trapping Non-Volatile Flash Memory Cells: A Case Study of HfO₂/Al₂O₃ Nanolaminated Stacks. *Nanomaterials* **13**, doi:10.3390/nano13172456 (2023).
- 68 Paskaleva, A., Rommel, M., Hutzler, A., Spassov, D. & Bauer, A. J. Tailoring the Electrical Properties of HfO₂ MOS-Devices by Aluminum Doping. *ACS Applied Materials & Interfaces* **7**, 17032-17043, doi:10.1021/acsami.5b03071 (2015).
- 69 Spassov, D., Paskaleva, A., Krajewski, T. A., Guziejewicz, E. & Luka, G. Hole and electron trapping in HfO₂/Al₂O₃ nanolaminated stacks for emerging non-volatile flash memories. *Nanotechnology* **29**, doi:10.1088/1361-6528/aae4d3 (2018).
- 70 Hong, Y., Liu, Y., Li, R. & Tian, H. Emerging functions of two-dimensional materials in memristive neurons. *Journal of Physics: Materials* **7**, doi:10.1088/2515-7639/ad467b (2024).
- 71 Dodda, A. *et al.* Active pixel sensor matrix based on monolayer MoS₂ phototransistor array. *Nature Materials* **21**, 1379-1387, doi:10.1038/s41563-022-01398-9 (2022).
- 72 Yoo, S. J. B. *et al.* Towards Reverse-Engineering the Brain: Brain-Derived Neuromorphic Computing Approach with Photonic, Electronic, and Ionic Dynamicity in 3D integrated circuits. *arXiv Preprint*, doi:10.48550/arXiv.2403.19724 (2024).
- 73 Kim, M.-S. & Kim, S. From Oxides to 2D Materials: Advancing Memristor Technologies for Energy-Efficient Neuromorphic Computing. *ACS Applied Electronic Materials* **6**, 3998-4015, doi:10.1021/acsaelm.4c00428 (2024).

Reviewer #1 (Remarks to the Author):

The authors have addressed most of the concerns, except for the following point. Please include this additional comment in the review:

"Following up on Question 12, the authors mentioned they may not have fully understood the question.

To clarify: the question pertains to device-to-device variation. In Figure 1, the threshold voltage exhibits a standard deviation of 0.23 V, and the subthreshold slope shows σ of 55.8 mV/decade. Could you elaborate on how such variations might affect inference accuracy, particularly in terms of potential confusion between similar output states or classifications? A detailed discussion on this aspect would be appreciated."

We are glad that our previous responses adequately addressed most of the reviewer's concerns. Furthermore, we thank the reviewer for clarifying Question 12 for us. We agree that device-to-device variation in threshold voltage and subthreshold slope shown in Figure 1 of the main text (included here as **Figure R1** for reference) could significantly affect inference accuracy and confusion between classes. While we had attributed our deviation between simulated and experimental inference accuracies shown in Figure 3 of the revised manuscript (included here as **Figure R2**) to these factors, this could stand to be made clearer in the main text.

We have adjusted the following statement in the final Abstract to better highlight the need for further testing to reduce device-to-device variation:

Finally, we benchmark the performance of MoS₂ memtransistors against other 2D material-based architectures and project their potential compared to state-of-the-art AI accelerators. We believe that this work furthers the ongoing development of in-memory processors for decentralized edge applications and that future studies aimed at reducing device-to-device

Figure R1. Overview of MoS₂-Memtransistor-based Crossbar Array Architecture. a) Schematic of the basic non-volatile memory (NVM) cell design containing a single memtransistor. Terminals are split into row (drain and gate) and column (source) accesses, with each cell occupying an area of 676 μm². b) Optical image of a representative 16×10 crossbar array based on the design shown in (a). Zoomed-in image shows constituent memtransistors with the drain, source, and gate lines labeled. Scale bar denotes 100 μm (10 μm for zoomed-in). c) Overlapped transfer characteristics, i.e., drain-to-source current (I_{DS}) versus back-gate voltage (V_{BG}), taken at a drain-to-source voltage (V_{DS}) of 1 V from constituent memtransistors. d) Hysteresis loops for a representative device from (b-c) taken at $V_{DS} = 1$ V. Multiple loops were taken by sweeping V_{BG} over the noted ranges to determine the presence/size of the memory window at different gate voltages; a sizable memory window of ~7 V can be noted for the +/- 10 V sweep. e) Three-dimensional scatter plot showing distribution of ON-state and OFF-state currents taken at $V_{DS} = 1$ V, denoted as I_{ON} (pink) and I_{OFF} (cyan), respectively; devices/cells marked in gray registered as an open circuit (OC) when measured. Notably, 158/160 devices were found to work (98.8% yield). f-g) Maps of f) threshold voltage (V_{th}) and g) subthreshold slope (SS) across the array. OC devices are marked. h-i) Histograms of h) V_{th} and i) SS. The means (μ) and standard deviations (σ) are noted for each.

variation and improving long-term non-volatile memory would only enhance inference capabilities.

We have adjusted the following statement to better highlight the connection between device-to-device variation and the deviation between simulation and experiments shown in Figure 3 (Figure R2) of the final manuscript:

Deviations from the ideal case demonstrated in **Figure R2f** can be attributed to device-to-device variation such as that shown in **Figure R1** and the non-ideal yield noted in **Figure R2g**; these factors can be optimized in future work through array-level compensation schemes¹⁻³ and optimization of the fabrication and MoS₂ growth processes.

We have adjusted the following statement in the final Results & Discussion to better highlight the need for further testing to reduce device-to-device variation:

We believe that this work furthers the ongoing development of in-memory processors for decentralized edge applications by demonstrating large-scale, hardware-based inference on MoS₂-based memtransistors and investigating their potential for implementing non-traditional logic operations through exploitation of their multi-terminal nature. Future material and fabrication optimization would serve to tighten variation, improve yield, and enhance the long-term non-volatile memory characteristics of individual devices while inference accuracy could be enhanced at the array-level through dedicated compensation schemes.

Figure R2. Demonstration of MNIST Handwritten Digit Classification on a 2 kb Memtransistor Crossbar Array. Optical images of a) a 1.5×1.5 cm chip containing a 64×32 crossbar array comprising 2,048 MoS₂ memtransistors, b) a zoomed-in image showing the full 2 kb array (scale bar of 100 μm), and c) a further zoomed-in image showing nine constituent memtransistors (scale bar of 10 μm). d) Schematic showing preprocessing performed on training and test (inference) images taken from the Modified National Institute of Standards and Technology (MNIST) database. The original images (28×28 pixels) were downsampled to 13×13 pixels and binarized to fit into a 64×30 sub-array for this demonstration. Downsampled images were then converted to 169×1 input vectors and split into three sub-vectors for input to the array. A dataset comprising 10,000 resized/reshaped images were then used for training and weight assignment; for this demonstration, simulated weights were split between logic “0” and “1” and later converted to targeted conductance states for hardware implementation. e) Heatmap showing the distribution of simulated weights following training. A test dataset of 1,000 resized/reshaped MNIST images was fed to the simulated array for verification of network inference/classification. f) Confusion matrix showing the classification results for the inference check described in (e). An overall accuracy of 88.1% was achieved. g) Heatmap showing the distribution of conductance states assigned to the hardware array in respect to the simulated weight distribution shown in (e), with weights of “1” mapped to a conductance state of ~50 nS and weights of “0” mapped to the OFF-state conductance (a few pS). Cells marked NaN either display an open circuit or high gate leakage; the overall yield of devices remained high at ~92.2%. h) Hysteresis characteristics of the 1,770 working devices in the 64×30 sub-array extracted at $V_{DS} = 1$ V, which show low intrinsic device-to-device variation, memory windows >5 V, and read margins >10⁴. i) Histogram showing the distribution of the final conductance states (weights) represented in (g). j) Confusion matrix showing the classification results for hardware-based inference performed on the memtransistor array shown in (a-c) as per the conductance state (weight) distribution shown in (g). A test dataset comprising 1,000 resized/reshaped MNIST images was applied to the drain terminals of the array as voltage inputs (either 0 V or 1 V depending on the corresponding pixel value); output currents along corresponding columns/nodes were then individually registered and compared to all other outputs to determine the inferred digit for each case. An overall accuracy of 85.6% was registered.

Reviewer #2 (Remarks to the Author):

My early concerns have been addressed by the authors. I thus recommend it for publication.

We are glad that our previous responses adequately addressed the reviewer’s concerns and thank the reviewer for their recommendation for publication in *Nature Communications*.

Reviewer #3 (Remarks to the Author):

The authors have addressed most of the questions in the revised manuscript and supporting information. However, the retention results of MoS₂ memtransistor show the drain current decays all the time until it reaches the off state. The authors should not claim that the device could achieve retention exceeding 3 years. Moreover, the MoS₂ memtransistor shows ~10 times degradation in 500 cycles. How about the endurance after

more cycles, such as 1E9 cycles, which are typical operation for NVMs endurance. From the retention and endurance tests, MoS₂ memtransistor in this work could not be considered as a NVM cell. Both retention and endurance will influence the training and inference of ANN. The authors should consider this impact in the work.

We are glad that our previous responses adequately addressed most of the reviewer's concerns. We also understand the reviewer's concerns regarding long-term memtransistor retention/endurance, which would require substantial further testing beyond the immediate scope of this work. We have therefore adjusted our claims to more accurately convey the results discussed in the final manuscript (i.e., a projection of long-term retention based on short-term experiments) and highlight the need for further memory testing.

We have removed all claims of long-term non-volatile retention exceeding three years in the final manuscript and have instead attempted to better emphasize the projected nature of these results.

We have adjusted the following statement in the final Abstract to more accurately convey our experimental results with long-term projections and better highlight the need for further testing to experimentally confirm non-volatility and long-term retention:

Here, we present dense, large-scale crossbar array architectures incorporating up to 2,048 MoS₂ memtransistors per array, achieving >92% yield across multiple arrays while individual memtransistors exhibit write energies as low as ~0.2 fJ, maintain read margins up to 10⁵, and offer a projected retention exceeding three years. These architectures demonstrate the ability to resolve inference ambiguities through gate modulation without the need for costly retraining or reprogramming. We also validate their performance by successfully classifying handwritten digits from the MNIST database. Finally, we benchmark the performance of

MoS₂ memtransistors against other 2D material-based architectures and project their potential compared to state-of-the-art AI accelerators. We believe that this work furthers the ongoing development of in-memory processors for decentralized edge applications and that future studies aimed at reducing device-to-device variation and improving long-term non-volatile memory would only enhance inference capabilities.

We have adjusted the following statement in the final Introduction section to more accurately convey our experimental results with long-term projections:

We achieved a high yield of >92% across multiple crossbar array demonstrations, with constituent devices displaying write energies as low as ~0.2 fJ while retaining read-margins (ratio between programmed and erased states) as high as 10⁵ and projected retentions exceeding 3 years.

We have adjusted the following statement in the final Results & Discussion section to more accurately convey our experimental results with long-term projections:

A high yield of >92% with low device-to-device variation was confirmed for each array investigated as part of this work, with constituent devices displaying switching energies as low as ~0.2 fJ, read margins as high as 10⁵, and projected retentions exceeding 3 years.

We have adjusted the following statement in the final Results & Discussion section to better highlight the need for further testing to experimentally confirm non-volatility and long-term retention:

We believe that this work furthers the ongoing development of in-memory processors for decentralized edge applications by demonstrating large-scale, hardware-based inference on MoS₂-based memtransistors and investigating their potential for implementing non-traditional logic operations through exploitation of their multi-terminal nature. Future

material and fabrication optimization would serve to tighten variation, improve yield, and enhance the long-term non-volatile memory characteristics of individual devices while inference accuracy could be enhanced at the array-level through dedicated compensation schemes.

Rebuttal References

- 1 Jain, S. & Raghunathan, A. CxDNN: Hardware-software Compensation Methods for Deep Neural Networks on Resistive Crossbar Systems. *ACM Transactions on Embedded Computing Systems* **18**, 1-23, doi:10.1145/3362035 (2019).
- 2 Li, B., Xia, L., Gu, P., Wang, Y. & Yang, H. in *Design Automation Conference*. (Association for Computing Machinery).
- 3 Zidan, M. A., Omran, H., Sultan, A., Fahmy, H. A. H. & Salama, K. N. Compensated Readout for High-Density MOS-Gated Memristor Crossbar Array. *IEEE Transactions on Nanotechnology* **14**, 3-6, doi:10.1109/TNANO.2014.2363352 (2014).